

# Differences in tropical high clouds among reanalyses: origins and radiative impacts

Jonathon S. Wright[1], Xiaoyi Sun[1], Paul Konopka[2], Kirstin Krüger[3], Andrea M. Molod[4], Susann Tegtmeier[5], Guang J. Zhang[6], and Xi Zhao[7]

[1]Ministry of Education Key Laboratory for Earth System Modeling, Department of Earth System Science, Tsinghua University, Beijing, China
[2]Forschungszentrum Jülich (IEK-7: Stratosphere), Jülich, Germany
[3]Department of Geosciences, University of Oslo, Oslo, Norway
[4]Global Modeling and Assimilation Office, NASA Goddard Space Flight Center, Greenbelt, Maryland, USA
[5]Department of Physics and Engineering Physics, University of Saskatchewan, Saskatoon, Canada
[6]Scripps Institution of Oceanography, La Jolla, California, USA
[7]Department of Atmospheric Sciences, Texas A&M University, College Station, Texas, USA

**Correspondence:** Jonathon S. Wright (jswright@tsinghua.edu.cn)

**Abstract.** We examine differences among reanalysis high cloud products in the tropics, assess the impacts of these differences on radiation budgets at the top of the atmosphere and within the tropical upper troposphere and lower stratosphere (UTLS), and discuss their possible origins in the context of the reanalysis models. We focus on the ERA5, ERA-Interim, JRA-55, MERRA-2, and CFSR/CFSv2 reanalyses, with MERRA included in selected comparisons. As a general rule, JRA-55 produces the
smallest tropical high cloud fractions and cloud water contents among the reanalyses, while MERRA-2 produces the largest. Accordingly, cloud radiative effects are relatively weak in JRA-55 and relatively strong in MERRA-2. Only MERRA-2 and ERA5 among the reanalyses produce tropical-mean values of outgoing longwave radiation (OLR) close to observed, but ERA5 tends to underestimate cloud effects while MERRA-2 tends to overestimate variability. ERA5 also produces distributions of longwave, shortwave, and total cloud radiative effects at top-of-atmosphere that are very consistent with observed. The other
reanalyses all exhibit substantial biases in at least one of these metrics, although compensation between the longwave and shortwave effects helps to constrain biases in the total cloud effect for most reanalyses. The vertical distribution of cloud water content emerges as a key difference between ERA-Interim and the other reanalyses. Whereas ERA-Interim shows a monotonic decrease of cloud water content with increasing height, the other reanalyses all produce distinct anvil layers. The latter is in better agreement with observations and yields very different profiles of radiative heating in the UTLS. For example, whereas
the altitude of the level of zero net radiative heating tends to be lower in convective regions than in the rest of the tropics in ERA-Interim, the opposite is true for the other four reanalyses. Differences in cloud water content also help to explain systematic differences in diabatic ascent in the tropical lower stratosphere among the reanalyses. We discuss several ways in which aspects of the cloud and convection schemes impact the tropical environment. Discrepancies in the vertical profile of moist static energy in convective regions are particularly noteworthy, as this metric is based exclusively on variables that are
directly constrained by data assimilation.



# 1 Introduction

Tropical high clouds play a central role in climate via their ability to modulate the radiation budget, altering both the reflection of incoming solar radiation and the atmospheric absorption of longwave radiation emitted by the Earth's surface (Trenberth et al., 2009; Dessler, 2010). The net effect of an individual cloud on the radiation budget depends on several factors, including

the type, phase, height, and microphysical characteristics of the cloud (Stevens and Schwartz, 2012). These features are difficult to parameterize, so that the integrated radiative impacts of clouds remain poorly represented in global models (Bony et al., 2015), including those used to produce atmospheric reanalyses (Dolinar et al., 2016; Li et al., 2017).

Clouds, circulation, and sea surface temperature (SST) are strongly coupled in the tropics (e.g. Hartmann and Michelsen, 1993; Emanuel et al., 1994; Fu et al., 1996; Su et al., 2011). These coupled interactions transport energy away from convec-

tive regions, which tend to be anchored over the warmest SSTs, into subsidence-dominated regions where SSTs are usually cooler. Associated tracer transports have extensive influences on humidity, ozone, and other constituents in the upper troposphere (Folkins et al., 2002; Jiang et al., 2007; Fiehn et al., 2017; Pan et al., 2017), while momentum transport, latent heat release, and radiative effects modulate circulation patterns in both the troposphere and stratosphere (LeMone et al., 1984; Carr and Bretherton, 2001; Lane and Moncrieff, 2008; Geller et al., 2016; Kim et al., 2017). Changes in precipitation are governed

to leading order by the balance of changes in radiative cooling and condensational heating in the atmosphere (O'Gorman et al., 2011), both of which are intimately connected with the distribution and properties of high clouds. The radiative and condensational heating effects of clouds have also been shown to influence atmospheric water budgets associated with a wide range of climatological phenomena, including the El Niño–Southern Oscillation (e.g. Posselt et al., 2011), the Madden–Julian Oscillation (e.g. Anber et al., 2016; Cao and Zhang, 2017), and the South Asian summer monsoon (e.g. Wang et al., 2015).

Given the influential role of high clouds in the tropical climate system and the complexity of their interactions with other variables, evaluation and intercomparison of reanalysis cloud products serves several purposes. First, reanalyses offer global coverage at relatively high resolution and regular intervals. It is therefore useful to assess the level to which reanalysis cloud and radiation products may be considered 'realistic'. Second, systematic differences in cloud fields can be used to diagnose problems or points of concern in the atmospheric model. Detailed evaluation of these biases can thus inform both interpretation

of model outputs and future efforts toward model development. Differences in cloud fields may likewise indicate pervasive biases in the model background state that influence more widely-used reanalysis products, such as temperatures and winds. Data assimilation helps to mitigate these effects in variables that are analyzed, but the extent of this mitigation depends on the availability and quality of assimilated observations (and thus varies in time and space), as well as the assimilation method used to combine observations with the model background state. No such mitigation can be expected for forecast-only variables that

are not analyzed, such as the radiative heating rates often used to drive diabatic transport simulations in the upper troposphere and stratosphere (Wright and Fueglistaler, 2013; Tao et al., 2019). Data assimilation may even exacerbate disagreements among these variables if the analysis pulls the model away from its internal equilibrium state.

Cloud fields in reanalyses are essentially model products, but many variables that influence the distribution of clouds in the tropics are altered during the data assimilation step (e.g. atmospheric temperatures, moisture, and winds). We therefore





anticipate that differences in cloud fields among reanalyses may arise from several factors, including the prescribed boundary conditions (such as SST), the physical parameterizations used in the atmospheric models (especially those pertaining to convection and large-scale condensation), the implementation of data assimilation, and the assimilated data (particularly satellite data from infrared and all-sky microwave humidity sounders). Traditional 3-dimensional variational (3D-Var) or 'first guess at appropriate time' (3D-FGAT) assimilation techniques provide only indirect constraints on cloud fields via the use of previous analyzed states to initialize subsequent forecasts. Constraints on cloud fields might be tightened by several approaches used in recent reanalyses, such as the incremental analysis update (IAU) and incremental 4-dimensional variational (4D-Var) methods. Under IAU, assimilation increments in analyzed fields are applied gradually during a 'corrector' forecast after they are calculated (Bloom, 1996; Takacs et al., 2018). Under incremental 4D-Var, the assimilation scheme iteratively adjusts the entire forecast to optimize the fit between the full temporal evolution of the model state and the available observations (Courtier et al., 1994). Both of these approaches produce cloud fields that are more consistent with analyzed temperatures, humidities, and winds, although this internal consistency is still governed by parameterized representations of sub-grid physics.

The purpose of this paper is to examine and evaluate upper tropospheric cloud fields in the tropics (30°S–30°N) as represented in recent atmospheric reanalyses, to identify differences among these reanalyses, and to explore the potential reasons behind these differences. We consider the fractional coverage of high clouds, total condensed water content in the tropical upper troposphere, and the radiative effects of clouds, both at the nominal top-of-atmosphere (TOA) and within the upper troposphere and lower stratosphere (UTLS). Our approach differs from and builds on other recent efforts in this direction (e.g. Dolinar et al., 2016) through an exclusive focus on tropical high clouds ($p < 500$ hPa), a deeper exploration of co-variability at daily time scales in addition to monthly means, discussion of cloud–radiation interactions in the tropical UTLS in addition to TOA fluxes, and the inclusion of some recently-released reanalyses. We also endeavor to systematically document key differences in parameterizations of clouds and radiation among the reanalyses, and discuss some of the ways these differences impact the state of the tropical atmosphere as represented in recent reanalyses.

We briefly introduce the reanalysis products, observationally-based data sets, and methodology in Sect. 2, with more detailed descriptions of the cloud and radiation parameterizations used in the reanalyses collected in Appendix A. In Sect. 3, we summarize the climatological distributions of high cloud fraction, total condensed water content, and outgoing longwave radiation produced by reanalyses in the tropics. In Sect. 4, we examine how differences in the distribution and properties of high clouds alter radiative fluxes and exchange at daily scales in the deep tropics, both at the TOA and within the tropical UTLS. In Sect. 5, we explore the potential origins of differences in high clouds in the context of different reanalysis model treatments of deep convection and in situ cloud formation near the tropical tropopause. In Sect. 6, we briefly assess temporal variability and agreement amongst the reanalyses. We close the paper in Sect. 7 by summarizing the results and providing recommendations and context for reanalysis data users.





## 2 Data and methodology

### 2.1 Reanalysis products

Our intercomparison focuses mainly on five relatively recent atmospheric reanalyses: the fifth generation European Centre for Medium-range Weather Forecasts (ECMWF) reanalysis (ERA5; Hersbach et al., 2018), the ECMWF Interim Reanalysis (ERA-Interim; Dee et al., 2011), the Japanese 55-year Reanalysis (JRA-55; Kobayashi et al., 2015), the Modern-Era Retrospective Analysis for Research and Applications, Version 2 (MERRA-2; Gelaro et al., 2017), and the Climate Forecast System Reanalysis (CFSR; Saha et al., 2010) and its extension via the Climate Forecast System Version 2 (CFSv2; Saha et al., 2014). The earlier MERRA reanalysis (Rienecker et al., 2011) is included in selected comparisons. All six of these products are 'full-input' reanalyses in that they assimilate both conventional and satellite data; however, they differ from each other with respect to their atmospheric models, assimilation techniques, and assimilated data sets. We document key details of the cloud, convection, and radiation schemes in Appendix A. Readers interested in these technical details and how they imprint on the intercomparisons presented in this paper may wish to consult this appendix before proceeding to the results. Other relevant aspects of most of these reanalysis systems (with the exception of ERA5) have recently been reviewed by Fujiwara et al. (2017). An expanded review (including ERA5) is provided in Chapter 2 of the forthcoming SPARC Reanalysis Intercomparison Project (S-RIP) report (Wright et al., 2019, in preparation; digital version for review available at https://jonathonwright.github.io/S-RIPChapter2E.pdf).

The full intercomparison period covers January 1980 through December 2014 and includes all six reanalyses. We also conduct a more sophisticated intercomparison of daily co-variability among key variables from five of the reanalyses (ERA5, ERA-Interim, JRA-55, MERRA-2, and CFSR), which covers January 2001–December 2010. Results for the full intercomparison are presented in Sects. 3, 4, and 6, while results based on daily co-variability are presented in Sects. 4 and 5. The intercomparison period includes the CFSR–CFSv2 transition in January 2011, as well as the intermediate year 2010 (as discussed by Fujiwara et al., 2017, among others). We show in Sect. 6 below that both transitions involved large changes in the cloud fields, much larger than the discontinuities at earlier production stream transitions. The January 2011 transition to CFSv2 also involved changes in the atmospheric model formulation governing interactions between clouds and radiation.

### 2.2 Observational data

We use several observationally-based data products to supply context, including TOA radiative fluxes, cloud fraction, cloud ice water content, and atmospheric thermodynamic state variables (Table 1). Observations of these variables are subject to a number of uncertainties, including lack of sensitivity to optically thin clouds or clouds composed of small particles (e.g. Dessler and Yang, 2003), uncertainties caused by overlapping cloud layers (e.g. Zhang et al., 2005), errors in cloud top height (e.g. Sherwood et al., 2004), and diurnal sampling biases (e.g. Fowler et al., 2000; Hearty et al., 2014). As our primary focus is on the intercomparison of reanalysis products, we have opted not to apply a satellite cloud observation simulator (e.g. Bodas-Salcedo et al., 2015; Stengel et al., 2018) to the reanalysis outputs. Accordingly, we stress that the most comparisons between reanalysis products and observational data presented in this paper are qualitative rather than quantitative.



**Table 1.** Summary of observational data sets, listed in alphabetical order by project. TOA stands for top-of-atmosphere and UT for upper troposphere, where the latter comprises pressures less than 500 hPa for CERES SYN1Deg and pressures less than 440 hPa for ISCCP and MODIS. Other abbreviations are defined in the text.

| Project | Product | Version | Variables | Period | Timestep | Grid | Levels | Reference |
|---|---|---|---|---|---|---|---|---|
| AIRS | TqJoint | v6 | $T, q, z$ | 2003–2010 | daily | 1° | 12 ($p$) | Texeira (2013) |
| CERES | EBAF | Ed4A | TOA radiation | 2001–2014 | monthly | 1° | TOA | Kato (2017) |
| CERES | SYN1Deg | Ed4A | TOA radiation | 2001–2010 | daily | 1° | TOA | Doelling (2017) |
| CERES | SYN1Deg | Ed4A | high cld cover | 2001–2010 | daily | 1° | UT | Doelling (2017) |
| CFMIP2 | GOCCP | v3.1.2 | cld frac profile | 2007–2014 | monthly | 2° | 40 ($z$) | Chepfer et al. (2010) |
| CloudSat | derived | v1 | cld frac profile | 2007–2010 | monthly | 2° | 40 ($z$) | Kay and Gettelman (2009) |
| CloudSat | CWC-RO | v5.1r4 | IWC profile | Aug 2006–Jul 2007 | monthly | 1° | 40 ($z$) | Austin et al. (2009) |
| ISCCP | HGM | v1 | high cld cover | 1984–2014 | monthly | 1° | UT | Rossow et al. (2017) |
| Terra MODIS | MOD08 | c6 | high cld cover | 2001–2014 | monthly | 1° | UT | Platnick (2015) |
| NASA-GEWEX | SRB | r3.1 | TOA radiation | 1984–2007 | monthly | 1° | TOA | Zhang et al. (2015) |

The International Satellite Cloud Climatology Project (ISCCP) has produced observationally-based descriptions of clouds and their attributes using geostationary and polar-orbiting satellite measurements (Rossow and Schiffer, 1991). We use the H-series Global Monthly (HGM) product for January 1984–December 2014 (Rossow and Schiffer, 1999; Rossow et al., 2017). As a supplement to the ISCCP cloud data, we use all-sky and clear-sky fluxes of longwave (LW) radiation at the TOA from the

NASA Global Energy and Water Cycle Experiment (GEWEX) Surface Radiation Budget (SRB) project covering January 1984 through December 2007 (Stackhouse et al., 2011; Zhang et al., 2015). These data are based on radiative calculations that combine observed fluxes and ozone with Goddard Earth Observing System Data Assimilation System, Version 4 (GEOS-4) analyses of temperature and water vapour. Pixel-level data from ISCCP are used to estimate cloud radiative effects in SRB.

   We use several products from the Clouds and the Earth's Radiant Energy System (CERES) experiment (Wielicki et al.,

1996). First, we use time-mean TOA fluxes calculated from Energy Balanced and Filled (EBAF) monthly-mean products at 1°×1° spatial resolution (Kato, 2017). Second, we use daily-mean Synoptic Radiative Fluxes and Clouds (SYN1Deg) products at 1°×1° spatial resolution (Doelling, 2017). The SYN1Deg data set represents an intermediate step in the production of the monthly EBAF dataset. SYN1Deg provides several estimates of TOA radiative fluxes, including direct measurements, outputs from initial 'untuned' radiative transfer model simulations, and outputs from a second set of radiative transfer simulations in

which the model input variables are adjusted to bring the simulated fluxes into better agreement with the observed fluxes. The initial atmospheric state for radiative computations is taken from the GEOS-5 data assimilation system, a different version of which is used for MERRA-2. Only the final adjusted fluxes are discussed, as these products are most appropriate for computing cloud radiative effects for comparison with reanalysis estimates. The results are similar when the direct measurements are used instead. Along with TOA radiative fluxes, the SYN1Deg data set includes estimates of cloud fraction retrieved using

measurements collected by the Moderate-Resolution Imaging Spectroradiometer (MODIS) and geostationary satellites (Minnis





et al., 2011; Doelling et al., 2013). We also use high cloud fractions from Collection 6 of the Terra MODIS Level 3 Atmosphere Product (MOD08; Platnick, 2015).

For observations of the thermodynamic state of the atmosphere, we use level 3 data from the Atmospheric Infrared Sounder (AIRS) version 6 'TqJoint' collection (Texeira, 2013). This data set provides gridded representations of temperature, moisture,

and other fields based on a consistent set of initial retrievals in each grid cell (Tian et al., 2013). As the finest temporal resolutions of other data examined in this study are daily means, we average data from ascending and descending passes together. Variables taken from AIRS TqJoint include temperature, water vapor mass mixing ratio, and geopotential height between January 2003 and December 2014.

Finally, we examine three products deriving from CloudSat and Cloud-Aerosol Lidar and Infrared Pathfinder Satellite Obser-

vation (CALIPSO) measurements. These include two monthly estimates of cloud fraction vertical profiles, one based on combined information from CloudSat and CALIPSO (Kay and Gettelman, 2009) and one based on CALIPSO alone (Chepfer et al., 2010). We use the combined CloudSat–CALIPSO product for the four years 2007–2010 and the GCM-Oriented CALIPSO Cloud Product (GOCCP) for the eight years 2007–2014. The first product was discontinued after CloudSat switched to sunlit-only observations in early 2011. We also use ice water content (IWC) measurements from CloudSat based on version 5.1

(release 4) of the radar-only retrieval algorithm (CWC-RO; Austin et al., 2009), mapped onto a $1° \times 1°$ grid and averaged over the 12 months from August 2006 through July 2007 (see also Zhao et al., 2017). CloudSat measurements in units of $\mathrm{mg\,m^{-3}}$ are converted to mass mixing ratios using dry-air densities estimated from a tropical-mean temperature profile computed using AIRS observations from the same period. CloudSat- and CALIPSO-based data sets are provided on a 40-level height grid, which we convert to pressure using the barometric equation with a constant scale height of 7.46 km. This approach introduces

uncertainty in the precise vertical location (in pressure coordinates) of features observed by CloudSat and CALIPSO, which should be taken into consideration when comparing these features to those produced by the reanalyses.

## 2.3 Derived variables and statistical treatments

We use several classes of variables in this intercomparison. Variables directly related to tropical high clouds include high cloud fraction and vertical profiles of cloud fraction and cloud water content, while variables used to explore the impacts of differ-

ences in high clouds include TOA radiative fluxes and vertically-resolved radiative heating rates within the upper troposphere, tropopause layer, and lower stratosphere. All vertically-resolved variables are evaluated on pressure levels, interpolated from height or model levels when necessary. Cloud radiative effects are computed as clear-sky minus all-sky fluxes using positive-upward fluxes at the TOA, so that LW effects are generally positive (the presence of clouds reduces OLR) and SW effects are generally negative (the presence of clouds increases the planetary albedo). We denote the LW cloud radiative effect as LWCRE

and the SW effect as SWCRE.

Variables used to diagnose the potential origins of differences in high clouds include SST, vertical velocity at 500 hPa, and vertical profiles of temperature, specific humidity, and geopotential height between 1000 and 100 hPa. The latter three variables



are used to compute moist static energy:

$$\text{MSE} = gz + c_p T + L_v q, \tag{1}$$

where $g$ is gravitational acceleration in Earth's lower atmosphere, $z$ is geopotential height, $c_p$ is the specific heat constant for dry air, $T$ is temperature, $L_v$ is latent enthalpy of vaporization at 0°C, and $q$ is specific humidity. Temperature and specific humidity are also used to calculate equivalent potential temperature ($\theta_e$), which is then used to diagnose the potential instability of the lower troposphere as the difference in equivalent potential temperature between the lower troposphere (850 hPa) and the middle troposphere (500 hPa):

$$\text{PI} = \theta_{e,850} - \theta_{e,500}. \tag{2}$$

Equivalent potential temperature is computed according to the formula proposed by Bolton (1980) using the MetPy software package (May et al., 2008 - 2018). Relative humidity (RH) is calculated with respect to liquid water using MetPy. This approach avoids inconsistencies in the implementation of the liquid–ice transition among the different datasets (see Appendix A1). Ratios between saturation vapor pressures with respect to ice and with respect to liquid water are provided for context, calculated using the empirical formulas proposed by Emanuel (1994). The level of zero radiative heating (LZRH) is determined for all profiles for which the daily-mean radiative heating rate is positive at 100 hPa. Radiative heating rates are linearly interpolated onto a 1000-level grid between 100 hPa and 500 hPa with equal spacing in $\ln(p)$. The LZRH is then defined as the largest pressure for which all radiative heating rates are positive between 100 hPa and that level (inclusive).

Statistical treatments mainly consist of composite averages or distributions conditioned on ranked quartiles of the LWCRE (i.e. four bins separated by the 25[th] percentile, the median, and the 75[th] percentile). Averages taken in the horizontal dimension are weighted by relative area. Two-dimensional kernel density estimates are computed using the $k$-dimensional tree-based implementation in SciKit-Learn (Pedregosa et al., 2011) with a Gaussian kernel. Optimal bandwidths for kernel density estimates are identified using a 20-fold grid-search cross-validation on randomly selected subsets of the data, and consistently converge to values near 1 (0.8–1.3) for the LWCRE and values near 2 (1.5–2.4) for the SWCRE.

## 3 Climatological distributions

We define a multi-reanalysis mean (MRM) to facilitate our intercomparison of cloud and radiation climatologies produced by the reanalyses climatological condition, calculated by averaging products from the ERA-Interim, JRA-55, MERRA-2, and CFSR/CFSv2 reanalyses. ERA5 and MERRA are omitted from the MRM, ERA5 because it was made available relatively late in the S-RIP activity and MERRA because we do not include MERRA products beyond this section of the paper. Figure 1 shows the time-mean distributions of high cloud cover in the tropics based on this MRM (1980–2014) and the ISCCP HGM observationally-based analysis (1984–2014). Differences of the six individual reanalyses relative to the MRM are also shown. Area-weighted mean values of high cloud fraction averaged over the tropics (30°S–30°N) are noted for each product. The definition of high cloud fraction varies somewhat among these data sets. For example, high clouds are defined at pressures

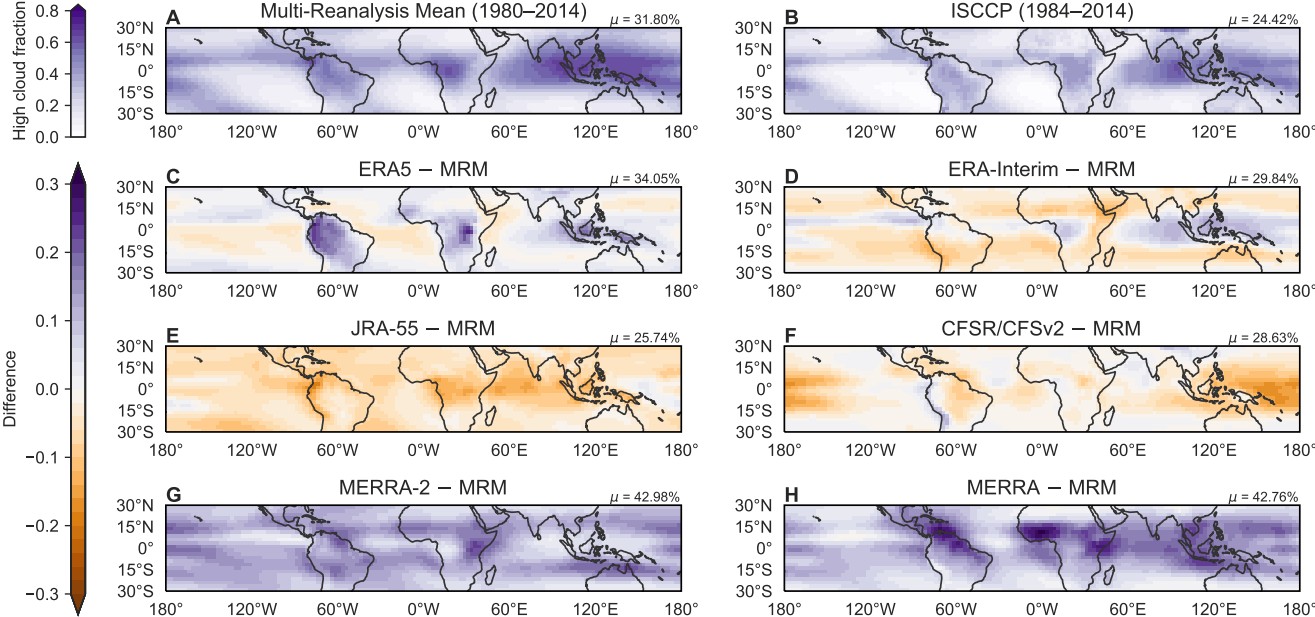

**Figure 1.** Climatological mean spatial distributions of high cloud cover for (A) the multi-reanalysis mean (MRM) over 1980–2014, calculated by averaging the distributions for ERA-Interim, JRA-55, MERRA-2, and CFSR, and (B) ISCCP over 1984–2014. Differences relative to the MRM are shown for (C) ERA5, (D) ERA-Interim, (E) JRA-55, (F) CFSR/CFSv2, (G) MERRA-2, and (H) MERRA over 1980–2014. The area-weighted tropical mean (30°S–30°N) high cloud fraction based on each product is shown at the upper right corner of the corresponding panel.

less than ∼500 hPa for JRA-55, pressures less than ∼400 hPa for CFSR/CFSv2, MERRA, and MERRA-2, and pressures less than ∼450 hPa for ERA-Interim and ERA5. For the observational data sets, high cloud fractions are based on clouds with tops diagnosed at pressures less than 440 hPa in ISCCP and MODIS, and at pressures less than 500 hPa in CERES SYN1Deg. We show below that reanalysis-derived cloud fraction profiles have minima between 400 and 500 hPa in the tropics, so that
5 differences in the precise definition of high cloud fraction should not greatly impact qualitative comparisons based on Fig. 1.

High cloud fractions in JRA-55 are almost exclusively smaller than the MRM, while those in MERRA-2 are larger than the MRM. Negative biases in JRA-55 are largest over canonical deep convective regions, such as the equatorial Indian Ocean, equatorial Africa, and the Maritime Continent. By contrast, positive biases in MERRA-2 are largest along the flanks of the deep convective regions. Tropical mean values of high cloud fraction in ERA-Interim and CFSR/CFSv2 are close to the
10 MRM; however, the spatial patterns of biases relative to the MRM are qualitatively opposite between these two reanalyses. Whereas ERA-Interim produces high cloud fractions larger than the MRM in the deep convective regions of the tropics (e.g. over the Maritime Continent and equatorial Indo–Pacific domain), CFSR/CFSv2 produces high cloud fractions smaller than the MRM in deep convective regions (especially over the western Pacific). ERA-Interim typically underestimates the MRM outside of the canonical deep convective regions, while CFSR/CFSv2 produces larger high cloud fractions over mountainous



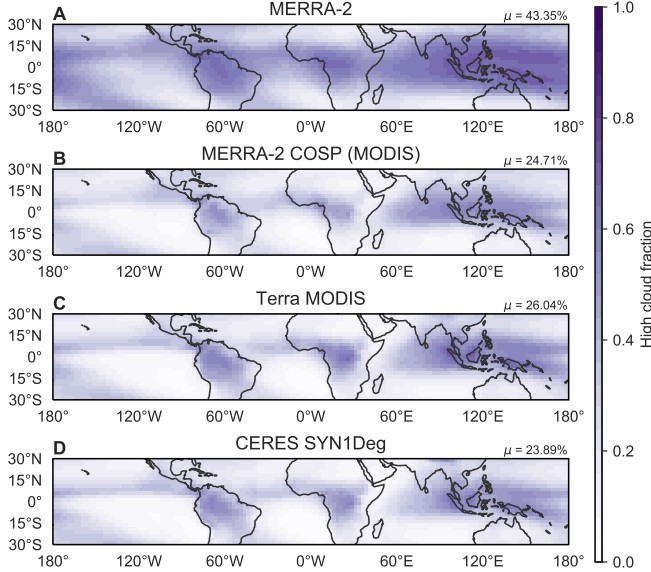

**Figure 2.** As in Fig. 1a, but for (A) direct output from MERRA-2, (B) MERRA-2-COSP (emulating MODIS observations of the MERRA-2 atmosphere), (C) Terra MODIS, and (D) CERES SYN1Deg (based primarily on Terra and Aqua MODIS) for 2001–2014. The area-weighted tropical mean (30°S–30°N) high cloud fraction based on each product is shown at the upper right corner of the corresponding panel.

regions, such as the Andes and the Tibetan Plateau. Differences between ERA5 and the MRM are similar in many ways to those found for ERA-Interim, but with further enhancements in tropical convective regions (especially over land). ERA5 has noticeably larger high cloud fractions than ERA-Interim in tropical South America and Africa, as well as in the South Asian monsoon region, the Pacific portion of the ITCZ, and the SPCZ. These differences contribute to an increase of 0.04

(4%) in the tropical mean high cloud fraction between ERA-Interim and ERA5. Additional analysis is necessary to assess which of these is more consistent with the actual distribution of high clouds. However, Bechtold et al. (2014) reported that modifications to parameterized convection in the ECMWF atmospheric model implemented between ERA-Interim and ERA5 yielded lower biases against observed brightness temperatures in land convective regions, especially for channels sensitive to the upper troposphere.

Initial comparison with high cloud fractions from ISCCP D2 suggests that the reanalyses systematically overestimate high cloud cover, with the tropical mean estimate from JRA-55 (25.74%) falling closest to that based on ISCCP D2 (23.93%). However, as discussed at the beginning of section 2, direct comparisons between cloud variables derived from observations and those derived from models may be misleading. MERRA-2 provides outputs from the Cloud Feedback Model Intercomparison Project (CFMIP) Observation Simulator Package (COSP; Bodas-Salcedo et al., 2015) as an ancillary product in the reanalysis.

Included in this product are estimates emulating high cloud fraction as observed by MODIS. Whereas MERRA-2 produces a tropical mean high cloud fraction of 42.98% (rising slightly to 43.35% during 2001–2014), the MERRA-2 COSP product indicates that MODIS would observe a tropical mean high cloud fraction of only 24.71%. This latter estimate is in good agreement





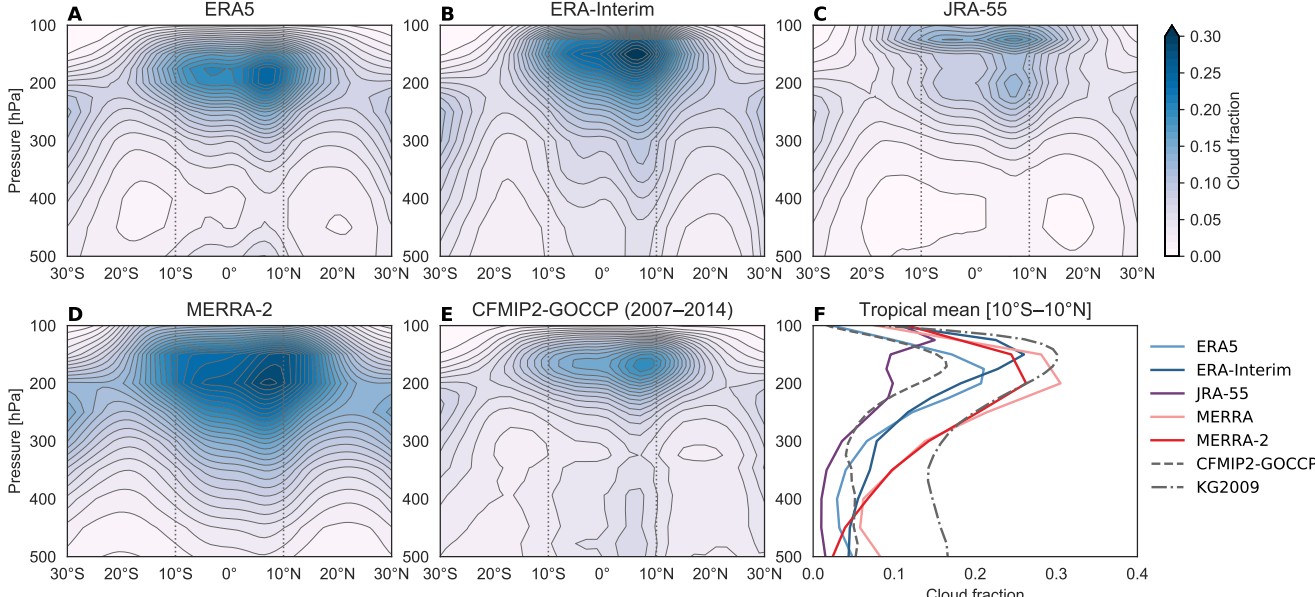

**Figure 3.** Time-mean zonal-mean distributions of cloud fraction based on the (A) ERA5, (B) ERA-Interim, (C) JRA-55, and (D) MERRA-2 reanalyses (1980–2014), along with (E) an observationally-based distribution from the GOCCP CALIPSO-based product produced for CFMIP2 (Chepfer et al., 2010), which covers 2007–2014. Inner tropical mean (10°S–10°N) profiles of cloud fraction based on these five estimates are shown in (F), along with profiles from MERRA and the combined CloudSat–CALIPSO product derived by Kay and Gettelman (2009). The latter is averaged over 2007–2011. Vertical lines in (A) through (E) mark the bounds of the averaging domain. CFSR does not provide vertical profiles of cloud fraction.

with both the Terra MODIS (26.04%) and CERES SYN1Deg (23.89%) gridded products over the same period, both in terms of the tropical mean value and the spatial distribution of high cloud cover (Fig. 2). The most pronounced difference between the standard MERRA-2 product and the MERRA-2 COSP product is a reduction in cloud fractions outside the canonical deep convective regions of the tropics, suggesting that the large high cloud fractions produced by MERRA-2 in these locations

5  are associated with optically thin clouds having small water paths, which cannot be readily observed by MODIS. The close agreement between MERRA-2 COSP and corresponding observational estimates does not necessarily mean that the larger high cloud fractions in MERRA-2 are more realistic (i.e. that the other reanalyses substantially underestimate high cloud fraction in the tropics). Rather, it indicates only that MERRA-2 produces a reasonably realistic distribution of the high clouds that can be readily observed by passive infrared instruments like MODIS. Affirming this point, a recent study in which a cloud simulator

10  was applied to ERA-Interim outputs also indicates good agreement with observed high cloud fractions in the tropics, with a slight high bias ($\leq 10\%$) in the same inner tropical regions where ERA-Interim overestimates the MRM (Stengel et al., 2018).

    Figure 3 shows time-mean zonal-mean distributions of cloud fraction in the tropical upper troposphere as functions of latitude and pressure. ERA5, ERA-Interim, JRA-55, and MERRA-2 all show maxima in cloud fraction near the base of the



tropical tropopause layer. The peak value in ERA-Interim is centered at 150 hPa, slightly above that in ERA5 (∼175 hPa) and MERRA-2 (∼200 hPa) and slightly below that in JRA-55 (∼125 hPa). JRA-55 also shows a secondary maximum near 200 hPa. All of these maxima are most pronounced in the Northern Hemisphere between 5°N and 10°N, reflecting the preferred position of the intertropical convergence zone (ITCZ). CFSR does not provide vertical profiles of cloud fraction and is therefore not

represented in Fig. 3.

Observationally-based estimates of vertically-resolved cloud fraction from CALIPSO (CFMIP2-GOCCP; Chepfer et al., 2010) are shown in Fig. 3E and Fig. 3F, with a tropical mean profile based on CloudSat and CALIPSO (KG2009; Kay and Gettelman, 2009) also included in Fig. 3F. The distribution based on KG2009 is qualitatively similar to that based on CFMIP2-GOCCP and is therefore omitted from Fig. 3; however, these two datasets show large differences in the magnitude of cloud

fraction within the tropical upper troposphere (Fig. 3F). The range of cloud fractions spanned by the two observationally-based estimates is comparable to that spanned by the reanalysis products. Like the reanalyses, the observational products indicate that the maximum cloud fraction is located in the Northern Hemisphere tropics. The vertical placement of this maximum is around 150–175 hPa, between that produced by ERA-Interim and that produced by ERA5. This implies that the altitude of the maximum in MERRA-2 is slightly too low, although the relatively coarse vertical resolution of the MERRA-2 pressure-

level product in this region and uncertainties associated with height–pressure conversion for the observational estimates (see Sect. 2.2) reduce our confidence in this conclusion. We find firmer ground in interpreting some of the other differences in Fig. 3. First, the bimodal structure of the cloud fraction profile and the extremely high altitude of the peak values (125 hPa) are unique to JRA-55. Together with the relatively small values of high cloud cover in JRA-55 (Fig. 1E), we conclude that this reanalysis underestimates high cloud fraction through most of the tropical upper troposphere. Second, the observational estimates indicate

secondary maxima in cloud fraction between 400–500 hPa, while most of the reanalyses produce local minima in this region. This difference suggests that the reanalysis models systematically underestimate the depth, frequency, or amount of cloud detrained by cumulus congestus in the tropics (Johnson et al., 1999).

Differences among the reanalyses are even more pronounced with respect to time-mean zonal-mean distributions of cloud water content (CWC) in the tropical upper troposphere (Fig. 4). Here CWC represents the sum of ice and liquid water content,

except for the CloudSat estimate shown in Fig. 4F, which is based on ice water content (IWC) alone (see also Zhao et al., 2017). Among the reanalyses, MERRA-2 (Fig. 4D) produces the largest CWCs in this region, with a pronounced peak at 300 hPa. Although MERRA-2 produces smaller cloud fractions in the tropical upper troposphere than its predecessor MERRA (Fig. 3F), it produces substantially larger CWCs (Fig. 4F). The assumed effective radius for ice particles was reduced between MERRA and MERRA-2, along with several other changes that collectively increased the average residence time of ice clouds in the

model (Molod et al., 2012, 2015). The large CWCs in MERRA-2 have significant impacts on radiative transfer, as discussed in Sect. 4 below. CFSR/CFSv2 (Fig. 4E) produces a similarly pronounced vertical maximum in CWC, but shifted slightly higher in altitude and with a peak magnitude (15.4 mg kg$^{-1}$ at 250 hPa) roughly half that produced by MERRA-2 (30.1 mg kg$^{-1}$ at 300 hPa) averaged over the inner tropics (10°S–10°N). JRA-55 (Fig. 4C) shows a qualitatively similar distribution to those of MERRA-2 and CFSR/CFSv2, but with much smaller magnitudes (maximum value: 2.4 mg kg$^{-1}$ at 250 hPa). This difference

is again consistent with JRA-55 underestimating cloud cover in the tropical upper troposphere. The zonal-mean distribution





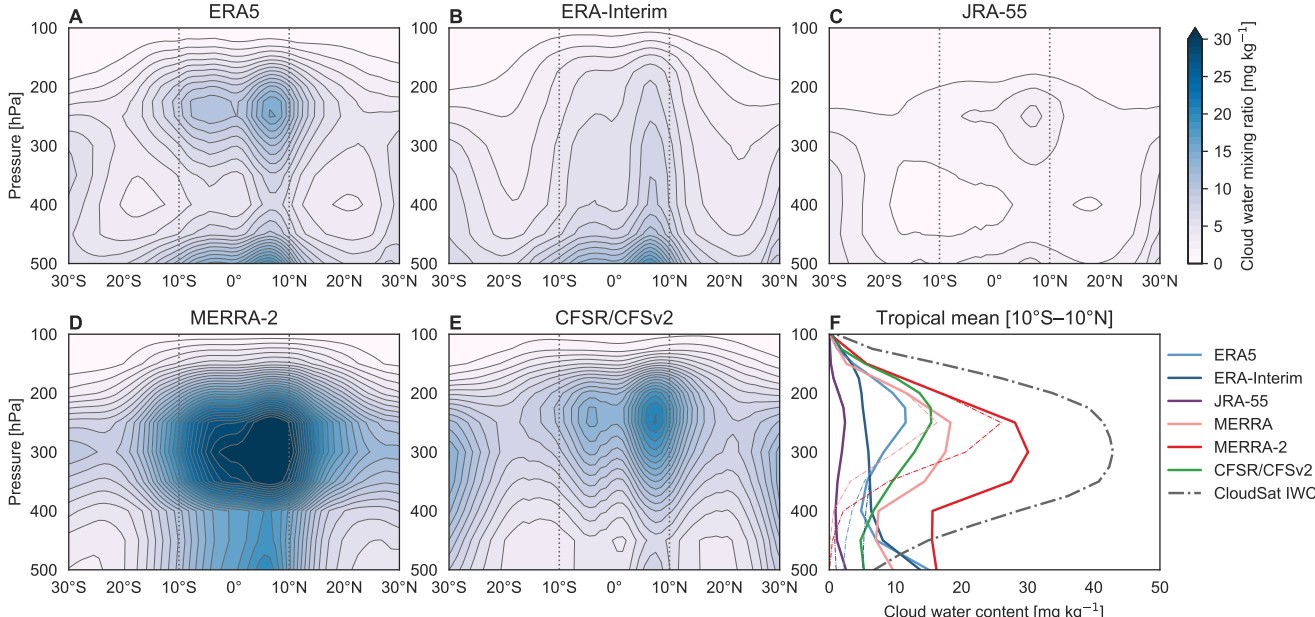

**Figure 4.** Time-mean zonal-mean distributions of total cloud water content based on the (A) ERA5, (B) ERA-Interim, (C) JRA-55, (D) MERRA-2, and (E) CFSR/CFSv2 reanalyses (1980–2014). Inner tropical mean (10°S–10°N) profiles based on these five reanalyses are shown in panel (F), along with profiles from MERRA and an observationally-based estimate of ice water content during August 2006–July 2007 from CloudSat. Vertical lines in panels (A) through (E) mark the bounds of the averaging domain. Dashed lines in (F) indicate ice-only water contents from the reanalyses that provide this information (all but CFSR/CFSv2).

of CWC in ERA-Interim (Fig. 4B) is remarkably different from that in the other reanalyses, including ERA5 (Fig. 4A), with no distinct maximum in the tropical upper troposphere. Instead, ERA-Interim shows a monotonic decrease in CWC with increasing altitude above 500 hPa. Although it is difficult to pinpoint the reason for the difference in vertical profiles of CWC between ERA-Interim and ERA5, changes to the treatment of entrainment and detrainment in the convective scheme (Ap-

5 pendix A2) likely play a key role. These changes, together with improvements in prognostic microphysics, alter the structure of the convective mass flux and improve the coupling between convection and the tropical environment (Bechtold et al., 2008, 2014).

The tropical-mean profile of IWC based on CloudSat radar-only retrievals between August 2006 and July 2007 is shown for context in Fig. 4F. The diurnal sampling of CloudSat along its initial orbit in the A-Train constellation (equator crossing

10 times around 01:30 and 13:30 local solar time) should be taken into account when comparing the profile observed by CloudSat to those produced by the reanalyses, as this orbit misses the late afternoon peak of continental convective activity in the tropics (e.g. Yang and Slingo, 2001). It is also important to note that the CloudSat estimate represents total IWC, including both precipitating and cloud ice. We may therefore expect the profile maximum to be both larger in magnitude and lower in altitude than one based on cloud ice alone (Li et al., 2012, 2016). This expectation is supported by Fig. 4F, as the peak value





of IWC based on CloudSat is larger and lower in altitude relative to the reanalysis profiles (42.9 mg kg$^{-1}$ at 300 hPa). Despite this difference, the structure of the CloudSat profile is qualitatively more consistent with the pronounced anvil layers in ERA5, MERRA-2, and CFSR/CFSv2 than with JRA-55 or ERA-Interim.

The cumulus congestus peak in the middle troposphere that does not appear in reanalysis estimates of cloud fraction (but does appear in observations) is evident in the reanalysis estimates of CWC but not in the CloudSat estimate. The latter may be attributable to the exclusion of liquid water from the CloudSat estimate, although previous analyses of CloudSat CWCs did not show a clear maximum here even when the liquid phase was included (see, e.g. Su et al., 2011). By contrast, MERRA-2, ERA-Interim, ERA5, and JRA-55 all indicate large liquid water fractions in clouds at these altitudes. In ERA-Interim, 12.5% of cloud water at 400 hPa averaged over the inner tropics is liquid, rising to 63.3% at 500 hPa. These ratios are larger in ERA5 (28.6% and 86.0%, respectively) and MERRA-2 (86.4% and 99.8%), and smaller in JRA-55 (3.3% and 60.4%). CFSR does not provide separate outputs for liquid and ice water contents. The prevalence of liquid water content at these altitudes in MERRA and MERRA-2 relative to CloudSat is a known feature of the GEOS-5 data assimilation system (Su et al., 2011).

## 4 Radiative impacts

### 4.1 Top-of-atmosphere radiation budget

Figure 5 shows spatial distributions of OLR and the LWCRE based on the MRM, along with observationally-based estimates of these quantities based on CERES EBAF and differences between six individual reanalysis products and the MRM. The MRM suggests a time-mean tropical-mean OLR of 265.8 W m$^{-2}$ over 1980–2014, much larger than the CERES EBAF estimate of 260.4 W m$^{-2}$ over 2001–2014. Accordingly, the time-mean tropical-mean LWCRE based on the MRM was 21.1 W m$^{-2}$ over 1980–2014, much less than the 30.2 W m$^{-2}$ value indicated by CERES EBAF. Isolines of LWCRE closely match those of OLR, confirming that the radiative effects of clouds dominate spatial variability in OLR within these datasets. Although differences between the MRM and the observational estimates shown in Fig. 5 may reflect differences in averaging period, observationally-based estimates with longer durations are either consistent with CERES EBAF (e.g. tropical-mean values of 259.4 W m$^{-2}$ for OLR and 27.7 W m$^{-2}$ for LWCRE based on NASA GEWEX SRB during 1984–2007) or suggest even smaller values for the tropical-mean OLR (e.g. 250.7 W m$^{-2}$ based on NOAA Interpolated OLR during 1980–2014). Rather than direct observations (with clear-sky fluxes taken only from cloud-free columns), the CERES and SRB fluxes discussed in this section are adjusted fluxes that use both direct observations and RTM calculations as constraints on the final product (Sect. 2.2).

The expected impacts of high clouds on OLR in the reanalyses are evident when panels C–H of Fig. 5 are compared to the corresponding panels of Fig. 1. For example, both ERA-Interim and CFSR/CFSv2 produce tropical mean values of OLR and LWCRE that are close to the corresponding MRM values, just as both ERA-Interim and CFSR/CFSv2 produce tropical mean high cloud fractions that are close to the MRM value. However, taking the deep convective portions of the Indo–Pacific domain as an example, ERA-Interim tends to underestimate OLR and overestimate LWCRE relative to the MRM, consistent with larger high cloud fractions in this region, while CFSR/CFSv2 tends to overestimate OLR and underestimate LWCRE, consistent with smaller high cloud fractions. ERA5 produces a slightly smaller tropical-mean OLR and slightly larger LWCRE



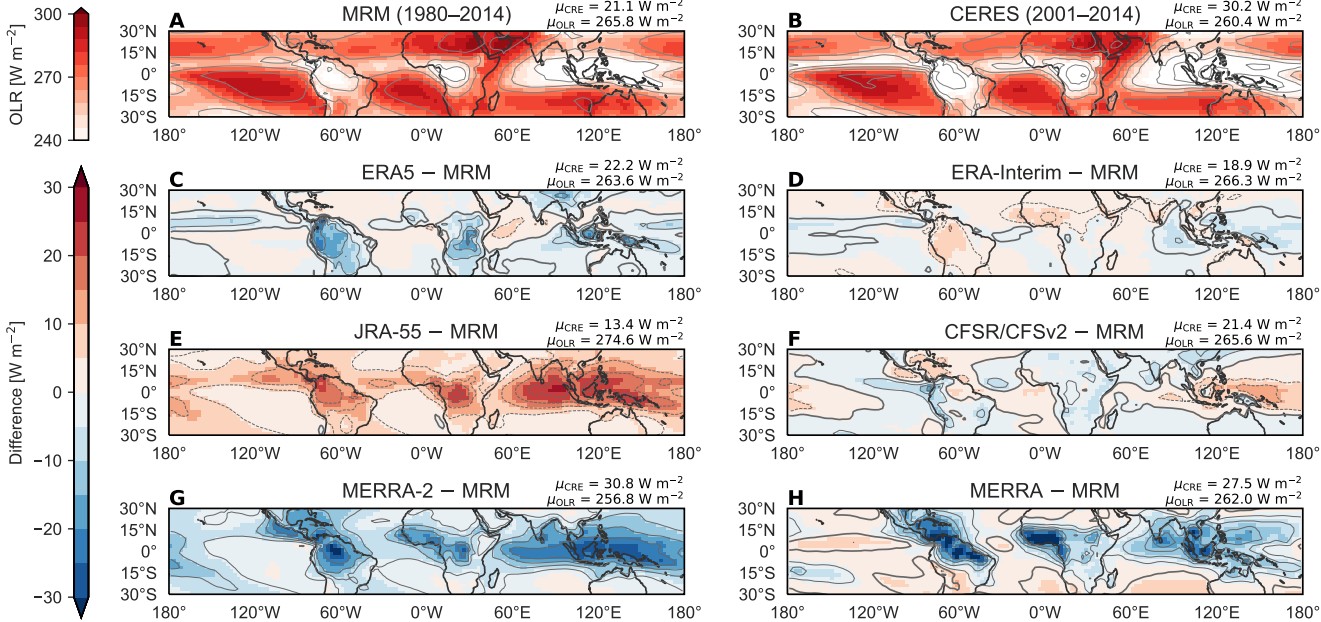

**Figure 5.** Climatological mean spatial distributions of outgoing longwave radiation (OLR; shading) and longwave cloud radiative effect (LWCRE; contours ranging from 10–60 W m$^{-2}$ at intervals of 10 W m$^{-2}$) for (A) the multi-reanalysis mean (MRM) over 1980–2014, calculated by averaging the distributions for ERA-Interim, JRA-55, MERRA-2, and CFSR, and (B) CERES EBAF over 2001–2014. Differences relative to the MRM are shown for (C) ERA5, (D) ERA-Interim, (E) JRA-55, (F) CFSR/CFSv2, (G) MERRA-2, and (H) MERRA over 1980–2014. Contours in panels (D) through (H) cover the range within ±20 W m$^{-2}$ at intervals of 5 W m$^{-2}$. Tropical mean (30°S–30°N) values of OLR and LWCRE based on each product are shown at the upper right corner of the corresponding panel, with values for OLR listed above those for the LWCRE.

than ERA-Interim, consistent with its larger tropical-mean high cloud fraction. The changes are again most pronounced over tropical land areas with strong convection, especially South America, Africa, and the South Asian monsoon region. JRA-55, which has the smallest high cloud fractions in the tropics among the reanalyses, likewise produces the largest tropical mean OLR and the smallest tropical mean LWCRE. Conversely, MERRA-2, with the largest high cloud fractions among the

5   reanalyses, produces the smallest tropical mean OLR and the largest tropical mean LWCRE. There are some notable exceptions to these relationships, such as differences between ERA-Interim and CFSR over Africa. ERA-Interim overestimates high cloud cover relative to the MRM over equatorial Africa but underestimates high cloud cover relative to the MRM over the Sahel. CFSR/CFSv2 produces a similar qualitative pattern, with slightly smaller differences relative to the MRM. However, whereas ERA-Interim tends to overestimate OLR and underestimate the LWCRE relative to the MRM over most of Africa, especially

10   over the Sahel, CFSR/CFSv2 tends to underestimate OLR and overestimate the LWCRE relative to the MRM over the same region. Such differences may reflect systematic differences in the depth of convection (and thus cloud top temperature) or the water paths associated with convective anvil clouds. Although we do not directly evaluate differences in cloud top height here





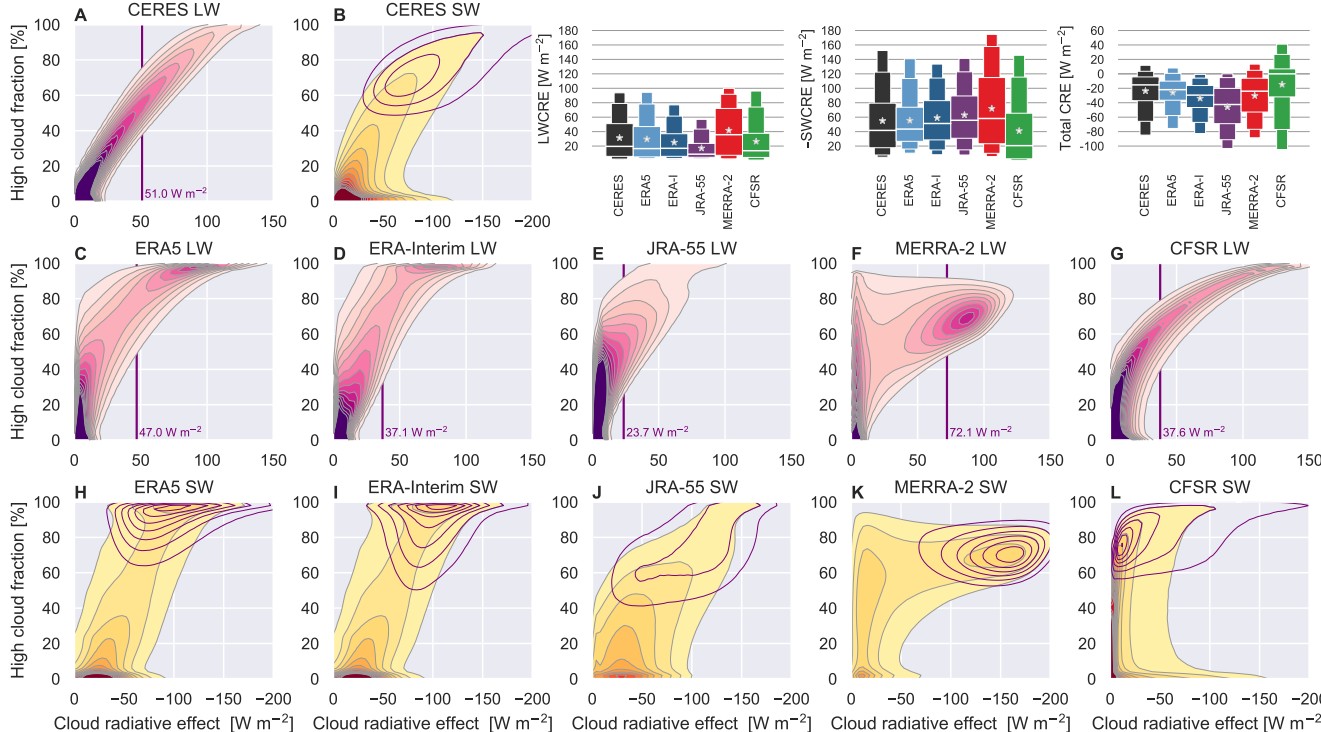

**Figure 6.** Joint distributions of daily-mean high cloud fraction against (A) LWCRE and (B) SWCRE based on CERES SYN1Deg using gridded data from 2001–2010. Corresponding joint distributions are shown for (C, H) ERA5, (D, I) ERA-Interim, (E, J) JRA-55, (F, K) MERRA-2, and (G, L) CFSR. The 75[th] percentile of the LWCRE is marked in the upper row. Sub-distributions of high cloud fraction against SWCRE associated with the values of LWCRE that exceeded the corresponding 75[th] percentile threshold are shown as purple contours in panel (B) and panels (H–L). Distributions of the LWCRE, SWCRE, and total CRE are shown in the upper right, with SWCRE multiplied by –1 for convenience of presentation. The thickest boxes mark the interquartile ranges, with the medians marked as horizontal lines and the means marked as stars. The narrower extended boxes indicate the 5[th], 10[th], 90[th], and 95[th] percentiles.

(owing in part to the lack of vertically-resolved cloud fraction profiles in CFSR/CFSv2), we note that CFSR/CFSv2 produces a more pronounced peak in cloud water content extending to relatively higher altitudes than ERA-Interim in the tropical mean (Fig. 4F).

Relationships between tropical high cloud fraction and the top-of-atmosphere radiation balance are examined in more detail
5 in Fig. 6, which shows joint distributions of high cloud fraction against the LW and SW cloud radiative effects. The joint distributions shown in Fig. 6 are analogous to scatter plots containing millions of points, where the shading indicates the density of the points and outlier values that occur infrequently are omitted. The data used to construct these distributions are daily-mean gridded values within 10°S–10°N during the period 2001–2010. This approach provides additional context on both geographical and temporal covariability of high cloud cover and radiative fluxes at the top-of-atmosphere. For this and other
10 analyses that do not span the CFSR/CFSv2 transition (1 January 2011), we omit any reference to CFSv2 and refer to this





reanalysis only as CFSR. Data have been interpolated (if necessary) to $1° \times 1°$ spatial grids. As with the LWCRE, values of the SWCRE are computed as clear-sky minus all-sky radiative fluxes. Values are positive-upward, so that the SWCRE in the tropics is typically negative (clouds enhance albedo). The abscissa is reversed for plots of the SWCRE so that larger absolute magnitudes of both LWCRE and SWCRE are located toward the right. Distributions of daily-mean gridded values of LWCRE,

SWCRE, and total CRE (LWCRE + SWCRE) are included at the upper right of Fig. 6 for context.

The joint distribution of high cloud fraction against LWCRE based on CERES SYN1Deg indicates a tight, nearly linear relationship between these two variables, in which a large value of high cloud fraction corresponds to a large LWCRE. The $75^{\text{th}}$ percentile value of LWCRE based on CERES SYN1Deg is $51.0\,\text{W}\,\text{m}^{-2}$, which corresponds to a high cloud fraction of roughly 57%. Among the reanalyses, the joint distribution of high cloud fraction against LWCRE based on CFSR is most

similar to that based on CERES SYN1Deg. However, this distribution shows a stronger curvature, so that the $75^{\text{th}}$ percentile of LWCRE corresponds to a smaller value of LWCRE ($37.6\,\text{W}\,\text{m}^{-2}$) despite a similar high cloud fraction (58%). JRA-55 has the smallest $75^{\text{th}}$ percentile value of LWCRE ($23.7\,\text{W}\,\text{m}^{-2}$). This value of LWCRE corresponds to a high cloud fraction of around 56% in JRA-55, whereas it corresponds to a high cloud fraction of only 27% in CERES SYN1Deg, implying that the relatively small mean high cloud fraction in JRA-55 is not the only reason behind relatively low values of LWCRE in this

reanalysis. Joint distributions based on ERA5, ERA-Interim, and MERRA-2 are qualitatively more distinct, with secondary modes at large values of LWCRE. In ERA-Interim, there is a clear distinction in both variables between the primary mode (associated with small high cloud fractions and small values of LWCRE and the secondary mode (associated with large high cloud fractions and large values of LWCRE). High cloud fractions associated with the latter mode are almost exclusively greater than 90%. The $75^{\text{th}}$ percentile value ($37.1\,\text{W}\,\text{m}^{-2}$) falls between the two modes and corresponds to a high cloud

fraction of 65%. The distribution based on ERA5 is similar to that based on ERA-Interim, but with a greater fraction of the data (and greater variability) in the large-LWCRE mode. The $75^{\text{th}}$ percentile value is thus substantially larger in ERA5 ($47.0\,\text{W}\,\text{m}^{-2}$) than in ERA-Interim, as is the mean cloud fraction associated with this value (75%). Bimodality in MERRA-2 takes a different form. The first mode corresponds to small values of LWCRE. Although the peak of this distribution is at small values of high cloud fraction, this small-LWCRE mode still exhibits relatively large occurrence frequencies at values of high

cloud fraction approaching 100%. The mean high cloud fraction associated with this mode is around 35%. The second mode peaks at relatively large values of LWCRE ($\sim 88\,\text{W}\,\text{m}^{-2}$) and high cloud fraction ($\sim 70\%$). The $75^{\text{th}}$ percentile of LWCRE ($72.1\,\text{W}\,\text{m}^{-2}$) is contained within the second mode, meaning that the large-LWCRE mode contains more than 25% of the inner tropical data points in MERRA-2. A LWCRE of $72.1\,\text{W}\,\text{m}^{-2}$ corresponds to a high cloud fraction of approximately 68% in MERRA-2, slightly less than that associated with the same value of LWCRE in CERES SYN1Deg (73%).

The unique bimodality of the high cloud–LWCRE distribution in MERRA-2 is a consequence of the separation of cloud condensate in the prognostic cloud scheme into 'large-scale' and 'anvil' cloud types. Of these two types, anvil clouds are assigned higher number densities that translate into greater values of optical thickness when the radiation calculations are performed (Bacmeister et al., 2006). Another potentially relevant difference is that the model used to produce MERRA-2 does not prognostically consider reductions of cloud fraction due to autoconversion (Rienecker et al., 2008), which may result in

relatively large values of cloud fraction persisting even as CWC declines. This differs from ERA-Interim and ERA5, for which





cloud fraction is among the prognostic variables treated by the cloud scheme (Tiedtke, 1993), and from CFSR, for which cloud fractions are explicitly tied to CWC (Xu and Randall, 1996). The treatment of prognostic cloud fraction used in MERRA-2 is similar to that used in JRA-55, in which a top-hat-type distribution is used to determine the portion of the grid box for which the total water content exceeds a threshold value (Smith, 1990; Molod, 2012). However, despite this similarity, the two reanalysis

systems produce very different behaviors, presumably owing to different characteristic lifetimes of cloud condensate within the atmosphere and the separate treatment of anvil condensate in MERRA-2.

Joint distributions of high cloud fraction against SWCRE are consistent with the expectation that the SWCRE is less tightly linked to high cloud fraction than the LWCRE in the tropics. However, large high cloud fractions are nonetheless associated with both large LWCREs and large SWCREs in most cases, as indicated by joint distributions conditional on the top quartile

of LWCRE. CERES SYN1Deg and four of the five reanalyses show extensive overlap between large values of LWCRE and large values of SWCRE. CFSR is a notable exception, with large values of LWCRE often corresponding to small values of SWCRE. As a consequence, the distribution of total CRE based on CFSR is broader than those based on CERES or the other reanalyses, with the middle 90% spanning more than $140\,\mathrm{W\,m^{-2}}$, from less than $-100\,\mathrm{W\,m^{-2}}$ to approximately $+40\,\mathrm{W\,m^{-2}}$. The weaker SWCRE associated with large high cloud fractions in CERES results in the total CRE being more positive on

average, with the median value close to zero. Although the LWCRE is weaker in JRA-55 than in any other data set evaluated here, the tropical-mean SWCRE is larger in JRA-55 than in any data set except MERRA-2. The total CRE is thus substantially more negative in JRA-55 than in any of the other data sets. Fewer than 5% of gridded values of total CRE in the tropics are positive in JRA-55. This latter statement is also true of ERA-Interim; however, greater compensation between the LWCRE and SWCRE in ERA-Interim leads to a narrower distribution and thus a smaller negative bias in the tropical-mean total CRE

relative to CERES. MERRA-2 tends to overestimate both the LWCRE and the SWCRE, especially for anvil clouds. However, compensation between these two biases produces a distribution of total CRE that is comparable to but slightly broader than that based on CERES SYN1Deg. Among the five reanalyses, ERA5 shows the greatest agreement with CERES SYN1Deg across all three flavors of CRE. The LWCRE in ERA5 is slightly weaker on average than that based on CERES, while the SWCRE is very similar on average but with a narrower distribution. The total CRE is thus slightly more negative in ERA5 than indicated

by CERES SYN1Deg, with a narrower distribution but very close agreement in the average value.

## 4.2   Radiative heating in the tropical UTLS

In addition to altering top-of-atmosphere radiative fluxes, differences in tropical high clouds may influence radiative heating rates locally within the UTLS. Among the reanalyses considered in this study, neither JRA-55 nor CFSR provide vertically-resolved estimates of radiative heating under clear-sky conditions. To skirt this limitation, we construct composite mean profiles

of radiative heating rates conditional on the four quartiles of LWCRE in an adaptation of the approach employed by Zhang et al. (2017), who composited heating rates on OLR rather than LWCRE. Figure 7 shows these composite profiles for the period 2001–2010, separated into total, LW, and SW radiative heating. Here, Q1 represents daily gridded heating rates for which the LWCRE (at TOA; Fig. 6A and C–G) is in the lowest 25% of all daily gridded values. Q2 and Q3 represent the lower middle and upper middle quartiles, respectively, while Q4 represents heating rates for which the associated LWCRE exceeds the 75[th]

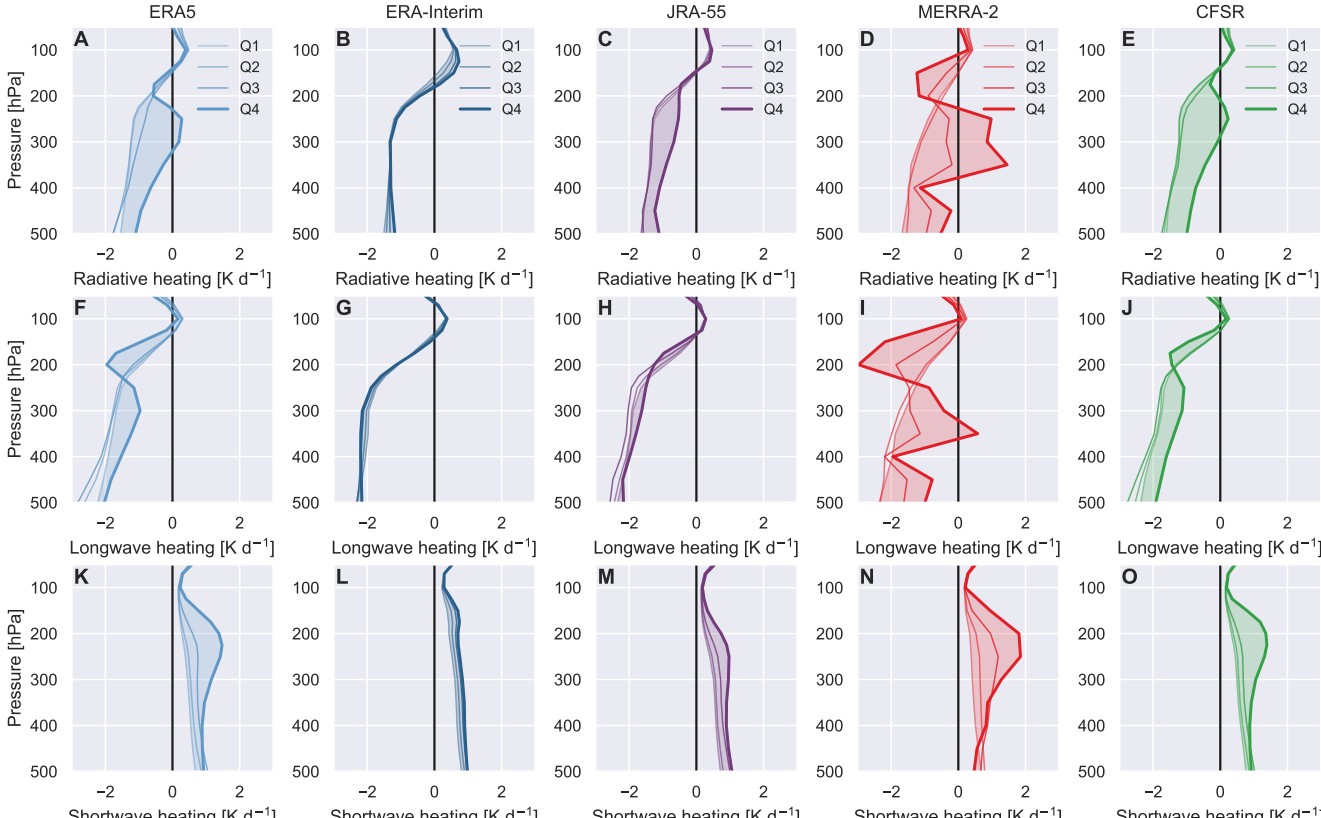

**Figure 7.** Composite mean profiles of daily-mean radiative heating rates as a function of pressure for the first through fourth quartiles (Q1–Q4) of longwave cloud radiative effect in the inner tropics (10°S–10°N; see also Fig. 6) based on (A) ERA5, (B) ERA-Interim, (C) JRA-55, (D) MERRA-2, and (E) CFSR during 2001–2010. Here Q1 refers to the bottom quartile (weak longwave CRE) and Q4 to the top quartile (strong longwave CRE). Total radiative heating rates (upper row; A–E) are separated into (F–J) longwave and (K–O) shortwave components in the lower two rows.

percentile value marked in Fig. 6. The impact of clouds on heating rates is then estimated as the difference between the Q4 and Q1 profiles.

Among these five reanalysis systems, cloud effects on radiative heating rates are generally smallest in ERA-Interim and largest in MERRA-2. The results for these two reanalyses are essentially consistent with those reported for ERA-Interim and
5 MERRA by Wright and Fueglistaler (2013), who showed that cloud impacts on radiative heating rates in MERRA are qualitatively opposite to those in ERA-Interim through much of the upper troposphere. The response in ERA-Interim is concentrated in the 100–200 hPa layer, where radiative heating rates are enhanced by the presence of high clouds. At lower altitudes in the upper troposphere (200–400 hPa), cloud-induced increases in SW heating are effectively balanced by cloud-induced increases in LW cooling. By contrast, ERA5, JRA-55, and CFSR show only weak cloud impacts on total radiative heating at pressures





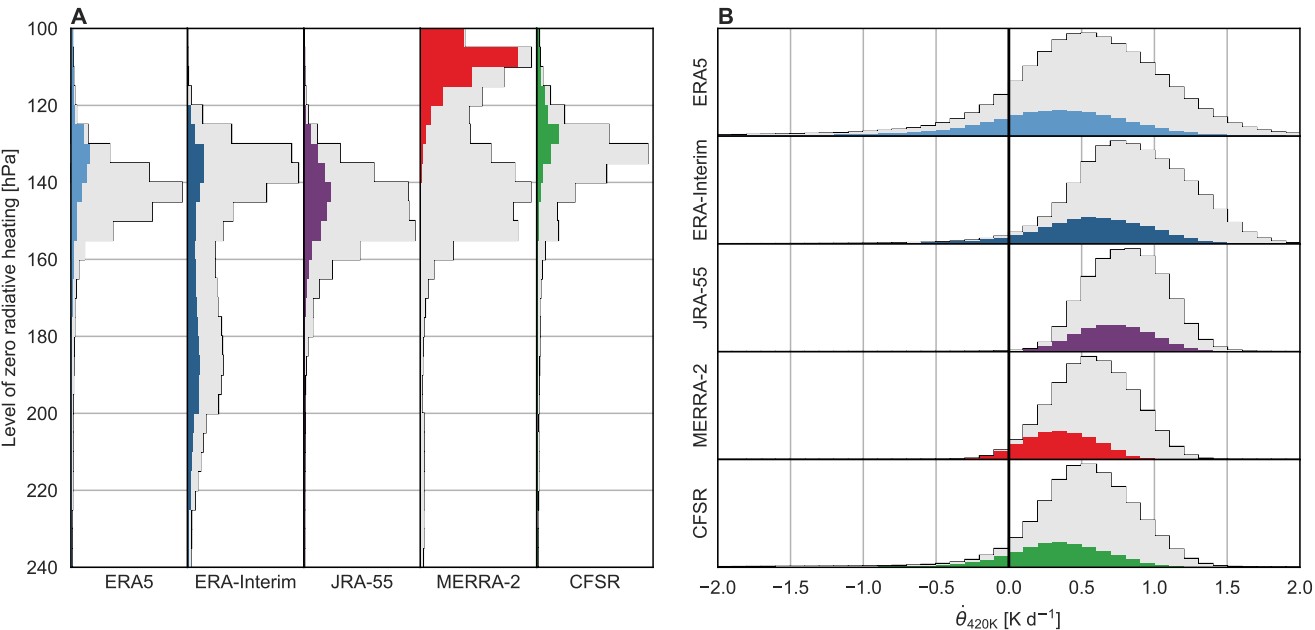

**Figure 8.** Histograms of (A) the vertical location of the level of zero radiative heating (LZRH) and (B) the vertical velocity in isentropic coordinates ($\dot{\theta}$ on the 420 K isentropic surface. Data are based on daily mean products from (left-to-right and top-to-bottom) ERA5, ERA-Interim, JRA-55, MERRA-2, and CFSR during 2001–2010. Colored histograms show distributions for the top quartile of longwave cloud radiative effect in each reanalysis (see Fig. 6).

less than 175 hPa. In all three cases, the insensitivity in total radiative heating rates traces back to a near-complete compensation between enhanced LW cooling and enhanced SW heating at these altitudes. Substantial cloud-related perturbations in the LW and SW components extend upward to around 100 hPa in ERA5 and CFSR, but only to around 150 hPa in JRA-55. MERRA-2 produces the largest cloud impacts on radiative heating rates. Indeed, direct comparison of cloud radiative effects between

5    MERRA and MERRA-2 (not shown) indicates that cloud radiative impacts in MERRA are further amplified in MERRA-2, consistent with the increase in CWC in the tropical upper troposphere between MERRA and MERRA-2 (Fig. 4F). The effects of high clouds in MERRA-2 are to reduce radiative heating rates in the 100–200 hPa layer (largely due to enhanced LW cooling, partially offset by enhanced SW heating) and increase radiative heating rates at pressures greater than 200 hPa. The latter is the result of enhanced SW heating near the top of the anvil layer (200–250 hPa) and enhanced LW heating near the base of

10   the anvil layer (300–350 hPa), taking the MERRA-2 profile of tropical-mean cloud water content (Fig. 4F) as a guide. Cloud effects in ERA5 are qualitatively similar to those in CFSR and MERRA-2, with an intermediate magnitude between CFSR and MERRA-2 and a smoother profile than MERRA-2. This is consistent with the relatively pronounced convective anvil in the tropical-mean CWC profile based on ERA5 (Fig. 4F), which is more consistent with the profiles produced by CFSR and MERRA-2 than with that produced by ERA-Interim.





Differences in the radiative impacts of tropical high clouds can lead in turn to differences in transport through the tropical tropopause layer and lower stratosphere (Fueglistaler and Fu, 2006; Yang et al., 2010). Oft-used metrics in this regard include the level of zero net radiative heating (LZRH) and the rate of diabatic ascent at the base of the 'tropical pipe', which defines the upward branch of the Brewer–Dobson circulation (e.g. Fueglistaler et al., 2009; Dessler et al., 2014). Here, the LZRH marks

the boundary between negative radiative heating rates (corresponding to net descent) in the tropical troposphere and positive radiative heating rates (corresponding to net ascent) in the tropopause layer and lower stratosphere. We identify this level by using linear interpolation of daily-mean gridded radiative heating rates in $\ln(p)$ to determine pressure at the zero crossing. We further require that radiative heating rates remain positive above the identified LZRH to at least the 70 hPa isobaric level. To represent the radiative signature of ascent at the base of the tropical pipe, we use the vertical velocity in potential temperature

coordinates ($\dot{\theta}_{\mathrm{rad}}$) as diagnosed at the 420 K isentropic level, near the top of the tropical tropopause layer. Figure 8 shows distributions of LZRH pressure and $\dot{\theta}_{\mathrm{rad}}$ at 420 K based on the ERA-Interim, JRA-55, MERRA-2, and CFSR reanalyses during 2001–2010. Distributions conditional on the top quartile of LWCRE (Q4) are shaded in color.

The largest differences in the distributions of LZRH altitudes are between ERA-Interim and MERRA-2 (Fig. 8A). Neglecting the influence of clouds, the primary mode of the ERA-Interim distribution ($p \sim 140$ hPa) is shifted to slightly higher altitudes

than that in MERRA-2 ($p \sim 150$ hPa). The altitudes of these primary modes match the vertical locations of the clear-sky LZRH in each system well (not shown). The more striking distinction between ERA-Interim and MERRA-2 concerns the impacts of clouds on the LZRH altitude. Whereas clouds tend to lower the LZRH in ERA-Interim (to around 170 hPa on average), clouds raise the LZRH significantly in MERRA-2 (to around 110 hPa), with important implications for the efficiency of mass and constituent transport from the deep convective detrainment level (200∼300 hPa) into the tropical lower stratosphere. In

MERRA-2, the cloudy and clear-sky modes of the distribution are almost completely distinct, suggesting that transport regimes in the tropical upper troposphere are approximately binary in this model. By contrast, the breadth of the LZRH distribution based on ERA-Interim (and especially the breadth of the distribution associated with the largest values of LWCRE) indicates that ERA-Interim produces a broad spectrum of cloudy states. This diagnostic thus helps to clarify the environmental conditions that give rise to the two very different tropical mean cloud water content profiles in Fig. 4F, with the pronounced anvil layer

in MERRA-2 in sharp contrast to the gradual decrease of cloud water content with height in ERA-Interim. Distributions of the LZRH location based on ERA5, JRA-55, and CFSR are more consistent with each other. Each distribution has one major mode, although the altitude of the LZRH tends to be highest in CFSR (median: 134 hPa), followed by ERA5 (144 hPa) and JRA-55 (148 hPa). All three of ERA5, JRA-55, and CFSR indicate slight upward shifts toward lower pressures (by around 5 hPa) in the median LZRH location associated with the largest values of LWCRE, but these shifts are much less pronounced

than that in MERRA-2.

Distributions of $\dot{\theta}_{\mathrm{rad}}$ at 420 K (Fig. 8B) are more consistent among the reanalyses. Differences in the mean value are consistent with those reported elsewhere (Schoeberl et al., 2012; Abalos et al., 2015; Tao et al., 2019), with ERA-Interim (average: 0.82 K day$^{-1}$) and JRA-55 (0.80 K day$^{-1}$) indicating stronger lower-stratospheric ascent than MERRA-2 (0.56 K day$^{-1}$) or CFSR (0.49 K day$^{-1}$). The mean value in ERA5 (0.49 K day$^{-1}$) is consistent with those in MERRA-2 and CFSR. Our fo-

cus here is mainly on the cloud effects and the role that they play in the overall differences. All five reanalyses indicate that



lower stratospheric radiative heating rates are reduced in atmospheric columns with large values of LWCRE. As with many of the diagnostics examined in this study, this effect is least pronounced in JRA-55, with differences between Q1 (smallest LWCREs) and Q4 (largest LWCREs) of $0.13\pm0.03\,\mathrm{K\,day^{-1}}$. This relatively small cloud influence likely contributes to the relatively narrow distribution of $\dot{\theta}_{\mathrm{rad}}$ in JRA-55. By contrast, the much broader distributions of $\dot{\theta}_{\mathrm{rad}}$ in ERA5 and ERA-

Interim are accompanied by large cloud effects, with differences of $0.49\pm0.12\,\mathrm{K\,day^{-1}}$ between Q1 and Q4 in ERA5 and $0.45\pm0.06\,\mathrm{K\,day^{-1}}$ in ERA-Interim. These large cloud effects are consistent with sharper spatial gradients in high cloud fraction (Fig. 1c) and LWCRE (Fig. 5c) between the canonical deep convective regions of the tropics and surrounding areas in the two ECMWF reanalyses relative to JRA-55. The cloud influence on $\dot{\theta}_{\mathrm{rad}}$ in MERRA-2 is comparable to that in ERA-Interim, with a difference of $0.39\pm0.03\,\mathrm{K\,day^{-1}}$ between Q1 and Q4. However, the distribution based on MERRA-2 is compressed

toward the mean relative to that based on ERA-Interim, with fewer extreme values and shorter tails. Only 8% of $\dot{\theta}_{\mathrm{rad}}$ values in MERRA-2 fall outside the interval $[0,1]\,\mathrm{K\,day^{-1}}$, as opposed to 36% of values in ERA-Interim (33% in ERA5). This pairing of large cloud effect and narrow distribution implies a strict stratification of lower stratospheric heating rates with respect to LWCRE, with values of $\dot{\theta}_{\mathrm{rad}}$ based on MERRA-2 approaching those based on JRA-55 as the effects of clouds are reduced. The mean difference in $\dot{\theta}_{\mathrm{rad}}$ between these two reanalyses is $\sim0.4\,\mathrm{K\,day^{-1}}$ in Q4 (where the mean LWCRE in MERRA-2 is more

than double that in JRA-55), but only $\sim0.1\,\mathrm{K\,day^{-1}}$ in Q1 (where mean values of LWCRE are $3.1\,\mathrm{W\,m^{-2}}$ in both systems). Our results thus support the suggestion by Tao et al. (2019) that differences in climatological high cloud cover in the tropics can explain much but not all of the difference in lower stratospheric air mass ascent between these reanalyses. The cloud effect on lower stratospheric heating rates is $0.31\pm0.09\,\mathrm{K\,day^{-1}}$ in CFSR. The uncertainty in this estimate is relatively large because distributions of $\dot{\theta}_{\mathrm{rad}}$ based on CFSR have larger variance within each quartile of LWCRE (primarily due to higher occurrence

frequencies of negative values in all four quartiles). Approximately 4% of $\dot{\theta}_{\mathrm{rad}}$ values associated with the relatively cloud-free Q1 and Q2 groupings in CFSR are negative (implying diabatic descent), an order of magnitude larger than the fraction in ERA-Interim and several orders of magnitude larger than the fractions in JRA-55 and MERRA-2. However, it is clear that the largest variance in $\dot{\theta}_{\mathrm{rad}}$ is that produced by ERA5. This variance decreases with decreasing LWCRE; however, the fraction of negative $\dot{\theta}_{\mathrm{rad}}$ values in Q1 and Q2 (9%) is more than double that in CFSR. The broader distribution of diabatic heating

rates in this reanalysis may be related to improved consistency between diabatic and kinematic vertical motion in the lower stratosphere in ERA5 relative to ERA-Interim (Hoffmann et al., 2019).

## 5 Possible origins

The prognostic cloud parameterizations used by the reanalysis models consider two sources of high clouds: detrainment from deep convection and in situ formation due to large-scale saturation (see Appendix A). Sinks include autoconversion of cloud

water to precipitation and evaporation or sublimation of cloud water into subsaturated air. In considering the origins of differences in high clouds among the reanalyses, we therefore focus on factors that might influence the sources and sinks of high cloud, as well as metrics that may reflect coupled relationships between high clouds and their environment. With respect to the convective source, we examine relationships between high cloud fraction and SST, thermodynamic stability in the lower tro-





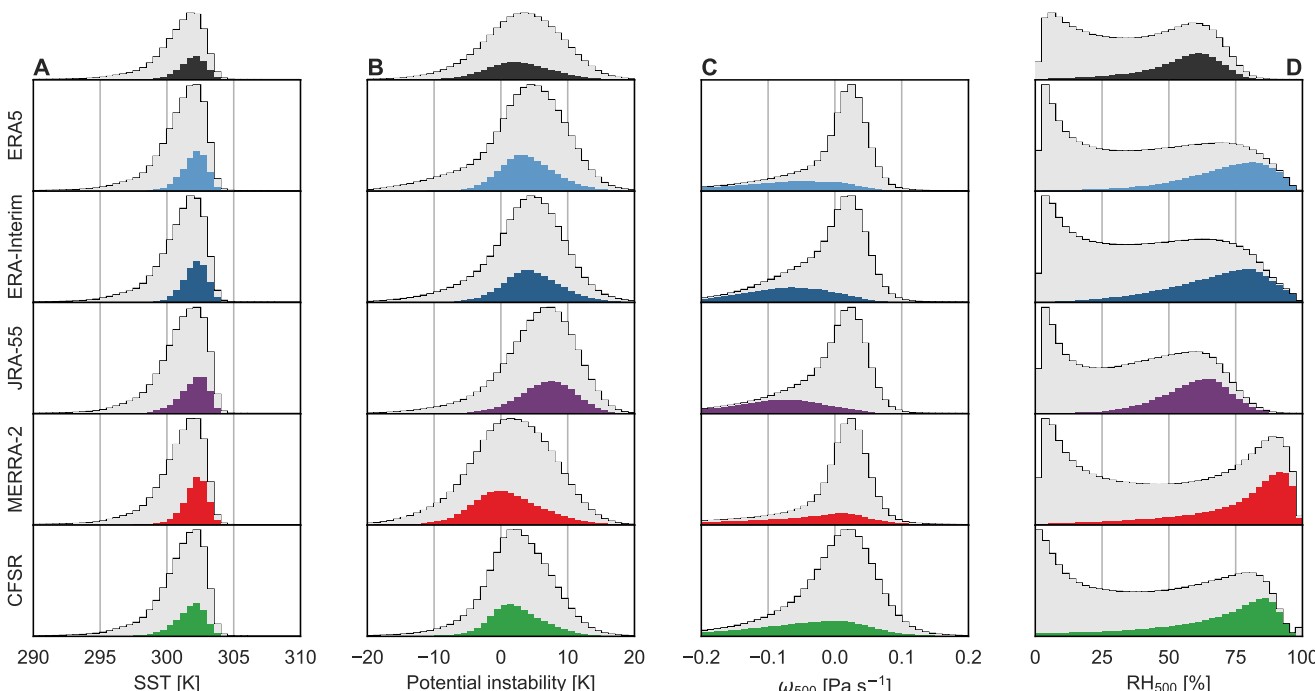

**Figure 9.** Histograms of (A) sea surface temperature (SST), (B) potential instability ($\theta_{e,850} - \theta_{e,500}$), (C) grid-scale vertical velocity in the middle troposphere ($\omega_{500}$), and (D) relative humidity in the middle troposphere (RH$_{500}$). Data are based on daily mean products from (top-to-bottom) ERA5, ERA-Interim, JRA-55, MERRA-2, and CFSR during 2001–2010 within the inner tropics (10°S–10°N). Observational estimates are shown along the top axis where available, and include data from CERES SYN1Deg (LWCRE), OISST version 2 (SST), and AIRS TqJoint (potential instability and RH$_{500}$). Distributions that include AIRS data are based on data from 2003–2010 rather than 2001–2010. Colored histograms show distributions for the top quartile of longwave cloud radiative effect in each reanalysis (see Fig. 6). Mean values for each distribution are listed in Table 2.

posphere, grid-scale vertical velocity and RH in the middle troposphere (500 hPa), and the mean vertical profile of moist static energy (MSE). To assess the in situ source and evaporation sink, we examine relationships among CWC, RH, and radiative heating rates near the base of the tropical tropopause layer (150 hPa) and just below the cold point tropopause (100 hPa). All relationships are assessed at the daily timescale within the inner tropics (10°S–10°N). The use of daily means collapses im-

5 portant diurnal variations in tropical convective activity that may be poorly represented by the reanalyses (e.g. Bechtold et al., 2014). This diurnal variability may imprint on relationships among daily-mean variables but we do not explore this possibility here.





**Table 2.** Mean values of the distributions shown in Fig. 9 for all data point in the inner tropics ('All'; 10°S–10°N) and for the top quartile of LWCRE in the same region ('Q4'). The row labeled 'Observed' summarizes the results when LWCRE is taken from CERES SYN1Deg, SST from OISST v2, and potential instability and mid-tropospheric RH from AIRS TqJoint.

| Product | Sea surface temperature | | Potential instability | | Mid-tropospheric $\omega$ | | Mid-tropospheric RH | |
|---|---|---|---|---|---|---|---|---|
| | All | Q4 | All | Q4 | All | Q4 | All | Q4 |
| ERA5 | 301.0 K | 302.0 K | 3.4 K | 3.9 K | $-0.02\,\mathrm{Pa\,s^{-1}}$ | $-0.11\,\mathrm{Pa\,s^{-1}}$ | 42% | 70% |
| ERA-Interim | 300.9 K | 302.1 K | 3.6 K | 4.7 K | $-0.02\,\mathrm{Pa\,s^{-1}}$ | $-0.09\,\mathrm{Pa\,s^{-1}}$ | 42% | 67% |
| JRA-55 | 301.0 K | 301.9 K | 5.4 K | 6.7 K | $-0.02\,\mathrm{Pa\,s^{-1}}$ | $-0.11\,\mathrm{Pa\,s^{-1}}$ | 37% | 58% |
| MERRA-2 | 300.9 K | 302.1 K | 1.4 K | 1.0 K | $-0.02\,\mathrm{Pa\,s^{-1}}$ | $-0.09\,\mathrm{Pa\,s^{-1}}$ | 49% | 76% |
| CFSR | 300.9 K | 301.6 K | 2.6 K | 2.4 K | $-0.01\,\mathrm{Pa\,s^{-1}}$ | $-0.08\,\mathrm{Pa\,s^{-1}}$ | 44% | 67% |
| Observed | 300.9 K | 301.9 K | 3.1 K | 2.8 K | -- | -- | 37% | 54% |

## 5.1 Convection and its environment

Tropical deep convection tends to cluster over the warmest SSTs. This behavior is captured in all five of the reanalysis systems, with the largest high cloud fractions systematically associated with the largest SSTs. Tropical-mean SSTs prescribed during the 2001–2010 analysis period are very similar among the reanalyses (Table 2). CFSR exhibits the weakest relationship between
SST and high cloud cover, with Q4 of the LWCRE associated with a mean SST of 301.6 K (relative to 301.9–302.1 K in the other four reanalyses). The observation-based benchmark, with CERES SYN1Deg used to estimate daily-mean LWCRE and OISST v2 (Reynolds et al., 2007) for daily-mean SST, likewise assigns a mean value of 301.9 K to Q4, 1 K warmer than the tropical mean. CFSR is the only coupled atmosphere–ocean data assimilation system among these five reanalyses, making it the only one with the potential for two-way interactions between high clouds and SST (although analyzed SST is still pegged quite
tightly to observations; Saha et al., 2010). Note that OISST v2 was used as an atmospheric lower boundary condition during portions of this intercomparison period by ERA-Interim (July–December 2001) and MERRA-2 (through March 2006), and as the primary input to SST analyses by CFSR throughout (Fujiwara et al., 2017, their Table 4). The observational benchmark distribution of SST is therefore not strictly independent.

Figure 9B summarizes distributions of lower tropospheric potential instability (defined as the difference in $\theta_e$ between
850 hPa and 500 hPa; Eq. 2) for all tropical points and for points associated with Q4 of LWCRE. Values of potential instability in the tropics tend to be positive in all five reanalyses. However, this tendency toward positive values is weaker for MERRA-2 and CFSR than for ERA5, ERA-Interim, or JRA-55, indicating systematic differences in the moist thermodynamic state of the tropical atmosphere among these reanalyses (see also Table 2). Moreover, while ERA5, ERA-Interim, and JRA-55 indicate larger potential instabilities associated with Q4 of LWCRE than in the tropical mean, MERRA-2 and CFSR indicate
the opposite. The latter is in better agreement with AIRS. For ERA-Interim, these differences may be linked to the convective closure (Appendix A2). The convection scheme in ERA-Interim specifies an adjustment timescale that, in practice, often exceeds the model time step (especially at coarser resolutions; Bechtold et al., 2008, their Fig. 1). As such, potential instability




in convective locations (Q4) may be shifted toward larger positive values in ERA-Interim (Fig. 9B). The new closure (Bechtold et al., 2014) and finer model resolution used in ERA5 reduce the difference between the Q4 and tropical mean values of potential instability by about half (Table 2). The discrepancy between JRA-55 and the other reanalyses has a different origin. Figure 10 shows vertical profiles of MSE averaged over the upper and lower quartiles of daily gridded LWCRE within the

inner tropics (10°S–10°N). A 'kink' is evident in the vertical profile for JRA-55 between 900 hPa and 850 hPa but not in any of the other profiles. The convective scheme in JRA-55 restricts cloud base to the model level at ∼900 hPa (JMA, 2013). Thermodynamic instabilities that develop at higher levels (such as the 850 hPa level used to compute potential instability) are thus more difficult for the convection scheme to eliminate. Decomposing differences in moist static energy into contributions from differences in temperature, specific humidity, and geopotential (not shown), we find that differences in atmospheric

moisture content are the most influential at both the 850 and 500 hPa levels. At 850 hPa, latent energy ($L_v q$) based on CFSR and MERRA-2 is 1–2 kJ kg−1 less than that based on JRA-55, ERA5, or ERA-Interim. Meanwhile, at 500 hPa, latent energy based on MERRA-2 is nearly 3 kJ kg$^{-1}$ larger than that in JRA-55, and more than 1 kJ kg$^{-1}$ larger than that in ERA5, ERA-Interim, or CFSR. Biases in the dry enthalpy component ($c_p T$) are on the order of 0.5 kJ kg$^{-1}$ at both levels. For JRA-55 and MERRA-2, temperature biases tend to compensate for humidity biases at 850 hPa but exacerbate the effects of humidity biases

at 500 hPa. The relationship between potential instability and LWCRE produced by CFSR is most similar to that based on observations in terms of mean values, with AIRS estimates 0.4–0.5 K larger than those based on CFSR for both the tropics as a whole and the top quartile of LWCRE (Table 2). However, the distribution of potential instability based on AIRS is broader than that based on CFSR, and in that sense is more reminiscent of the distributions based on MERRA-2 or ERA5 (Fig. 9B).

Negative biases in both moisture content and temperature at 500 hPa in JRA-55 relative to the other reanalyses may stem in

part from the inability to trigger convection from instabilities at altitudes above 900 hPa; however, the shift of the liquid–ice transition toward warmer temperatures also plays a role in dehydrating the middle troposphere. The much larger moisture content at 500 hPa in MERRA-2 can also be linked to details of the cloud parameterization. We discuss these possibilities further after briefly highlighting two other features of the MSE profiles shown in Fig. 10. First, lower tropospheric values of MSE associated with Q4 are evidently larger in all of the reanalyses than those based on AIRS observations. This may indicate

that the reanalyses are systematically too moist or too warm in the lower troposphere, but may also reflect systematic errors or sampling biases (e.g. cloud clearing) in the AIRS observations. Second, MERRA-2 shows substantially larger values of MSE in the upper troposphere in convective regions relative to the other reanalyses. This bias results from both greater humidity (perhaps due to greater detrainment of cloud water and subsequent condensate evaporation; Fig. 4) and systematic warm biases (possibly linked to more intense cloud radiative heating at anvil level; Fig. 7). At 300 hPa, the excess Q4 MSE in MERRA-2

relative to ERA-Interim is on average 62% attributable to differences in the dry enthalpy component ($c_p T$) and 35% attributable to differences in the latent energy component ($L_v q$). The residual discrepancy (3%) arises from differences in geopotential. This difference in upper tropospheric MSE is systematic throughout the tropics (e.g. the Q1 profile in Fig. 10), but with smaller magnitudes and temperature biases a proportionally greater contributor outside of the deep convective regions. Greater upper tropospheric MSE in MERRA-2 implies stronger gross moist stability, and specifically a stabilization of the upper troposphere

that may suppress the average depth of convection. The resulting lower, more extensive anvil deck in MERRA-2 appears to be





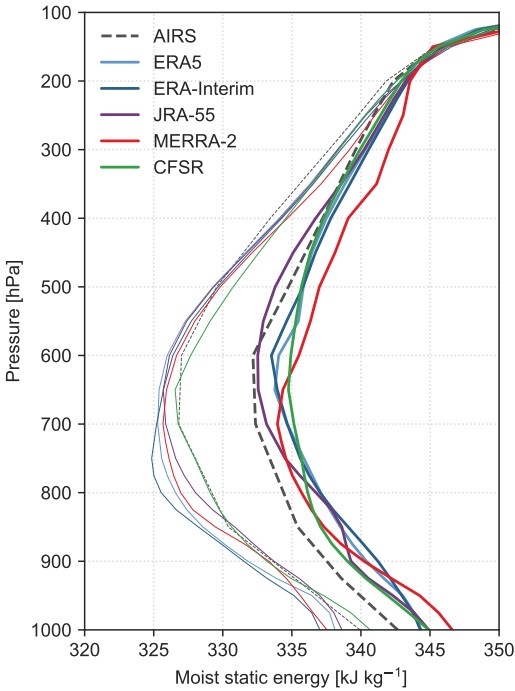

**Figure 10.** Composite vertical profiles of moist static energy (MSE) for ERA5 (cyan), ERA-Interim (blue), JRA-55 (purple), MERRA-2 (red), and CFSR (green) averaged for the upper (Q4; thick lines) and lower (Q1; thin lines) quartiles of daily-mean LWCRE during 2001–2010. Profiles calculated from AIRS observations (September 2002–December 2010; grey dashed lines) are shown for context. AIRS profiles are conditioned on quartiles of daily-mean LWCRE from CERES SYN1Deg.

a key factor in the relatively strong cloud-top radiative cooling in this reanalysis (Fig. 7) and the inability of convective heating to compensate for this cooling. As noted previously for MERRA, this combination yields a physically implausible layer of time-mean zonal-mean diabatic descent centered near 200 hPa that extends across the entire tropics (Wright and Fueglistaler, 2013).

5     Figure 9C and D show distributions of grid-scale vertical velocity ($\omega$) and RH in the middle troposphere (500 hPa). Distributions of vertical velocity for the whole tropics are qualitatively similar across the five reanalyses, with peaks at small positive values (subsidence) and long tails toward large negative values (ascent). Larger values of LWCRE in ERA-Interim and JRA-55 are associated almost exclusively with grid-scale ascent in the middle troposphere. This relationship is less pronounced in MERRA-2 and CFSR (i.e. large LWCREs are more frequently associated with mid-tropospheric descent), although

10  the strongest mid-tropospheric ascent rates are associated with Q4 in all five reanalyses. These differences may be understood in terms of differences in the convective triggers (Appendix A2), which explicitly consider large-scale convergence in ERA-Interim and JRA-55 but not in MERRA-2 or CFSR. Explicit dependence of the convective trigger on large-scale vertical velocity was eliminated from the ECMWF atmospheric model between the version used for ERA-Interim and that used for



ERA5 (Bechtold et al., 2008). No observational benchmark is available for evaluating these distributions. Distributions of mid-tropospheric RH (defined here with respect to liquid water) are bimodal in all five reanalyses, with peaks at both very small values ($< 10\%$) and relatively large values ($> 50\%$). The largest values of LWCRE tend to be associated with large values of mid-tropospheric relative humidity, although this relationship is tighter for MERRA-2 and CFSR than for ERA5, ERA-Interim, or JRA-55. The largest differences among the distributions are at the upper end of the range, and can be explained to some extent by differences in the treatment of the liquid–ice transition (Appendix A; Fig. A1). As JRA-55 has the strictest transition from liquid to ice, mid-tropospheric RH with respect to liquid water is generally less than 75%. ERA-Interim and ERA5 prescribe more gradual transitions from liquid to ice, and thus produce larger relative humidities with respect to liquid water. Another potentially important parameter is the critical RH at which large-scale cloud formation (or evaporation of cloud water) is assumed to occur. This value is more than 90% in MERRA-2 at 500 hPa as opposed to around 80% in ERA5 and ERA-Interim, leading MERRA-2 to produce a larger frequency of very high relative humidities at this level. Tighter distributions of mid-tropospheric RH associated with the largest values of LWCRE (Fig. 9D) also suggest that deep convection may be more sensitive to mid-tropospheric entrainment of dry air in MERRA-2 and CFSR than in ERA-Interim or JRA-55. For MERRA-2 this is consistent with the application of a Tokioka-type entrainment condition (Bacmeister and Stephens, 2011): whether a plume is triggered depends on whether the required entrainment rate exceeds a randomly-selected minimum. For small values of RH, entrainment is efficient in diluting the updraft, so that only small entrainment rates will permit plumes that reach the upper troposphere. If the required value of entrainment is smaller than the randomly-selected Tokioka parameter, plumes that reach the upper troposphere will not be allowed. The application of the Tokioka condition thus tightens the preference for deeper convection to occur in more humid environments. Entrainment rates are also relatively large in CFSR, which uses a base entrainment rate equal to the maximum entrainment rate in JRA-55 and approximately an order of magnitude larger than the base entrainment rate in ERA-Interim. Among the reanalyses, the distribution of mid-tropospheric RH in JRA-55 is most consistent with that based on AIRS. However, just as for lower tropospheric MSE, caveats concerning sampling and cloud-clearance biases apply when interpreting AIRS-based estimates.

## 5.2 Clouds in the TTL

Tropical high clouds in the reanalyses may also originate via the parameterized effects of grid- or subgrid-scale saturation. In the TTL, such in situ cloud formation is often associated with adiabatic cooling linked to wave activity or radiatively-driven ascent (Massie et al., 2002; Schoeberl et al., 2019). Figure 11 summarizes relationships among CWC, radiative heating rates, and RH at isobaric levels near the base of the TTL (150 hPa) and near the tropopause (100 hPa). This figure links average CWCs with paired ranges of heating rate and RH, along with differences in occurrence frequency between the largest (Q4) and smallest (Q1) values of LWCRE.

At 150 hPa, the distributions based on ERA5 (Fig. 11F) and MERRA-2 (Fig. 11I) show striking similarities, with 'wings' of large CWCs at large positive and negative radiative heating rates bracketing a central axis in which radiative heating is weak and CWC depends mainly on RH. The largest values of LWCRE are mainly associated with strong radiative cooling at 150 hPa (the left wing), while strong radiative heating (the right wing) is more often associated with Q2 or Q3 rather than



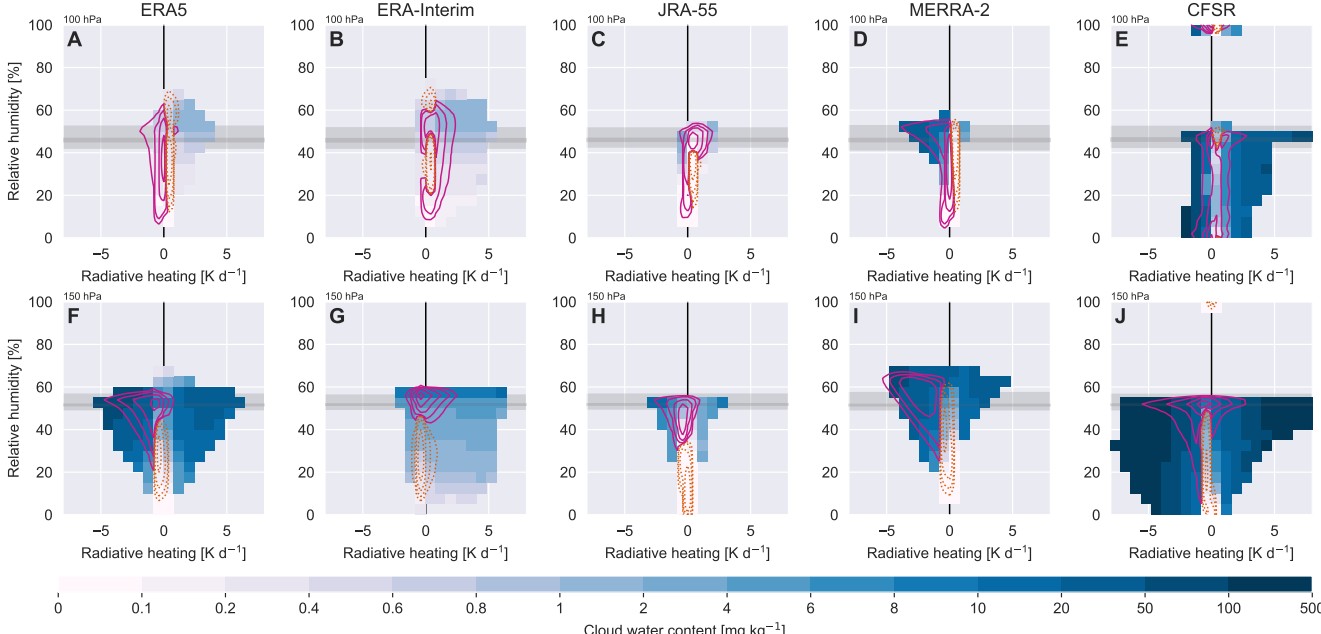

**Figure 11.** Composite distributions of daily-mean high cloud fraction as a function of radiative heating rate and grid-scale relative humidity (RH) in (A, F) ERA5, (B, G) ERA-Interim, (C, H) JRA-55, (D, I) MERRA-2, and (E, J) CFSR on the 100 hPa (upper row; A–E) and 150 hPa (lower row; F–J) isobaric surfaces. RH is calculated with respect to liquid water. Grey shaded regions in each panel mark ranges of ice saturation ratios ($e_i^*/e_\ell^*$) at these levels, with light shading marking the minimum and maximum and dark shading marking the interquartile range. Solid pink contours mark paired values of radiative heating and RH that are more commonly associated with cloudy conditions (Q4 of the daily-mean LWCRE) than with clear-sky conditions (Q1 of the LWCRE); dashed orange contours mark the opposite (values more commonly associated with Q1 than Q4). Composite mean CWCs are masked for bins containing fewer than 200 samples.

Q4 (not shown). This difference is consistent with composite mean profiles of CWC, as negative heating rates at this level are associated with lower and more extensive anvils than positive heating rates (Fig. 12; see also Fig. 7). The two wings may thus represent the different radiative impacts of growing versus mature convective systems, and specifically the more extensive anvil clouds associated with the latter (Machado et al., 1998). The distribution based on JRA-55 is similar to those based on ERA5 and MERRA-2 but with a much smaller range of radiative heating rates, consistent with less extensive anvil clouds and smaller water paths in this reanalysis (Fig. 4C). The distribution based on CFSR also shows similar features, but with additional variance in radiative heating that results from the occasional occurrence of very large daily mean CWCs (up to 1408 mg kg$^{-1}$) at this level. Approximately 1% of daily mean CWCs at 150 hPa in CFSR exceed 100 mg kg$^{-1}$, far more than in the other reanalyses (maximum: 0.1% in ERA5). The distribution based on ERA-Interim is more distinctive, with CWC (and LWCRE) more tightly linked to RH and an asymmetry toward positive heating rates. ERA-Interim produces very few instances of large negative heating rates, as cloud effects tend to increase radiative heating at 150 hPa in this reanalysis (Fig. 7).





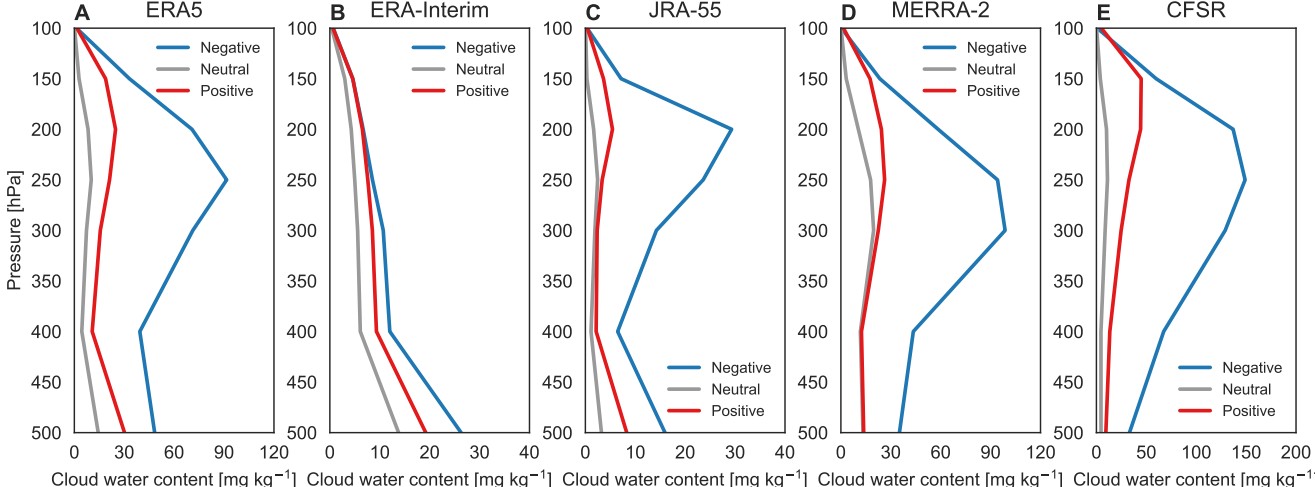

**Figure 12.** Composite mean profiles of cloud water content (CWC) from (A) ERA5, (B) ERA-Interim, (C) JRA-55, (D) MERRA-2, and (E) CFSR associated with different ranges of radiative heating rates at 150 hPa: negative rates less than $-1\,\mathrm{K\,d^{-1}}$ (blue), neutral rates within $\pm 1\,\mathrm{K\,d^{-1}}$ (grey), and positive rates greater than $+1\,\mathrm{K\,d^{-1}}$ (red). Note different $x$ axis ranges for CWC.

At 100 hPa, ERA5 and ERA-Interim show similar distributions of composite mean CWC, with larger values concentrated in regions of positive radiative heating and supersaturation with respect to ice. However, these two systems show opposite relationships with LWCRE. Whereas large values of LWCRE are more commonly associated with positive heating rates at this level in ERA-Interim, the largest values of LWCRE are mainly associated with negative heating rates at this level in

ERA5. ERA5 and ERA-Interim are the only models considered here that explicitly consider supersaturation with respect to ice. Figure 11 indicates that preferred locations of in situ cloud formation near the tropopause may differ between these two reanalyses because convective clouds influence radiative heating differently within the TTL (see also Fig. 7A,B). The distribution in JRA-55 (Fig. 11C) also shows a peak in composite-mean CWC associated with high RH and positive heating rates, along with a secondary peak at high RH and weak negative heating rates. The distribution based on MERRA-2 likewise

has a bimodal structure (Fig. 11D), but in this reanalysis the most prominent mode evokes the wing-like structure at 150 hPa. This mode features relatively large CWCs, negative radiative heating rates, and large LWCREs, and is thus consistent with anvil clouds that penetrate to near the tropical tropopause. The second mode features positive radiative heating rates and saturation with respect to ice, and is thus consistent with expectations for high clouds with small water contents that form near the tropopause (Fusina et al., 2007). Indeed, with the exception of ERA-Interim, the largest CWCs at 100 hPa are associated with

very large water paths through the UTLS and large negative radiative heating at 100 hPa. The smallest CWCs at 100 hPa are associated with near-zero radiative heating rates. Mean CWCs associated with positive radiative heating ($> +0.5\,\mathrm{K\,d^{-1}}$) are significantly larger, but still at least 1–2 orders of magnitude smaller than those associated with large negative radiative heating rates. Taking large negative radiative heating ($< -0.5\,\mathrm{K\,d^{-1}}$) and large CWCs ($> 10\,\mathrm{mg\,kg^{-1}}$) at 100 hPa as a crude indicator



of overshooting convection reaching the tropopause, these events occur around 0.2% of the time in MERRA-2 and 0.1% of the time in CFSR and ERA5. These criteria are never met in JRA-55 or ERA-Interim. Conversely, taking large positive radiative heating ($> +0.5 \, \mathrm{K \, d^{-1}}$) and above-average CWCs ($> 0.01 \, \mathrm{mg \, kg^{-1}}$) as indicative of thin cirrus in air rising through the TTL, this regime covers 35% of the tropics in ERA-Interim, 24% in JRA-55, and 9–11% in ERA5, MERRA-2, and CFSR.

The distribution based on CFSR at 100 hPa (Fig. 11E) indicates severe problems with humidity fields around the tropopause. Values of RH in CFSR at this level cluster around three values: zero (7% of samples), saturation with respect to ice (6% of samples), and saturation with respect to liquid water (77% of samples). Values between zero and saturation with respect to ice account for the remaining 10% of samples. Relatively large values of CWC are found throughout the range of RH at 100 hPa. Saturation with respect to liquid water in CFSR occurs occasionally at 150 hPa as well (Fig. 11J), although these instances

differ from those at 100 hPa in that they are associated mainly with small values of LWCRE and negligible CWCs, and only represent a small fraction of samples (1.5%). Humidity fields in the stratosphere are known to be unrealistically small in this reanalysis (Davis et al., 2017). Figure 11 shows that this unrealistic behavior often extends downward into the TTL.

Although MERRA-2 contains no explicit representation of supersaturation with respect to ice, RH in this reanalysis exceeds the saturation threshold with respect to ice for around 33% of gridded daily means at 150 hPa and 20% of gridded daily means

at 100 hPa. This decrease with height differs from ERA-Interim and ERA5, in which daily-mean supersaturation frequencies increase from 15–25% at 150 hPa to 30–40% at 100 hPa. The occurrence of ice supersaturation in MERRA-2 can result from the partitioning of liquid and ice and subsequent gradual relaxation of liquid condensate to ice as implemented in the model's prognostic cloud scheme (Appendix A1), but it is surprising that it remains so prevalent in the TTL. This feature may be explained by temporal truncation: the model limits the water vapor content at grid-scale saturation, the temperature is then

modified by some other process, and output is written without further adjustment to the water vapor field. All of the supersaturated points in MERRA-2 have non-zero CWCs, and CWC tends to increase with increasing supersaturation ($r = 0.24$). Liquid water is present in trace amounts for almost all supersaturated points at 150 hPa (93%) and a large fraction of supersaturated points at 100 hPa (31%). This persistence of positive liquid water contents at very low temperatures will be addressed in a forthcoming version of the GEOS-5 model.

**6   Temporal variability**

Figure 13 shows deseasonalized monthly anomalies for high cloud cover, OLR, and LWCRE in the inner tropics (10°S–10°N) based on the five reanalyses and CERES-based data sets (CERES SYN1Deg for high cloud cover; CERES EBAF for OLR and LWCRE). Anomalies are calculated relative to the mean annual cycle over all full years in the CERES overlap period (2001–2014). Most of the reanalyses produce temporary increases in high-cloud fraction and LWCRE (and corresponding

decreases in OLR) around the major El Niño events of 1982–83 and 1997–98, although the timing, amplitude, and duration of these excursions varies. However, the most pronounced variations appear to be systemic rather than physical. Most notably, the tropical-mean high cloud fraction in CFSR jumped suddenly by more than 0.1 between the end of 2009, when CFSR was initially planned to end, and the beginning of 2010. Tropical-mean high cloud fraction then increased again at the beginning





of 2011 with the transition to CFSv2, to a value very close to that in MERRA-2 (not shown). The bridge year 2010 is not well documented, but has been shown to feature discontinuities in other variables as well (e.g. stratospheric water vapor; Davis et al., 2017). Sudden jumps in the CFSR time series are not limited solely to the CFSR/CFSv2 transition, with transient reductions in tropical-mean high cloud fraction after every production stream transition in the initial 1979–2009 run (1 January

1987, 1990, and 1995; 1 April 1999 and 2005 Saha et al., 2010; Fujiwara et al., 2017). However, whereas these latter stream-related discontinuities are also evident in OLR and LWCRE (as is the transition at the beginning of 2010), neither OLR nor LWCRE shows large changes following the transition to CFSv2 in January 2011. Despite suggestions that CFSv2 can serve as an extension of CFSR, researchers should be cautious in adopting this approach for any study that spans the 2010 bridge year or the 2011 transition to CFSv2.

In addition to the production stream transitions in CFSR/CFSv2, several of the anomaly time series show long-term drifts. To assess whether these long-term changes are consistent across data sets, we evaluate trends over the 1980–2014 and 2001–2014 periods (Fig. 13D–E). Trends and confidence intervals are calculated using the robust Theil–Sen estimator (Sen, 1968). Note that even where trends are statistically significant, the signs and magnitudes of these trends are subject to uncertainties associated with data processing and changes in the observing system over time. These caveats apply not only to reanalyses, but

also to observationally-based analyses (like the ISCCP and CERES cloud products; e.g. Dai et al., 2006) and derived products that depend on these analyses (like the SRB and CERES radiation products; e.g. Trenberth et al., 2009). The trend values shown here are for intercomparison purposes, without assessment of their realism or reliability.

Over the full record, JRA-55 shows the most obvious and temporally consistent increase in high cloud fraction, along with corresponding changes in OLR (towards smaller values) and LWCRE (towards larger values). These changes bring JRA-

55 closer to the other reanalyses by the later part of the record, although absolute biases in tropical-mean OLR relative to ERA-Interim and CFSR/CFSv2 remain on the order of $10\,\mathrm{W\,m^{-2}}$ (as opposed to $\sim 15\,\mathrm{W\,m^{-2}}$ in the early 1980s). Most of the other reanalyses show qualitatively similar trends in high cloud fraction (increasing), OLR (decreasing), and LWCRE (increasing) over 1980–2014, although with magnitudes smaller than those based on JRA-55. Decreasing trends in clear-sky OLR are qualitatively robust, except for the 1980–2014 trend in MERRA-2. Decreasing trends in clear-sky OLR suggest that

increases in atmospheric greenhouse gas absorption outpace increases in the effective emission temperature in the tropics over this period, changes that may be explained by the so-called 'hiatus' in surface warming during the early 2000s (Song et al., 2016). Prescribed greenhouse gas concentrations in the reanalyses increased throughout this period (Fujiwara et al., 2017, their Fig. 4), as observed surface temperatures cooled or stayed roughly constant through much of the tropics for more than a decade (e.g. Kosaka and Xie, 2013). Most of the reanalyses also suggest decreasing trends in all-sky OLR over this period,

although the signs (over 1980–2014) and magnitudes (over 2001–2014) of these trends are not supported by observations. Associated declines in upwelling LW radiation at the tropopause may help to explain long-term decreases in tropical cold point temperatures based on JRA-55, MERRA-2, and CFSR (Tegtmeier et al., 2019). With the exception of CFSR/CFSv2 (affected by discontinuities around the CFSR–CFSv2 transition as discussed above) and ERA5 (for which trends are small), relatively large decreasing trends in all-sky OLR among reanalyses reflect relatively large increases in LWCRE, which are

linked in turn to increases in high cloud fraction. The increases in LWCRE and high cloud cover implied by reanalyses are





**Figure 13.** Time series of deseasonalized anomalies in (A) monthly mean high cloud cover (HCC), (B) monthly mean OLR, and (C) monthly mean LWCRE averaged over the inner tropics (10°S–10°N) for 1980–2014 based on ERA5 (cyan), ERA-Interim (blue), JRA-55 (purple), MERRA-2 (red), and CFSR/CFSv2 (green). Observational analyses from CERES SYN1Deg (A; March 2000–December 2014) and CERES EBAF (B and C; March 2000–December 2014) are shown for context. Anomalies are calculated relative to the mean annual cycle during 2001–2014. Thick lines show time series after applying a 12-month uniformly-weighted rolling mean. Trends are listed for annual-mean anomalies during the (D) 1980–2014 and (E) 2001–2014 periods in percentage points per decade for HCC and units of W m$^{-2}$ per decade for OLR, clear-sky OLR, and LWCRE. Stars indicate statistical significance at the 90% (*), 95% (**), 99% (***), and 99.5% (****) confidence levels. Light grey shading indicates that the 90% confidence interval of the Theil–Sen slope contains zero. Blue colors mark negative trends and red colors positive trends, with darker shades signifying larger trend magnitudes (0 to 1, 1 to 2, 2 to 3, and greater than 3).



generally not supported by the observationally-based time series; however, discrepancies in high cloud cover trends among CERES (positive), ISCCP (negative), and MODIS (no significant trend) reveal endemic uncertainty regarding whether and in what direction this variable changed. Long-term trends over 1980–2014 are small in both ERA5 and ERA-Interim. This picture changes considerably in the later part of the record, over which trends in ERA5 remain small but trends in ERA-Interim are

among the largest across all data sets. For ERA-Interim, weak long-term trends thus reflect relatively large excursions in these variables in the early 1980s acting to offset relatively large changes in the same direction after the turn of the century. These early-1980s excursions have also been reported for TTL temperatures based on ERA-Interim (Tegtmeier et al., 2019).

Figure 14 summarizes paired correlations and normalized standard deviations among the five reanalyses and available observation-based benchmarks. Monthly anomalies and evaluation metrics are calculated for the longest overlapping period

common to both datasets (CERES: 2001–2014; ISCCP/SRB: 1984–2007; CFSR: 1980–2009; all other reanalyses: 1980–2014). For CFSR we truncate the time series after 2009 to avoid the 2010 bridge year and the CFSR–CFSv2 transition. Extending the time series through 2014 reduces the correlations and increases the normalized standard deviations. Two sets of summary results are provided, one for the tropics as a whole (30°S–30°N; Fig. 14A–C) and one for the inner tropics (10°S–10°N; Fig. 14D–F). Correlations and standard deviations are calculated first for data on a common 2.5°×2.5° latitude–longitude grid

and then averaged for the corresponding region.

Among the reanalyses, monthly anomalies based on ERA5 consistently show the highest correlations against observational benchmarks for all metrics (high cloud cover, OLR, and LWCRE), regions (full tropics and inner tropics), and analysis periods (1984–2007 for ISCCP/SRB and 2001-2014 for CERES). By contrast, MERRA-2 shows relatively poor correlations for high cloud cover, especially in the inner tropics. Correlations for high cloud cover relative to CERES are larger than those relative to

ISCCP for all five reanalyses. Although this difference is also found for CERES versus SRB with respect to variability in OLR and LWCRE, the difference is less pronounced in these cases. Paired correlations for OLR and LWCRE almost all exceed 0.7, with only correlations against CFSR (complicated by the issues around production stream transitions) falling below 0.6.

Most of the reanalyses (except for JRA-55 in both regions and MERRA-2 in the inner tropics) show stronger variability in high cloud fraction than indicated by CERES SYN1Deg or ISCCP D2. However, this may reflect shortcomings in the

observational analyses, such as sampling biases or limited sensitivity to thinner high clouds. The smaller standard deviation in high cloud cover in JRA-55 is likewise consistent with JRA-55 tending to underestimate high cloud cover relative to the other reanalyses (Fig. 1E). Conversely, the results for MERRA-2, where variability is stronger than observed when averaged over the full tropics and weaker than observed when averaged over the inner tropics, may be associated with MERRA-2 producing relatively large cloud fractions outside of the core convective regions (Fig. 1G). Results for variations in OLR and LWCRE

are fairly robust, with JRA-55 consistently underestimating variability and MERRA-2 consistently overestimating variability relative to all other data sets. ERA-Interim also tends to underestimate variations in OLR and LWCRE relative to CERES or SRB, while standard deviations based on ERA5 and CFSR are similar to observed.



**Figure 14.** Metrics measuring agreement in monthly anomalies of (A, D) high cloud fraction, (B, E) outgoing longwave radiation, and (C, F) the longwave cloud radiative effect among the reanalyses and observational data sets examined in this paper. The upper left triangle in each panel shows correlation coefficients between each pair of products, while the lower right triangle shows the standard deviation in the product marked on the horizontal axis relative to that in the product marked on the vertical axis. Both metrics are evaluated first for individual grid cells in the 2.5° common grid (see, e.g., Fig. 1) and then averaged. The upper row (A–C) shows results for the entire tropics (30°S–30°N), while the lower row (D–F) shows results for the inner tropics only (10°S–10°N). Solid grey lines separate evaluations relative to the observational benchmarks based on CERES, ISCCP, and NASA-GEWEX SRB from those based on intercomparison of reanalysis products.



## 7   Summary and outlook

We have presented and evaluated differences in tropical high clouds and their radiative impacts in five recent reanalyses: ERA5, ERA-Interim, JRA-55, MERRA-2, and CFSR. As a general rule, JRA-55 has less cloud water and smaller high cloud fractions than other reanalyses in the tropical upper troposphere. MERRA-2 represents the opposite bookend, with more cloud water and

larger high cloud fractions. Accordingly, JRA-55 significantly overestimates OLR and underestimates the top-of-atmosphere LWCRE in the tropics relative to observations and other reanalyses, while MERRA-2 produces smaller values of OLR and larger values of the LWCRE, in better agreement with observations. Tropical-mean values from ERA-Interim and CFSR are similar to each other (and to the multi-reanalysis means) despite substantially different bias distributions. Relative to these two reanalyses, ERA5 produces slightly larger cloud fractions and smaller values of OLR.

Systematic differences in CWC translate into differences in radiative heating rates within the tropical upper troposphere and lower stratosphere, with the largest CWCs (MERRA-2) corresponding to extensive disruption of the radiative heating profile and the smallest CWCs (ERA-Interim and JRA-55) corresponding to relatively weak effects. On one extreme, large CWCs in MERRA-2 result in a physically unreasonable time-mean zonal-mean layer of diabatic cooling in the tropics around 200 hPa (e.g. Tao et al., 2019, their Fig. D1). A similar layer in MERRA is known to cause problems with transport simulations

in the TTL (e.g. Schoeberl et al., 2012). On the other extreme, the vertical distribution of CWC in ERA-Interim lacks the distinctive anvil layer found in observations and other reanalyses. As a result, only ERA-Interim among these five reanalyses indicates that cloud effects shift the LZRH toward lower altitudes on average. All other reanalyses indicate upward shifts, with the largest shift in MERRA-2. It is worth noting that an upward shift runs counter to results based on applying radiative transfer models to observed cloud distributions, which indicate that cloud effects lower the LZRH (Corti et al., 2005; Fueglistaler and

Fu, 2006; Yang et al., 2010). This disagreement appears to arise from a combination of (1) the peak positive shortwave effect being located at lower altitudes in the reanalyses and (2) the negative longwave effect being much stronger in the reanalyses (Fig. 7; cf. Yang et al., 2010, their Fig. 10). The former suggests that the reanalyses may systematically underestimate the depth of convective anvil clouds, although this is not immediately evident in Figs. 3 or 4. For the latter, systematic underrepresentation of thin cirrus and/or their radiative effects within the TTL seems the most likely explanation (Corti et al., 2005; Yang

et al., 2010), especially as we represent cloud effects here in terms of the relative magnitude of the LWCRE.

Heating rates in the lower stratosphere are also impacted (Norton, 2001; Fueglistaler and Fu, 2006; Tao et al., 2019). Large CWCs and a strong LWCRE, as in MERRA-2, correspond to weaker convergence of LW radiation in the lower stratosphere and hence smaller diabatic ascent rates in the tropical lower stratosphere. Conversely, small CWCs and a weak LWCRE, as in JRA-55 or ERA-Interim, correspond to larger rates of diabatic ascent in this region. At the nominal TOA, most of the reanalyses

show substantial compensation between the LWCRE and SWCRE associated with thick high clouds, as the largest LWCREs are also associated with relatively large opposing SWCREs. Exceptions are JRA-55, in which a weak LWCRE and a strong SWCRE result in a negative bias in the total CRE, and CFSR, in which a moderate LWCRE and a weak SWCRE result in a positive bias in the total CRE. Assuming equal clear-sky fluxes, these systematic differences translate to a net loss of energy by the tropical atmosphere in JRA-55 and a net gain of energy by the tropical atmosphere in CFSR relative to the other reanalyses.





These radiative biases may in turn contribute to differences in other processes, such as horizontal energy advection, convective activity, or the effects of data assimilation. Many of the differences in high clouds and their radiative impacts can be traced back to assumptions and simplifications applied in the model convection schemes or in special treatments of detrained condensate in the prognostic cloud scheme. However, these differences also often involve feedbacks between parameterized cloud fields

and the tropical environment that are not completely mitigated by assimilation of observational data into the reanalysis system.

    The reanalyses demonstrate a range of cloud behaviors near the tropical tropopause. Further evaluation will be needed given the current lack of suitable observational constraints on these behaviors, but values in CFSR are often unrealistic. Water vapor and cloud fields from CFSR should be avoided at these levels. We have also reported evident discontinuities at production stream transitions in CFSR, indicating that this data set should be used with caution, especially in analyses that span the

2010 bridge year and/or the 2011 transition to CFSv2. Taking all factors into account (an absence of major drifts or jumps, consistently high correlations, and standard deviations quite close to those found in observationally-based analyses), ERA5 appears to provide a better representation of temporal variability in high cloud fraction, OLR, and LWCRE within the tropics than other recent reanalyses. However, it is important to note that the current version of ERA5 contains known discontinuities in some variables in the early 2000s (e.g. temperature biases around and above the tropopause; Hersbach et al., 2018). A

replacement version ERA5.1 covering the problematic period is intended to address this issue.

    We have highlighted several notable differences between ERA-Interim and ERA5 that may be of interest to users familiar with ERA-Interim. First, ERA5 produces more extensive cloud cover than ERA-Interim over continental convective regions in the tropics. This difference has previously been reported to reduce brightness temperature biases in these regions (Bechtold et al., 2014). Second, the maximum cloud fraction in the tropical upper troposphere is shifted to lower altitudes in ERA5

(∼175 hPa) relative to ERA-Interim (∼150 hPa). Comparison with observations does not clearly demonstrate which of these is more realistic. Third, a pronounced anvil maximum in CWC is present in the tropical upper troposphere in ERA5 but not in ERA-Interim. The distribution in ERA5 is more consistent with observations of ice water content from CloudSat, but still shows substantial discrepancies. Fourth, as a consequence of the increased CWC in the upper troposphere, the high bias in OLR and low bias in LWCRE in ERA-Interim are both reduced in ERA5. Distributions of the LWCRE, SWCRE, and total

CRE based on ERA5 are more consistent with those inferred from CERES data. However, in both ERA5 and ERA-Interim, the low biases in LWCRE relative to CERES EBAF are twice as large in absolute magnitude as the high biases in OLR, indicating that clear-sky OLR may be underestimated in these reanalyses even as all-sky OLR is overestimated. This may indicate issues with composition, emissivity, or other aspects of the LW radiation scheme in addition to clouds. We find the same feature in CFSR, which uses the same base model for LW radiative transfer as ERA5 and ERA-Interim (Appendix A3). Finally, cloud

effects on radiative heating rates in the tropical upper troposphere, tropopause layer, and lower stratosphere are very different between ERA5 and ERA-Interim. Results for ERA5 are more in line with those found in other reanalyses (as noted above for the LZRH), but should be further evaluated against independent data types (such as the CloudSat FLXHR-LIDAR product; L'Ecuyer et al., 2008).

    Much of the information on the origins and impacts of biases in high clouds in this paper derives from relationships between

cloud cover and other variables, including radiative exchange and the moist thermodynamic environment. Such comparisons





not only help to reveal issues in the cloud parameterizations, but also highlight where and in what ways such issues may affect reanalysis variables more tightly constrained by the data assimilation, such as temperatures, winds, and humidities. The vertical profile of moist static energy (Fig. 10) is an instructive example. MSE is calculated solely using variables targeted during data assimilation. However, our results reveal important differences in the vertical profile of MSE, especially in convective

regions. Nor are these differences attributable solely to discrepancies in the water vapor fields, as illustrated by the stabilizing effects of upper tropospheric warm biases in MERRA-2. Biases in upper tropospheric temperatures imply differences in the vertical location and spatiotemporal variability of isentropic surfaces as well. Such differences may impact the results of reanalysis-driven transport model simulations, regardless of whether those studies assume isentropic, diabatic, or kinematic representations of vertical motion. It is worth reiterating that we use the 'assimilated' (ASM) products from MERRA-2, which

derive from the IAU corrector forecast, as opposed to the 'analyzed' (ANA) outputs, which derive from the 3D-FGAT analysis directly. The latter are expected to provide a closer match with the assimilated observations. However, in the case of MSE the ANA and ASM products are still in closer agreement with each other (figure not shown) than with AIRS or other reanalyses: both MERRA-2 products show large positive biases in the upper troposphere over convective regions, with comparable biases in the temperature, moisture, and geopotential components, respectively. We have focused on the ASM products in this inter-

comparison both because these variables are self-consistent with MERRA-2 cloud and radiation products and because NASA GMAO recommends the use of ASM products over ANA products in transport model simulations.

Several observationally-based data sets are used to establish context for the cloud fields. Such observationally-based analyses are limited by perspective, especially when clouds occur into multiple overlapping layers, and the information that goes into them is neither homogeneous in space nor continuous in time. Both issues can be addressed to some extent, the former through

the use of active remote sensing techniques such as lidar and radar (Stephens and Kummerow, 2007), and the latter through systematic analyses of imagery collected by the global network of geostationary satellites (Rossow and Schiffer, 1999), but no current observational platform addresses both simultaneously. Moreover, discrepancies among observationally-based estimates arising from differences in measurement capabilities and techniques remain quite large (e.g. Pincus et al., 2012; Stubenrauch et al., 2013). Our results suggest that the range of variability among observational estimates of cloud fraction is at least com-

parable in magnitude to that among recent reanalyses, suggesting that current observations may not by themselves constrain quantitative biases in reanalysis (or other model-based) products. Observation simulators can help, not least by enabling new sets of sensitivity tests (e.g. Stengel et al., 2018), but are still limited to the cloud populations that can be effectively observed by the observational platforms being emulated.

Beyond occasional references in the context of other variables, we have neglected cloud top height in this intercomparison.

The rationale for this omission is that the reanalyses do not typically provide these metrics directly, so that they must be inferred by other means. However, systematic biases in cloud top height may have implications for the magnitude and spatial distribution of cloud radiative effects. Such biases may also influence the spatiotemporal distribution of convective source regions for air entering the stratosphere as inferred from transport model simulations. A systematic intercomparison of cloud top height metrics based on consistent methodologies may be useful for revealing further deficiencies or idiosyncrasies of



the convective parameterizations used in the reanalysis models, as well as how these features might imprint upon both more widely-used reanalysis products and model simulations that use reanalysis fields to drive transport within the atmosphere.

## Appendix A: Cloud and radiation parameterizations in the reanalyses

In this appendix, we briefly document selected aspects of the cloud, convection, and radiation parameterizations in the reanaly-
sis atmospheric models. Additional information on the models, data assimilation schemes, and other elements of the reanalysis systems has been provided by Fujiwara et al. (2017) and in Chapter 2 of the forthcoming SPARC Reanalysis Intercomparison Project report (Wright et al., 2019, review version available at https://jonathonwright.github.io/S-RIPChapter2E.pdf).

### A1  Prognostic cloud parameterizations

All reanalyses examined in this paper use prognostic parameterizations of large-scale clouds that consider two sources of high
clouds: detrainment from deep convection and in situ condensation associated with large-scale vertical motion or diabatic cooling.

The evolution of high clouds in ERA-Interim is governed by the scheme outlined by Tiedtke (1993), in which a pair of equations are used to simultaneously track cloud water mass and cloud fraction accounting for transport, convective and large-scale source terms, as well as losses due to evaporation and precipitation. The scheme does not distinguish between liquid water
and ice; rather, the ice-phase fraction is diagnosed as a quadratic function of temperature between 0°C (entirely liquid) and –23°C (entirely ice) at each time step (Fig. A1). The model also includes a parameterization to represent supersaturation with respect to ice at temperatures below –23°C. ERA5 uses an updated version of the same scheme. One of the most important changes is that both liquid and ice condensate are treated prognostically in ERA5, eliminating diagnostic partitioning between the two phases. The resulting behavior cannot be easily summarized in Fig. A1, but a comparison between the approach used
in ERA5 and that used in ERA-Interim has been provided by Forbes et al. (2011, their Fig. 3). Clouds are assumed to be exclusively ice at temperatures below –40°C. Parameterized supersaturation with respect to ice applies at all temperatures below the freezing point in ERA5, rather than only at temperatures below –23°C as in ERA-Interim. JRA-55 uses a version of the approach suggested by Sommeria and Deardorff (1977) and modified by Smith (1990) to represent large-scale clouds at high altitudes. Cloud fraction depends on joint probability density functions (PDFs) of total water content and liquid water
temperature, assuming uniform distributions of both variables. Note that this formulation differs from the large-scale condensation scheme used to represent the evolution of marine stratocumulus, which follows Kawai and Inoue (2006). Partitioning between the ice and liquid phases is determined as a linear function of temperature between 0°C and –15°C (Fig. A1). Like JRA-55, MERRA-2 also uses a two-moment PDF-based approach to represent cloud cover and cloud water content, but with the total water PDF constrained as suggested by Molod (2012). Condensate formed in anvil clouds and condensate formed via
large-scale saturation are tracked separately in the prognostic cloud scheme, with 'anvil' condensate gradually converted to 'large-scale' condensate (Bacmeister et al., 2006). New condensate is partitioned among the liquid and ice phases as a linear function of temperature between 0°C and –20°C (Fig. A1), with liquid condensate gradually converted to ice in the prognostic





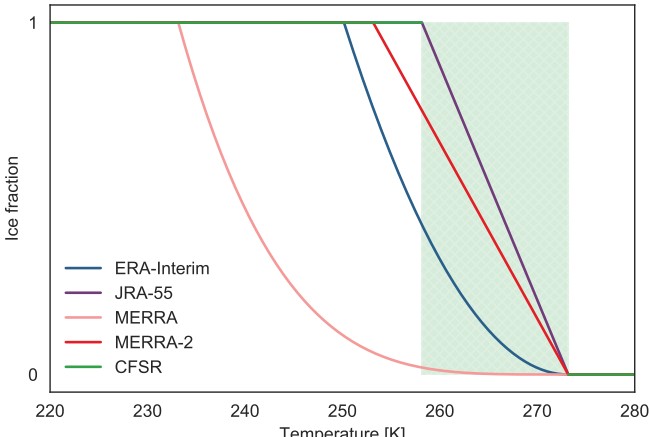

**Figure A1.** Fraction of new condensate in the ice phase as a function of temperature for ERA-Interim (blue), JRA-55 (purple), MERRA (light red), MERRA-2 (dark red), and CFSR (green). The green shaded region with hatching marks the range of temperatures for which new condensate in CFSR is assigned to the ice phase if ice already exists at or above the grid cell in question, and is assigned to the liquid phase otherwise.

scheme when temperatures are less than 0°C. The approach used in MERRA is similar to that used in MERRA-2, but with a quartic function governing the partitioning of new condensate into liquid and ice (Fig. A1) and without the constraints on total water proposed by Molod (2012). In CFSR and CFSv2, cloud water content is parameterized using the formulation of Zhao and Carr (1997). Cloud fraction is then diagnosed following Xu and Randall (1996). Cloud water content is the primary

determinant of cloud fraction, with RH a secondary contributor. The cloud scheme does not explicitly distinguish between liquid and ice. Condensate is assumed to be liquid for temperatures greater than 0°C and ice for temperatures less than –15°C (Fig. A1). At temperatures between these bounds, condensate is assumed to be liquid unless ice crystals already exist at or above the grid cell.

    All reanalyses allow for evaporation and sublimation of condensed water and ice, along with losses of condensate due to

autoconversion, accretion, and sedimentation. As with the parameterized formation of clouds, parameterizations of these loss processes differ amongst the reanalyses. For example, while all six reanalyses allow for condensate loss to vapor when grid-scale RH falls below a critical threshold, only ERA5, ERA-Interim, MERRA, and MERRA-2 explicitly include representations of 'cloud munching' (evaporative loss due to turbulent mixing with clear air near the edges of the cloud; e.g. Del Genio et al., 1996). These parameterizations depend on the saturation specific humidity or vapor pressure, and therefore have less effect for

clouds at high altitudes (where temperatures are low) than for clouds at low altitudes. The 'cloud munching' parameterizations in MERRA and MERRA-2 apply only to anvil-type condensate detrained from deep convection. Implementations of the critical threshold for cloud evaporation are also influential. For example, lowering the critical threshold from saturation to the critical RH used for cloud formation contributes to increases in cloud residence times and re-evaporation of ice particles between MERRA and MERRA-2 (Molod et al., 2012).





**Table A1.** Summary information on deep convective parameterizations used in the reanalyses. Here CAPE is convective available potential energy, PCAPE is an entraining CAPE evaluated in pressure coordinates (Bechtold et al., 2014), LCL is the lifting condensation level, ABL is the atmospheric boundary layer, and A-S stands for Arakawa–Schubert.

| Reanalysis | Plumes | Trigger | Closure | Cloud base | Detrainment |
|---|---|---|---|---|---|
| ERA5[a] | updraft: single downdraft: single | buoyancy > threshold Bechtold et al. (2006) | PCAPE + ABL coupling Bechtold et al. (2014) | LCL ~lowest 350 hPa | above max ascent (RH dependence) |
| ERA-Interim[a] | updraft: single downdraft: single | buoyancy > threshold Bechtold et al. (2006) | CAPE-based Gregory et al. (2000) | LCL ~lowest 350 hPa | above max ascent |
| JRA-55[b] | updraft: ensemble downdraft: single | dynamic CAPE Xie and Zhang (2000) | quasi-equilibrium | $p \sim 900\,\mathrm{hPa}$ | $T < 0°C$ |
| MERRA-2[c] | updraft: ensemble downdraft: ensemble | sub-cloud RH > 60% | quasi-equilibrium | ABL top | plume top |
| CFSR/CFSv2[d] | updraft: single downdraft: single | buoyancy > threshold Hong and Pan (1998) | quasi-equilibrium | LCL ~lowest 300 hPa | plume top |

[a] deep convection based on the scheme described by Tiedtke (1989).
[b] deep convection based on the 'economical prognostic' Arakawa–Schubert scheme described by JMA (2013).
[c] deep convection based on the 'relaxed' Arakawa–Schubert scheme described by Moorthi and Suárez (1992).
[d] deep convection based on the simplified Arakawa–Schubert scheme described by Pan and Wu (1995) and Moorthi et al. (2001).

## A2 Parameterizations of deep convection

All four reanalyses apply mass-flux representations of deep convection (e.g. Arakawa and Schubert, 1974; Tiedtke, 1989), but with substantially different treatments (Table A1). Mass-flux convective parameterizations represent the statistical effects of convection in a given grid cell via one or more updraft and downdraft plumes. Both updraft and downdraft plumes are then
coupled to the background environment via entrainment and detrainment, diabatic heating, and the vertical transport of tracers and momentum. Differences in the convective parameterizations used by the reanalysis systems include the trigger function, the principal closure, whether and to what extent momentum and tracer transport are included, constraints on the properties of the individual plumes (e.g. entrainment, detrainment, cloud base, and cloud top), and assumptions governing the production and partitioning of rainfall and cloud condensate.
ERA-Interim uses the scheme proposed by Tiedtke (1989), with a single pair of plumes representing updrafts and downdrafts. Deep convection is triggered when the updraft vertical velocity diagnosed at the lifting condensation level (LCL) is positive and the estimated cloud depth exceeds 200 hPa (Bechtold et al., 2006). Convection can be triggered from any level in the lowest 350 hPa of the atmosphere. Active convection consumes convective available potential energy (CAPE) over a specified time scale of 60 minutes. ERA5 uses the same core convection scheme as ERA-Interim (Table A1), but with several important modifications.
The deep convective closure has been reformulated in terms of an effective CAPE where only a fraction of the daytime surface heating is available for deep convection and the remainder goes into turbulent and shallow convective mixing of the boundary-layer. This produces a more realistic diurnal cycle of convection over land, with maximum convective rainfall and heating occuring in the late afternoon as opposed to around noon in ERA-Interim (Bechtold et al., 2014). The convective adjustment





time scale has also been set proportional to convective turnover, replacing the constant time scale for CAPE consumption used in ERA-Interim (Bechtold et al., 2008). JRA-55 uses the 'economical prognostic Arakawa–Schubert' scheme developed by the Japan Meteorological Administration (JMA, 2013). Convection is triggered using the 'dynamic CAPE' approach proposed by Xie and Zhang (2000), in which convection occurs when the time rate of change in CAPE due to large-scale forcing exceeds a

critical value. Cloud base is restricted to the model level at $\sim$900 hPa. The convective closure is based on a modified version of the 'quasi-equilibrium' hypothesis, in which the generation of convective instability by the large-scale circulation is balanced by an ensemble of convective plumes that act to reduce the cloud work function below zero (Arakawa and Schubert, 1974). MERRA-2 uses the relaxed Arakawa–Schubert parameterization proposed by Moorthi and Suárez (1992). The convection scheme is triggered when the sub-cloud RH exceeds 60%. Convection is then represented via an ensemble of plumes with

different levels of entrainment, subject to a Tokioka-type entrainment condition (Bacmeister and Stephens, 2011). The scheme randomly samples an empirically-based power-law distribution to set a minimum entrainment rate, disallowing any plume for which the diagnosed entrainment rate falls below this level. This triggering procedure means that MERRA-2 only occasionally permits the deepest convective clouds (Lim et al., 2015). Cloud base in MERRA-2 is defined as the top of the atmospheric boundary layer (ABL). A modified CAPE-based closure is used to determine mass flux for each plume at this cloud base. The

ensemble of convective plumes acts to gradually relax the environment toward a specified equilibrium state. Convection in MERRA is similar, but without the stochastic trigger (potentially allowing more low-entrainment plumes with very high cloud tops) and with cloud base assigned to the lowest two model levels (rather than the boundary layer top). CFSR and CFSv2 use the simplified Arakawa–Schubert parameterization proposed by Pan and Wu (1995), updated as described by Moorthi et al. (2001). The convective trigger couples boundary layer turbulence and deep convection following the approach proposed

by Hong and Pan (1998). Convection occurs when an air parcel corresponding to the maximum moist static energy (MSE) within the boundary layer would be positively buoyant at the LCL. Sub-grid variability associated with surface conditions, parameterized turbulent mixing in the boundary layer and lower free troposphere, grid-scale vertical velocity, and entrainment during ascent to the LCL are considered. The cloud base can be any level between the surface and 700 hPa, provided the trigger condition is met. Convective closure is based on the quasi-equilibrium hypothesis as in JRA-55.

Different treatments of entrainment into convective clouds and detrainment from convection into the large-scale cloud scheme also have important impacts on the behaviors and distributions of high clouds in reanalyses. ERA-Interim allows for turbulent exchange through the lower half of the convective column (equal entrainment and detrainment at fractional rates of $1.2 \times 10^{-4}\,\mathrm{m}^{-1}$), as well as organized entrainment below the level of maximum ascent and organized detrainment above this level. Organized entrainment is diagnosed as proportional to moisture convergence and organized detrainment according

to decreases in upward mass flux assuming a constant cloud area. ERA5 includes several major changes to entrainment and detrainment (Bechtold et al., 2008). First, the dependence of organized entrainment on large-scale moisture convergence has been eliminated and replaced by a local approach where the bulk entrainment of positively buoyant plumes decreases with height according to the saturation specific humidity. The base entrainment rate at cloud base of $\mathrm{O}(10^{-3}\,\mathrm{m}^{-1})$ is also an order of magnitude larger than that in ERA-Interim and more in line with data from large-eddy simulations. This adjustment allows

a unified treatment of the turbulent and organized components of entrainment. Second, RH-dependent factors have been in-





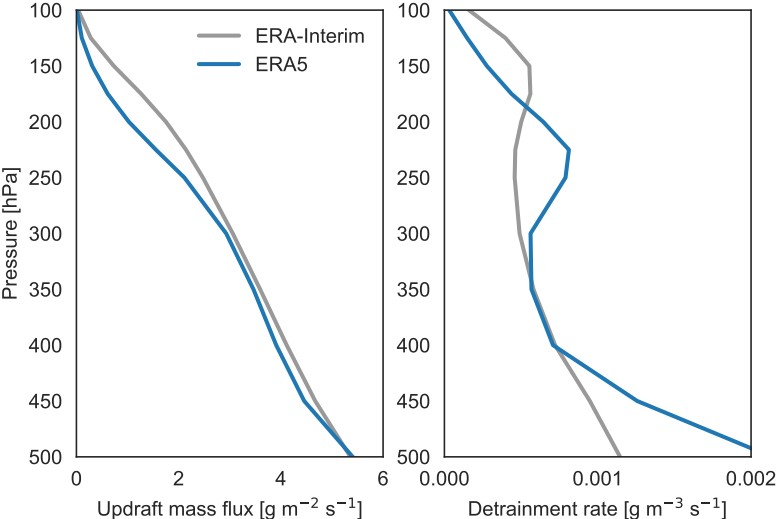

**Figure A2.** Vertical profiles of updraft mass fluxes and fractional detrainment rates in ERA5 (blue) and ERA-Interim (grey) during January 2010.

troduced for both entrainment and detrainment. Outside of this new RH dependence the treatments of turbulent and organized detrainment are similar to those in ERA-Interim, but with a reduced turbulent detrainment rate ($0.75 \times 10^{-4}\,\mathrm{m}^{-1}$). Figure A2 shows that fractional detrainment rates in ERA5 are enhanced in both the middle ($\sim500\,\mathrm{hPa}$) and upper ($200\sim300\,\mathrm{hPa}$) troposphere relative to ERA-Interim, but reduced in the TTL ($100\sim150\,\mathrm{hPa}$). This reflects larger variability in the cloud field, with

a more realistic occurence of cumulus congestus clouds and fewer quasi-undilute convective cores reaching the TTL relative to ERA-Interim. JRA-55 diagnoses entrainment rates for each deep convective plume based on a zero-buoyancy condition at cloud top, suppressing fractional entrainment rates greater than $1 \times 10^{-3}\,\mathrm{m}^{-1}$. Detrained cloud water is distributed among layers with temperatures below the freezing level according to a fixed, height-dependent relationship for partitioning rain and cloud water content. MERRA-2 specifies the cloud top for each updraft plume in the ensemble, with all model levels between

$p = 100\,\mathrm{hPa}$ and the level immediately above cloud base considered as candidates. Assuming that cloud top corresponds to the level of neutral buoyancy (LNB) for a candidate plume, the entrainment rate for that plume is then diagnosed based on conditions at the cloud base. Only plumes with diagnosed entrainment rates larger than the stochastically-determined minimum are triggered. In CFSR and CFSv2, the cloud top is randomly chosen from the set of levels between the level of minimum MSE and the LNB. The base entrainment rate ($1 \times 10^{-3}\,\mathrm{m}^{-1}$) is then adjusted to achieve this randomly-chosen cloud top. Detrainment

in both MERRA-2 and CFSR/CFSv2 occurs exclusively at the plume top. However, where MERRA-2 considers an ensemble of plumes with different entrainment rates, CFSR and CFSv2 use only a single pair of updraft/downdraft plumes.





**Table A2.** Summary information on radiation parameterizations, cloud overlap, and cloud optical properties used in the reanalyses. In the column labeled 'Optical properties', L stands for liquid water clouds and I for ice clouds. Sources of cloud optical properties may differ between the LW and SW schemes.

| | Radiation scheme | | Cloud representations | |
| Reanalysis | Longwave | Shortwave | Overlap | Optical properties |
|---|---|---|---|---|
| ERA5 | RRTMG-LW Iacono et al. (2008) 16 bands (3.08–1000 μm) | RRTMG-SW Iacono et al. (2008) 14 bands (0.2–12.195 μm) | McICA w/ generalized overlap | $L_{SW}$: Slingo (1989) $L_{LW}$: Lindner and Li (2000) $I_{SW}$: Fu (1996) $I_{LW}$: Fu et al. (1998) |
| ERA-Interim | RRTMG-LW Mlawer et al. (1997) 16 bands (3.33–1000 μm) | Fouquart and Bonnel (1980) 6 bands (0.2–4.0 μm) | max–random | $L_{SW}$: Fouquart (1988) $L_{LW}$: Smith and Shi (1992) I: Ebert and Curry (1992) |
| JRA-55 | Murai et al. (2005) 11 bands (3.33–400 μm) | Briegleb (1992) Freidenreich and Ramaswamy (1999) 16 bands (0.174–5.0 μm) | max–random (LW) random (SW) | $L_{SW}$: Slingo (1989) $L_{LW}$: Hu and Stamnes (1993) I: Ebert and Curry (1992) |
| MERRA-2 | CLIRAD-LW Chou et al. (2001) 11 bands (3.33–400 μm) | CLIRAD-SW Chou and Suárez (1999) 10 bands (0.175–10.0 μm) | max–random | L: Tsay et al. (1989) $I_{SW}$: Fu (1996) $I_{LW}$: Fu et al. (1998) |
| CFSR | RRTMG-LW Clough et al. (2005) 16 bands (3.08–1000 μm) | RRTMG-SW Clough et al. (2005) 14 bands (0.2–12.195 μm) | max–random | L: Hu and Stamnes (1993) $I_{SW}$: Fu (1996) $I_{LW}$: Fu et al. (1998) |
| CFSv2 | RRTMG-LW Clough et al. (2005) 16 bands (3.08–1000 μm) | RRTMG-SW Clough et al. (2005) 14 bands (0.2–12.195 μm) | McICA w/ max–random overlap | L: Hu and Stamnes (1993) $I_{SW}$: Fu (1996) $I_{LW}$: Fu et al. (1998) |

## A3 Parameterizations of radiative transfer

Details of the radiation parameterizations and their treatments of clouds are listed in Table A2. All of the parameterizations are broadband schemes, in which the radiative spectrum is discretized into a discrete set of spectral bands. The form of this discretization is dictated primarily by the presence of radiatively active constituents in the atmosphere and the wavelengths at which these constituents are active (e.g. Clough et al., 2005). Each band may feature parameterizations of radiative transfer due to multiple species, as well as scattering, absorption, and emission by clouds or aerosols. Radiative fluxes and heating rates (defined as the convergence of radiative fluxes) are computed by integrating across all spectral bands. ERA-Interim, JRA-55, MERRA-2, and CFSR (ending in 2010) all assume maximum–random overlap for cloudy columns: cloud layers that are contiguous in the vertical are assumed to have maximal overlap and cloud layers that are not contiguous in the vertical coordinate are assumed to overlap randomly. The Monte Carlo Independent Column Approximation (McICA; Pincus et al., 2003) is used in ERA5 (with generalized overlap; Morcrette et al., 2008) and CFSv2 (with maximum–random overlap, starting from 2011; Saha et al., 2014). The introduction of McICA is therefore a potential source of discontinuity at the CFSR–CFSv2 transition. Representations of the optical properties of ice and liquid water clouds are also noted in Table A2.



*Data availability.* ERA-Interim products were acquired from the public MARS archive maintained by ECMWF (http://apps.ecmwf.int/datasets). ERA5 products were acquired from the Copernicus Climate Data Store (https://cds.climate.copernicus.eu) and the ECMWF MARS archive. JRA-55 products were obtained from archives maintained by the Japan Meteorological Agency (http://jra.kishou.go.jp/JRA-55) and the NCAR Research Data Archive (RDA; https://rda.ucar.edu). CFSR and CFSv2 products were acquired exclusively through the NCAR RDA.

MERRA and MERRA-2 products were obtained from the NASA Goddard Earth Sciences Data and Information Services Center (GES DISC; https://disc.gsfc.nasa.gov/daac-bin/FTPSubset2.pl). Access dates for these products range from March 2015 to August 2019 depending on reanalysis, variable, and temporal resolution. ISCCP products were acquired from the NOAA National Centers for Environmental Information (https://www.ncdc.noaa.gov/isccp; accessed 9 December 2019). CERES data were obtained from the NASA Langley Research Center Atmospheric Science Data Center, and AIRS data from the GES DISC; see related data citations for availability and access in-

formation. CloudSat–CALIPSO combined cloud fractions were provided by Jennifer Kay (personal communication, 15 December 2017), and CFMIP-GOCCP products by the Institute Pierre Simon Laplace (http://climserv.ipsl.polytechnique.fr/cfmip-obs/goccp_v3.html; v3.1.2 accessed 21 June 2018). CloudSat IWC retrievals were acquired from the CloudSat ftp server (ftp.cloudsat.cira.colostate.edu; CWC-RO R04 v5.1). NOAA Interpolated OLR data were acquired from the NOAA/OAR/ESRL Physical Science Division, Boulder, Colorado, USA (https://www.esrl.noaa.gov/psd; accessed 23 March 2017). The NASA GEWEX-SRB data were acquired from the NASA Langley Atmo-

spheric Science Data Center (https://gewex-srb.larc.nasa.gov; v3.1 accessed 6 July 2019).

*Author contributions.* JSW and XS conducted the analysis and wrote the initial draft. XZ processed the CloudSat IWC data and helped to interpret the results. PK, KK, AM, ST, and GJZ suggested additional analyses and revisions for the manuscript. AM checked and clarified key aspects of MERRA-2.

*Competing interests.* The authors declare no competing interests.

*Acknowledgements.* This work was supported by the Ministry of Science and Technology of the People's Republic of China (2017YFA0603902) and a joint research project funded by the National Natural Science Foundation of China (NSFC project number 41761134097) and the German Research Foundation (DFG project number 392169209). We thank Peter Bechtold from ECMWF, Yayoi Harada and Ryoji Nagasawa from the JMA Meteorological Research Institute, and Kris Wargan from NASA GMAO for help in clarifying aspects of the reanalysis data. We are also grateful to Xianglei Huang for advice on OLR data and radiative transfer models, to Jennifer Kay for providing the CloudSat–

CALIPSO data, and to Yi Qin and Xiaolu Yan for helpful discussions. A condensed version of this material has been prepared for Chapter 8 of the S-RIP report. We thank Masatomo Fujiwara, Gloria Manney, and Lesley Gray for their leadership of this activity, and all S-RIP participants for their contributions.



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
