# Peer review of "Differences in tropical high clouds among reanalyses: origins and radiative impacts"

_Atmospheric Chemistry and Physics, 2019_

## Referee Comment (RC1) · Yi Huang (Referee) · 25 Feb 2020

This paper compares a few reanalyses with a focus on their representation of tropical high clouds. As the reanalysis datasets surveyed here are widely used, it is useful to detect and document their biases. Besides documenting the inter-dataset differences, this paper makes especial efforts to relate the cloud biases to radiation impacts and to discuss possible physical causes (how they may have resulted from parameterization schemes) in respective reanalysis models. My assessment is that this paper is well motivated, logically organized and likely to be widely cited if published. I do have a few comments, which hopefully help improve the clarity of the paper. I recommend this paper be accepted for publication after these comments are addressed.

[Figure]

1. My first comments concern the presentation of the results. First, it baffles me why many of the biases (e.g., fig. 1 and 5) are presented with respect to multi-reanalyses mean (MRM) instead of observations. While it is understood that the observational datasets are subject to their own uncertainties and sampling discrepancies, it is still of the interest of most readers to see how each of the reanalyses compares to an observational ground truth. I strongly suggest the bias results be presented with respect to relevant observations wherever available.

Second, I find some results are presented in unconventional, and probably not advantageous, ways. One example is Fig 11. The authors may be seeking a concise way to present rich information from many variables: HTR, RH, CRE, CWC, etc., although the plots become difficult to interpret. I suggest the authors decouple these variables and use more straightforward plots to evidence their points, or, less preferably, identify what features are for the readers to recognize and explain how they relate to their points.

My last complaint about presentation is that I find some potentially very interesting and important results omitted. This applies to a few places: Fig 5. What about SW and net (LW+SW) results? A central radiative question about the high clouds is to what extent their LW and SW effects compensate [e.g., Kolly & Huang 2018; Wall et al. 2019] and how different datasets may bias this compensation [e.g., Zhu et al. 2019 Fig. S1 and relevant texts]. Fig. 10. Why not show the three related variables: T, q and z (components of MSE), respectively here? Fig. 13. What about the clear-sky OLR?

2. A technical comment: it should be cautioned that CRE, defined as the difference between clear- and all-skies, is subject to influence of clear-sky [e.g., Soden 2004]. I'd suggest where appropriate (e.g., Fig. 5) clear-sky biases be also examined to ensure that the CRE difference measures cloud effect, instead of being affected by the clear-sky differences between the reanalyses.

3. Additional comments

P3, ~L5. Note there are methods, such as latent heat nudging and particle filter, that

make use of cloud and precipitation info in data assimilation.

P4 31. Another benefit of simulator, if properly configured, is that it also addresses sampling consistency issue.

P4, L33. I'm surprised to read that the aim is stated to be "qualitative" – despite many quantitative – why? If q

P6, L27; P13 L26. Note the latest CERES data includes a version of clear-sky values, computed using the same clear-sky definition as in GCM [Loeb 2019].

P7, L15, Why linear with ln(p) instead of p?

P9, L8. Note that cloud top temperature (CTT) is another potential cause of (compensating) errors.

P11, ~L22 and Fig. 4. Are the CWC averaged over only cloudy profiles or over both cloudy and clear profiles? Consistently between all the reanalyses? Both averages would be of interest to compare.

P15, Fig. 6. How is the purple line drawn exactly?

P17, L30. OLR vs. CRE – this seems significant methodological difference. What's the rational to use CRE here?

P18, Fig. 7. I am surprised to see the lack of distinctions in the ERAi results here, reminiscent of fig. 10e of Zhang et al. [2017]. Some basic radiative signatures such as cloud top cooling and cloud bottom warming are totally missing. May this be related to the use of CRE as a state indicator - it may fail to identify cloud effect due to clear-sky bias difference (see comments above)? Would it be useful to simply use cloud fraction instead to identify the regimes (Q1-4)? How does cloudy heating rate profile compare to clear-sky in this reanalysis?

P19, Fig. 8. Is it LW or net radiative heating that is used to define LZRH and the relevant results?

[Figure]

P20, L20. Isn't it the diabatic heating (instead of radiative only) that better inferences ascent/descent? Why focused on radiative?

P21 "Possible origins". The discussions in this section don't distinguish cause and effect of previously presented cloud biases. Maybe worth some clarification and discussion here about this.

P24, L5. The way the current plot is made makes it difficult to discern the "kink".

P26, section 5.2 and Fig 11. Besides the confusing way the figure is presented as noted above, I find the discussion here doesn't show enough recognition of the cloud position with respect to the respective levels focused (100 and 150 hPa). A basic signature of clouds is cloud top cooling and bottom warming. The sign and magnitude of the cloud radiative impact is strongly dependent on where the clouds are placed.

P29, L31. "systemic" => "artificial"?

P29, L32 and Fig. 13. A striking feature is that OLR doesn't seem "jumped" despite of the cloud fraction jump! How could cloud cover change be consistent with OLR and CRE with regard to long-term trend but inconsistent with regard to this jump? Also, the long-term trends seem similar among the reanalyses – should this be taken seriously as a sign of real trend in nature? It is important to reason and caution whether we can use these reanalyses for studying climate trend in this critical region. It should be noted that the all-sky OLR trend appears to contradict the FAT hypothesis [Hartmann & Larson 2002].

P30, L23. The clear-sky OLR change described sounds very interesting and ought to be shown. Relevant to the above point, another important question the reanalyses may or may not answer is whether broadband fluxes, either clear- or all-sky, may be useful for climate change monitoring. As shown by Huang & Ramaswamy [2009, Fig. 5], there may be intrinsic compensation between greenhouse gas forcing and Planck response that results in no trend signal. This point, together with the above one, is

worth noting and discussing here.

P31, Fig. 13. Some of the time series apparently don't have zero mean. How are the anomalies defined?

P34, Summary. A general suggestion for this section is to reference the respective summary points to the relevant figures.

P34, L20. It is striking to find the lack of agreement among the studies in terms of what direction cloud drives the LZRH. Can you discuss why and how would one elucidate this matter?

P36, L29. Cloud top temperature, as related to in some of the above comments, is perhaps another aspect to note.

P37, Appendix. It may be worth reviewing the difference in assimilated data in this appendix as well. This is apparently relevant to the trend discussions (section 6) and potential affects climatology as well.

P37, L25. What is liquid water temperature?

P43, L1. Sufficient info to ensure reproducibility of the results should be included. Regarding the data sources, how were the data, such as ERA5 heating rates, obtained exactly, as they are not normally available from the webpages stated here? If scripts were used, it is useful to post a sample script and explain how relevant parameters, e.g., analysis vs. forecast and, if latter, forecast times and steps, are set. Moreover, are the these parameters set consistently for all the variables: heating rate and state variables such as cloud fraction, temperature, humidity, etc. from the same time steps?

References

Hartmann, D. and K. Larson, (2002), An important constraint on tropical cloud ‐ climate feedback, https://doi.org/10.1029/2002GL015835

Huang, Y. and V. Ramaswamy (2009), Evolution and trend of outgoing longwave radiation spectrum, J. Climate, doi:10.1175/2009JCLI2874.1.

Kolly, A. and Y. Huang, (2018), The radiative feedback during the ENSO cycle: observations vs. models, J. Geophys. Res.-Atmos. https://doi.org/10.1029/2018JD028401

Loeb, N.G., F.G. Rose, S. Kato, D.A. Rutan, W. Su, H. Wang, D.R. Doelling, W.L. Smith, and A. Gettelman, 2020: Toward a Consistent Definition between Satellite and Model Clear-Sky Radiative Fluxes. J. Climate, https://doi.org/10.1175/JCLI-D-19-0381.1

Soden, B.J., A.J. Broccoli, and R.S. Hemler, 2004: On the Use of Cloud Forcing to Estimate Cloud Feedback. J. Climate, https://doi.org/10.1175/1520-0442(2004)017<3661:OTUOCF>2.0.CO;2

Wall, C.J., D. L. Hartmann and J.R. Norris , 2019: Are cloud radiatie effects constrained to cancel over the tropical warm pools? Geophys. Res. Lett.. , , doi;10.1029/2019GL083642.

Zhu, T., Y. Huang and H. Wei, (2019), Estimating climate feedbacks using a neural network, J. Geophys. Res.-Atmos. https://doi.org/10.1029/2018JD029223

---

## Referee Comment (RC2) · Anonymous Referee #2 · 17 Mar 2020

Review of ACP-2019-1187 : Differences in tropical high clouds among reanalyses: origins and radiative impacts.

This is a substantial comparison of five widely used reanalyses that exhibit a range of cloud behaviours near the tropical tropopause. Geographical and vertical distributions of high top cloud fractions and cloud water contents are evaluated. There is, in addition, an exploration of cloud radiative impacts and their wider influence on the atmosphere, through changes to the vertical profile of radiative heating in the tropics. The authors also take some care to investigate possible reasons for differences identified between the reanalyses as declared in their aims for the paper. As such the comparison makes informative reading for anyone interested in the relative performance of these reanalyses and their potential for estimating temperature biases or validating cloud and radiation metrics in unconstrained model simulations.

Specific / technical comments:

Page 4, l23: might it be clearer to say '… involved changes in the cloud fields which are much larger …'?

Page 4, l32: suggest '… we stress that most …' without 'the'.

Page 5, Table 1 / section 2.1: Table 1 provides a useful "at a glance" summary of the characteristics of the different observation datasets. There would be much to be said for extending it by the 5 or 6 lines needed to add equivalent details for the different reanalyses. I take the point made in section 2.1 that this paper provides detailed information of key aspects regarding cloud etc. and that other reviews have covered more general information, for those wishing to spend the time to track it down, but a brief summary set of 'vital statistics' would reduce the reliance the paper is otherwise placing upon a reader's prior knowledge of the reanalyses.

Page 7, l3: do you intend 'specific heat constant' rather than 'heat capacity'?

Page 14, Fig5 caption: 'with values for OLR listed above …' actually look as though the OLR mean values on the Figure are beneath?

Page 15, Fig 6 caption: 'is marked in the upper row.' Does this refer to the violet lines with numbers beside them in A, C-G?

Page 15, l6: is 'are analogous to scatter plots containing millions of points' a roundabout way of saying 'represent a 2-dimensional probability density function'?

Page 17, l5: 'presumably owing' reads a little loosely to a reviewer. But more precisely, at the bottom of page 16 the point has been made that MERRA-2 can

persist large cloud fraction for declining cloud water, which implies reduced in-cloud water contents that might well lead to different LWCRE. It is thus unclear when the issue of characteristic lifetimes appears in the text whether this is meant to have been inferred from the differences in in-cloud water content or a separate (and presumably verifiable) behaviour of the different schemes discussed that is however (not shown). This sentence would benefit from clarification.

Page 17, l13: 'more than 140Wm-2' seems a bit redundant when sitting beside 'from -100 to +40 Wm-2'

Page 23, Table 2 caption: 'all data points' (plural)?

Page 23, ll2-13: To me, lines 10-13 'Note … independent.' seem to follow more naturally from the statement of similarity that ends on line 4 '(Table 2).' In addition, the lines 4-7 seem to spend a lot of time simply repeating the data that readers can see in Table 2 whereas I think they would add more if they pulled out what seems to be the key message by showing the differences [Q4 –All] that appear scattered either side of the observed 1K. e.g. 'CFSR exhibits the weakest increase in SST (0.7K) between mean cloud and high …' or 'observation …, assigns a mean value to Q4 that is 1K warmer than the tropical mean.'

Page 25, l5: The first half of the first sentence would be better if merged into the following sentence 'Distributions of 500hPa grid-scale vertical velocity (w) for the whole tropics (Figure 9C) …' and the second half similarly into line 2 of P26.

Page 26, l16-18: there is a lot of repetition that could be reduced by swapping the order in line 16, for instance, 'so that plumes are only permitted to reach the upper troposphere when entrainment rates are small, that is potentially smaller than the … Tokioka parameter.'

Page 33, Figure 14 caption: are the separation lines 'Solid grey' or 'Solid black'? (I do like the plot, though!).

Page 36, l18: 'occur in …' rather than 'into'?

Page 36, l33: I found this sentence somewhat hard to digest, especially as a finale. I would suggest putting a stop in at 'in the reanalysis models.' then rewording the rest into a following sentence as appropriate.

Page 42, l3: 'discrete' is redundant here as 'discretization' guarantees it!

---

## Author Comment (AC1) · 14 May 2020

**Response to reviewers, ACP-2019-1187**
**'Differences in tropical high clouds among reanalyses: origins and radiative impacts.'**

We extend our thanks to both reviewers for your positive feedback and your thoughtful comments and advice. Following your suggestions, we have eliminated the multi-reanalysis mean, benchmarking the reanalyses against observed distributions instead. We have also adopted the most recent version of CERES EBAF, expanded our discussion of clear-sky and net radiative fluxes at the top-of-atmosphere, added further information on the reanalysis systems examined in the paper, and clarified several aspects of the methodology.

**Response to major comments:**
1. My first comments concern the presentation of the results. First, it baffles me why many of the biases (e.g., fig. 1 and 5) are presented with respect to multi-reanalyses mean (MRM) instead of observations. While it is understood that the observational datasets are subject to their own uncertainties and sampling discrepancies, it is still of the interest of most readers to see how each of the reanalyses compares to an observational ground truth. I strongly suggest the bias results be presented with respect to relevant observations wherever available.

AR: This choice has been a subject of much internal discussion as well. We initially opted to use the MRM for three reasons. First, this study is part of the SPARC Reanalysis Intercomparison Project, which is primarily aimed at intercomparison of reanalyses against each other. Second, using the MRM allows us to use the same 1980-2014 base period for both figures (initially this would have been 1984-2009 to match ISCCP and 2001-2014 to match CERES). Third, we view the satellite cloud products in particular as a questionable quantitative benchmark for the reanalyses (a large part of the motivation for focusing on 'qualitative' rather than 'quantitative' comparisons of reanalysis and observationally-based cloud fields, as mentioned below).

After further discussion, we decided that the first rationale can be mitigated by offering the MRM-based presentation in the S-RIP report and the observationally-based presentation here; the second by the H-series extension of ISCCP to recent years and an overall weak sensitivity of the results to the choice of base period (with the notable exception of the CFSR / CFSv2 transition as discussed in section 6); and the third by CERES EBAF being a more suitable quantitative benchmark for OLR than the MRM. Accordingly, we adopted your suggestion to use ISCCP with a 1984-2014 base period for high cloud cover and CERES EBAF with a 2001-2014 base period for OLR. To help make room for several additions in the revised text, we have eliminated both the MRM concept and maps based on MERRA from the paper. We retain the MERRA profiles in Figure 3F and Figure 4F to help illustrate the effects of model changes between MERRA and MERRA-2. The MRM and MERRA maps will still be included in chapter 8 of the S-RIP report.

Adopting the new benchmark directly works well for OLR (Figure R1, which replaces Figure 5 in the main text). We have inverted the color scale for OLR so that dark colors of OLR represent low values, which helps to emphasize the difference between the difference plots and the CERES climatology, since most of the reanalyses produce OLR larger than indicated by CERES.

[Figure]

**Figure R1:** Climatological mean spatial distributions of all-sky outgoing longwave radiation (OLR; shading) and clear-sky outgoing longwave radiation (CLR; contours at intervals of 10 W m$^{-2}$) for (A) CERES EBAF over 2001–2014. Differences relative to CERES EBAF for the same period are shown for (B) ERA5, (C) ERA-Interim, (D) JRA-55, (E) MERRA-2, and (F) CFSR/CFSv2. Contours in panels (C) through (F) cover the range within ±10 W m$^{-2}$ at intervals of 4 W m$^{-2}$. Tropical mean (30°S–30°N) values of OLR and CLR based on each product are shown at the upper right and left corners, respectively, of the corresponding panel. Tropical mean values for the longwave cloud radiative effect (LWCRE; CLR − OLR) are listed above those for OLR.

The high cloud fraction plot needed a little more modification: simply replacing the MRM with ISCCP made it difficult to describe the inter-reanalysis differences in the text and seemed to contradict our statement that "direct comparisons between cloud variables derived from observations and those derived from models may be misleading." To address this, we show maps of HCC for both ISCCP and each reanalysis with contour overlays to show differences relative to ISCCP (Figure R2, which replaces Figure 1 in the text).

[Figure]

**Figure R2:** Climatological mean spatial distributions of high cloud cover (HCC) for (A) ISCCP HGM, (B) ERA5, (C) ERA-Interim, (D) JRA-55, (E) MERRA-2, and (F) CFSR/CFSv2 over 1984–2014. Differences relative to ISCCP HGM are shown for each reanalysis as orange contours (dashed for negative values) at intervals of 0.1. The area-weighted tropical mean (30°S—30°N) HCC based on each product is shown at the upper right corner of the corresponding panel.

Second, I find some results are presented in unconventional, and probably not advantageous, ways. One example is Fig 11. The authors may be seeking a concise way to present rich information from many variables: HTR, RH, CRE, CWC, etc., although the plots become difficult to interpret. I suggest the authors decouple these variables and use more straightforward plots to evidence their points, or, less preferably, identify what features are for the readers to recognize and explain how they relate to their points.

**AR: Thank you for this comment. The purpose of Figure 11 is to examine differences in cloud fields within the TTL, above the typical levels of convective detrainment and often above the level of zero net radiative heating. Clouds in this layer are primarily associated with slow radiatively-balanced ascent, and occasionally with very deep convection that penetrates into the TTL (e.g., Fueglistaler et al., 2009). We expect these two cloud populations to be distinguished by their cloud water contents (smaller for in situ cirrus; larger for convective overshoots) and radiative heating rates (weak radiative heating for slow ascent; usually strong cloud-top cooling for deep convection, though the latter depends on anvil depth as shown in previous Fig 12). Choosing radiative heating as one axis thus helps to distinguish the different types of clouds at these levels: (1) in situ clouds, which occur at high RH in tandem with weak positive heating rates (i.e., close to the 'spine' of the plot); (2) deep convection that detrains near the base of the TTL, which is associated with large CWCs and negative radiative heating (the left 'wing', exemplified by the blue profiles in previous Fig 12); and (3) deep convection that penetrates to near the tropopause and detrains within the TTL, which is associated with larger CWCs and positive heating rates (the right 'wing', red profiles in previous Fig 12). Selecting RH as the other axis helps to highlight some important differences and unexpected features, including differences in ice supersaturation and in situ cloud occurrence between ERA5 and ERA-Interim (supersaturation and related clouds are typically collocated with deep convective areas in ERA-Interim but form mostly away from deep convective areas in ERA5); the unexpected prevalence of ice supersaturation in MERRA-2, especially at 150 hPa; and the unrealistic behavior of CFSR water vapor fields at these levels. We have added text along these lines to better to better explain the motivation and interpretation of this figure (see below).**

My last complaint about presentation is that I find some potentially very interesting and important results omitted. This applies to a few places: Fig 5. What about SW and net (LW+SW) results? A central radiative question about the high clouds is to what extent their LW and SW effects compensate [e.g., Kolly & Huang 2018; Wall et al. 2019] and how different datasets may bias this compensation [e.g., Zhu et al. 2019 Fig. S1 and relevant texts].

**AR: Thank you for this suggestion. We have produced a figure that shows the distribution of the TOA net radiative flux (all-sky and clear-sky) together with the tropical mean net cloud effect. This figure is included here as Fig R3, again adopting the 2001-2014 period for overlap with CERES (the results are only weakly sensitive to this choice). Rather than difference plots, we prefer to show the absolute distribution of net radiative flux for each reanalysis here, and have taken care to specify this difference relative to Figs 1 and 5 in the caption. We have added this as Fig 6 in the revised manuscript, along with the accompanying text:**

"Figure 6 shows spatial distributions of all-sky net radiation based on CERES EBAF and the five reanalyses, with positive values indicating time-mean energy fluxes into the tropical climate system. Mean values across the tropics are positive (incoming solar radiation exceeds OLR), as indicated here by CERES EBAF (net gain of 45.0 W m$^{-2}$). This excess of incoming energy in the already energy-rich tropics is an essential component of the 'heat engine' driving the circulations of the atmosphere and ocean, and is contributed primarily by imbalances in the clear-sky fluxes (e.g. Stephens and L'Ecuyer, 2015, and references therein). Net clear-sky fluxes into the tropics are typically somewhat larger in the reanalyses than in CERES, with overestimates as large as 7 W m$^{-2}$ (in ERA-Interim). The closest match in the tropical mean is provided by JRA-55, which is within 0.1 W m$^{-2}$ of CERES; however, this good agreement does not extend to the all-sky net radiation flux, as detailed below. Cloud effects reduce the energy excess provided by clear sky radiation, as the negative SWCRE (cloud albedo) outweighs the positive LWCRE. However, most of the reanalyses greatly overestimate the magnitude of this reduction relative to CERES. Such overestimates have implications for atmospheric energy transport, and could result at least in part from the lack of two-way coupling between cloud fields and SST in the reanalyses (e.g. Kolly and Huang, 2018; Wall et al., 2019). For JRA-55, which overestimates the net CRE by 22.5 W m$^{-2}$ relative to CERES, a little more than half of the bias in the net CRE is attributable to the bias in the LWCRE. The remainder is due to overestimated cloud albedo effects. Similar ratios hold for ERA5 and ERA-Interim, with biases in the LWCRE contributing approximately 55% of the overall biases in each case. For MERRA-2, overestimated cloud albedo effects more than compensate for the stronger LWCRE, producing a net CRE similar to that in ERA5 (approximately 9 W m$^{-2}$ stronger than that from CERES). CFSR/CFSv2 produces a net CRE very similar to that indicated by CERES, implying compensating biases in the SWCRE and LWCRE. However, the horizontal gradients of net TOA radiation are much sharper in this reanalysis than in any of the other data sets shown in Fig. 6."

[Figure]

Figure R3: Climatological mean spatial distributions of all-sky net incoming radiation (ALL; shading) for (A) CERES EBAF, (B) ERA5, (C) ERA-Interim, (D) JRA-55, (E) MERRA-2, and (F) CFSR/CFSv2 during 2001–2014. Tropical mean (30°S–30°N) values of ALL and clear-sky net incoming radiation (CLR) based on each product are shown at the upper right and left corners, respectively, of the corresponding panel. Tropical mean values for the net cloud radiative effect (CRE = CLR − ALL) are listed above those for ALL. Positive values indicate time-mean energy fluxes into the tropical climate system.

Fig. 10. Why not show the three related variables: T, q and z (components of MSE), respectively here?

**AR: The motivation for showing MSE is because of its links with both the occurrence and effects of convection, but we like this idea as well. In the revised submission, along with the profiles of MSE, we have included distributions of $c_pT$, $L_vq$, and $gz$ for Q4 in each of the five reanalyses at the 300 hPa, 500 hPa, and 850 hPa levels (Fig R4; now Fig 11 in the revised text). Note that ERA5 is adopted as the reference rather than ERA-Interim (as in the original text for comparison with MERRA-2) or AIRS (as the 'observational' distribution in the profile panel). This modification supports and expands the existing discussion in the paper.**

[Figure]

**Figure R4: (A) Composite vertical profiles of moist static energy (MSE) for ERA5 (cyan), ERA-Interim (blue), JRA-55 (purple), MERRA-2 (red), and CFSR (green) averaged for the upper (Q4; thick lines) and lower (Q1; thin lines) quartiles of daily-mean LWCRE during 2001–2010. Profiles calculated from AIRS observations (September 2002–December 2010; grey dashed lines) are shown for context. AIRS profiles are conditioned on quartiles of daily-mean LWCRE from CERES SYN1Deg. At right are distributions of the (B) temperature ($c_pT$), (C) moisture ($L_vq$), and (D) geopotential ($gz$) components of MSE for Q4 from each reanalysis at 850 hPa (lower row), 500 hPa (centre row), and 300 hPa (upper row). Levels correspond to the horizontal yellow dashed lines in panel A. Mean values are marked as vertical lines; biases in these mean values relative to the mean value from ERA5 are color-coded at the upper left of each panel (each list from top: ERA-Interim, JRA-55, MERRA-2, CFSR).**

Fig. 13. What about the clear-sky OLR?

**AR: We have added a panel showing the evolution of anomalies in the clear-sky OLR averaged over the inner tropics. (Figure R5; now Fig 14 in the revised text).**

[Figure]

**Figure R5: Time series of deseasonalized anomalies in (A) monthly mean high cloud cover (HCC), (B) monthly mean OLR, and (C) monthly mean LWCRE averaged over the inner tropics (10°S–10°N) for 1980–2014 based on ERA5 (cyan), ERA-Interim (blue), JRA-55 (purple), MERRA-2 (red), and CFSR/CFSv2 (green). Observational analyses from CERES SYN1Deg (A; March 2000–December 2014) and CERES EBAF (B and C; March 2000–December 2014) are shown for context. Anomalies are calculated relative to the mean annual cycle during 2001–2014. Thick lines show time series after applying a 12-month uniformly-weighted rolling mean. Trends are listed for annual-mean anomalies during the (D) 1980–2014 and (E) 2001–2014 periods in percentage points per decade for HCC and units of W m⁻² per decade for OLR, clear-sky OLR, and LWCRE. Stars indicate statistical significance at the 90% (\*), 95% (\*\*), 99% (\*\*\*), and 99.5% (\*\*\*\*) confidence levels. Light grey shading indicates that the 90% confidence interval of the Theil–Sen slope contains zero. Blue colors mark negative trends and red colors positive trends, with darker shades signifying larger trend magnitudes (0 to 1, 1 to 2, 2 to 3, and greater than 3).**

2. A technical comment: it should be cautioned that CRE, defined as the difference between clear- and all-skies, is subject to influence of clear-sky [e.g., Soden 2004]. I'd suggest where appropriate (e.g., Fig. 5) clear-sky biases be also examined to ensure that the CRE difference measures cloud effect, instead of being affected by the clearsky differences between the reanalyses.

**AR: Thank you for raising this point. In the revised manuscript we have addressed differences in clear-sky radiative fluxes more directly, both by including them in Fig 5 (as contours in place of LWCRE) and by discussing the differences in both all-sky and clear-sky OLR in the text. Please see further discussion below.**

**Response to detailed comments:**

YH: P3, L5. Note there are methods, such as latent heat nudging and particle filter, that make use of cloud and precipitation info in data assimilation.
**AR: Thank you for mentioning this. We have added a sentence at the end of the paragraph: "Methods that directly make use of cloud or precipitation information in data assimilation, such as latent heat nudging or particle filters (e.g. Bannister et al., 2020), have yet to be implemented in global atmospheric reanalyses."**

R2: Page 4, l23: might it be clearer to say '… involved changes in the cloud fields which are much larger …'?
**AR: Changed as suggested.**

YH: P4, L31. Another benefit of simulator, if properly configured, is that it also addresses sampling consistency issue.
**R: We agree. We have added a sentence to clarify our reasoning: "Although use of a satellite simulator could address sensitivity and sampling biases for easier comparison with observations, it could also obscure inter-reanalysis differences in cloud populations that are not well observed and complicate analysis of cloud radiative effects within each reanalysis."**

R2: Page 4, l32: suggest '… we stress that most …' without 'the'.
**AR: Changed as suggested.**

YH: P4, L33. I'm surprised to read that the aim is stated to be "qualitative" – despite many quantitative – why?
**AR: We have included extensive quantitative comparisons amongst the reanalyses, and some with respect to OLR/LWCRE, but had focused on qualitative comparisons with respect to cloud observations (e.g. the approximate height, thickness, and distribution of anvil clouds in the time-mean tropical-mean, as opposed to the amount of cloud water or the magnitude of cloud fraction). One exception, of course, is Fig 2 which does use simulator output from MERRA-2, as well as the revised Fig 1. Following the revision of Fig 1 (see above), we have changed this sentence to read: "Accordingly, comparisons between reanalysis products and satellite cloud observations in this paper are not strictly like-with-like and should be interpreted with care."**

R2: Page 5, Table 1 / section 2.1: Table 1 provides a useful "at a glance" summary of the characteristics of the different observation datasets. There would be much to be said for extending it by the 5 or 6 lines needed to add equivalent details for the different reanalyses. I take the point made in section 2.1 that this paper provides detailed information of key aspects regarding cloud etc. and that other reviews have covered more general information, for those

wishing to spend the time to track it down, but a brief summary set of 'vital statistics' would reduce the reliance the paper is otherwise placing upon a reader's prior knowledge of the reanalyses.

**AR: Thank you for this suggestion. Because the most relevant information for the reanalyses differs from that for the observations, we have added a separate table summarizing key details of the reanalyses (Table R1) to section 2.1 of the manuscript.**

| Reanalysis | Model | Model Grid | HCC[a] | Profiles[a] | Fluxes[a] | Reference |
|---|---|---|---|---|---|---|
| ERA5 | IFS 41R2 (2016) | N320 ($\sim$31 km) 137 levels | $\sigma < 0.45$ 1-hourly | $T, q, \phi$, CC, I/LWC 3-hourly | TOA, RHR 12-h forecasts | Hersbach et al. (2018) |
| ERA-Interim | IFS 31R2 (2007) | N128 ($\sim$79 km) 60 levels | $\sigma < 0.45$ 6-hourly | $T, q, \phi$, CC, I/LWC 6-hourly | TOA, RHR 12-h forecasts | Dee et al. (2011) |
| JRA-55 | JMA GSM (2009) | N160 ($\sim$55 km) 60 levels | $p < 500$ hPa 3-hourly | $T, q, z$, CC, I/LWC 6-hourly | TOA, RHR 6-h forecasts | Kobayashi et al. (2015) |
| MERRA-2 | GEOS 5.12.4 (2015) | C180 ($\sim$50 km) 72 levels | $p < 400$ hPa 1-hourly | $T, q, z$, CC, I/LWC 3-hourly | TOA, RHR 3-h forecasts | Gelaro et al. (2017) |
| CFSR | NCEP CFS (2007) | F288 (0.3125°) 64 levels | $p < 400$ hPa 6-hourly | $T, q, z$, CWC 6-hourly | TOA, RHR 6-h forecasts | Saha et al. (2010) |
| CFSv2 | NCEP CFS (2011) | F440 (0.2045°) 64 levels | $p < 400$ hPa monthly | CWC monthly | TOA monthly | Saha et al. (2014) |

[a] HCC, CC, CWC (or I/LWC), and TOA variables from all reanalyses are also used at monthly resolution when constructing climatological means.

**Table R1: Summary of reanalysis products. HCC stands for high cloud fraction; CC for cloud fraction; CWC for cloud water content and I/LWC for separate ice and liquid water contents; TOA for top-of-atmosphere fluxes (shortwave and longwave; clear-sky and all-sky); RHR for radiative heating rates (shortwave and longwave; all-sky). We use CFSR products for 1980–2010, CFSv2 for 2011–2014, and all other reanalysis products for 1980–2014.**

YH: P6, L27; P13 L26. Note the latest CERES data includes a version of clear-sky values, computed using the same clear-sky definition as in GCM [Loeb 2019].

**AR: Thank you for bringing this to our attention. We have replaced CERES EBAF Ed4A with CERES EBAF Ed4.1 throughout the paper, with reference to Loeb et al. (2020) and the dataset citation. We have continued using the 'adjusted' fluxes from SYN1Deg for the analyses based on daily data.**

R2: Page 7, l3: do you intend 'specific heat constant' rather than 'heat capacity'?
**AR: Thanks for noticing this; we have changed it to 'specific heat capacity'.**

YH: P7, L15, Why linear with ln(p) instead of p?
**AR: This approach was adapted from code for identifying the tropopause height, for which it is advantageous to interpolate temperature and height to a high-resolution grid in $\ln(p)$ (or $z$), rather than in pressure coordinates directly. We have tested the sensitivity of the LZRH statistics to this choice and find little influence on the results (see Fig R6 for comparisons using the ERA5 and ERA-Interim products from 2005). As using $\ln(p)$ produces a slightly smoother distribution for ERA-Interim, we make no change to the method.**

[Figure]

**Figure R6: Distributions of LZRH locations in pressure calculated by conducting the interpolation (A,C) linearly in $\ln(p)$ versus (B,D) linearly in $p$ for (A,B) ERA5 and (C,D) ERA-Interim during 2005. Distributions for Q4 based on the LWCRE are shown in light purple. The median for each distribution is marked by the dash-dot line (purple for Q4; grey for the full distribution); the middle 50% (dark grey shading; 25th to 75th percentile) and middle 80% (light grey shading; 10th to 90th percentile) are marked in each panel.**

YH: P9, L8. Note that cloud top temperature (CTT) is another potential cause of (compensating) errors.

**AR: Thank you for mentioning this. We have added a sentence, so that the text now reads: "Bechtold et al. (2014) reported that changes to parameterized convection in the ECMWF atmospheric model implemented between ERA-Interim and ERA5 yielded lower biases against observed brightness temperatures in land convective regions, especially for channels sensitive to the upper troposphere. However, as differences in cloud top temperatures between the two model versions could also influence the simulated brightness temperatures, these lower biases cannot be directly attributed to changes in HCC."**

YH: P11, L22 and Fig. 4. Are the CWC averaged over only cloudy profiles or over both cloudy and clear profiles? Consistently between all the reanalyses? Both averages would be of interest to compare.

**AR: All reanalysis CWCs evaluated in this paper are grid-scale, not in-cloud products. We agree that both averages would be interesting to compare, but with length already an issue we only examine the grid-scale products in this paper.**

R2: Page 14, Fig5 caption: 'with values for OLR listed above …' actually look as though the OLR mean values on the Figure are beneath?

**AR: Yes, you are correct, and we have changed the caption accordingly. Please note that we have changed this figure (1) to add tropical-mean values for clear-sky OLR (at upper left), (2) to show contours for clear-sky OLR rather than LWCRE, and (3) to use CERES EBAF (updated to v4.1) as the benchmark with 2001-2014 as the comparison period instead of using MRM as the**

**benchmark with 1980-2014 as the comparison period. We still show area-mean values for LWCRE at the upper right of each panel (Figure R2).**

YH: P15, Fig. 6. How is the purple line drawn exactly?
**R: The purple line is the 75th percentile of all LWCREs included in the distribution (daily-mean products on a 1° grid between 10°S and 10°N from 1 January 2001 through 31 December 2010), meaning that 25% of the gridded daily-mean LWCREs are greater than this value.**

R2: Page 15, Fig 6 caption: 'is marked in the upper row.' Does this refer to the violet lines with numbers beside them in A, C-G?
**AR: Yes; we have changed the caption accordingly.**

R2: Page 15, l6: is 'are analogous to scatter plots containing millions of points' a roundabout way of saying 'represent a 2-dimensional probability density function'?
**AR: Yes; however, we have found that the analogy to scatter plots is helpful when introducing this type of plot. We have changed this to read "two-dimensional frequency distributions analogous to ..."**

R2: Page 17, l5: 'presumably owing' reads a little loosely to a reviewer. But more precisely, at the bottom of page 16 the point has been made that MERRA-2 can persist large cloud fraction for declining cloud water, which implies reduced in-cloud water contents that might well lead to different LWCRE. It is thus unclear when the issue of characteristic lifetimes appears in the text whether this is meant to have been inferred from the differences in in-cloud water content or a separate (and presumably verifiable) behaviour of the different schemes discussed that is however (not shown). This sentence would benefit from clarification.
**AR: Yes, this sentence was not worded well. We have changed it to read "despite this similar starting point, JRA-55 and MERRA-2 produce very different relationships between cloud fraction and condensate. Tuning efforts to increase the amount of cloud ice in the upper troposphere in MERRA-2 were motivated by a desire to improve OLR (knowing that the convective detrainment altitude is too low in GEOS5, the developers accepted overestimating cloud ice to get OLR right; personal communication from A. Molod). The cloud fraction was then kept small relative to the cloud ice content to prevent a worsening of the SWCRE as the LWCRE was increased." We have also corrected some descriptions of the MERRA-2 model in the preceding sentences.**

R2: Page 17, l13: 'more than 140Wm-2' seems a bit redundant when sitting beside 'from -100 to +40 Wm-2'
**AR: We have removed the phrase 'more than 140 W m$^{-2}$'.**

YH: P17, L30. OLR vs. CRE – this seems significant methodological difference. What's the rational to use CRE here?
**AR: Thank you for asking this question. The initial rationale for adopting LWCRE, like the use of ln($p$) above, was from a separate study; however, on further thought that rationale does not clearly apply to this analysis and OLR may indeed be the better choice. We have checked the**

sensitivity of the conditional composite and distribution results to using LWCRE instead of OLR. The results indicate that there is little sensitivity to this choice for the inner tropical band (10°S–10°N) that we focus on (see, e.g., Figure R7).

[Figure]

Figure R7: Composite mean profiles of daily-mean radiative heating rates as a function of pressure for the first through fourth quartiles (Q1–Q4) based on longwave cloud radiative effect (upper row; CRE1–CRE4), outgoing longwave radiation (centre row; OLR1–OLR4), and high cloud fraction (lower row; HCC1–HCC4) in the inner tropics (10°S–10°N) based on (left-to-right) ERA5, ERA-Interim, JRA-55, MERRA-2, and CFSR during 2001--2010. Here Q1 refers to the bottom quartile (weak LWCRE, large OLR, small HCC) and Q4 to the top quartile (strong LWCRE, small OLR, large HCC).

Using the bias-corrected Cramér's V ($\phi_c$; Bergsma, 2013), which tests agreement between two categorical variables, we find that the data points selected for Q4 are extremely similar between CRE and OLR, with $\phi_c > 0.9$ for each individual data set. Interpretation of $\phi_c$ is analogous to that for absolute linear correlations between numerical variables, with a value of 0 indicating no relationship and a value of 1 indicating perfect agreement. The overall quartile classification as a whole is also similar, with $\phi_c$ ranging from 0.7 (JRA-55) to 0.82 (MERRA-2). Given this similarity, particularly with Q4 as our primary condition for data selection, we prefer to minimize changes by continuing to use LWCRE instead of switching the condition to quantiles of OLR. This choice also fits well with the existing presentation, as the threshold values for LWCRE Q4 are already marked in Figure 6 of the original submission.

YH: P18, Fig. 7. I am surprised to see the lack of distinctions in the ERAi results here, reminiscent of fig. 10e of Zhang et al. [2017]. Some basic radiative signatures such as cloud top cooling and cloud bottom warming are totally missing. May this be related to the use of CRE as a state indicator - it may fail to identify cloud effect due to clear-sky bias difference (see comments above)? Would it be useful to simply use cloud fraction instead to identify the regimes (Q1-4)? How does cloudy heating rate profile compare to clear-sky in this reanalysis?

**AR: We initially did construct the composites based on high cloud fraction rather than LWCRE and have also tried OLR (Figure R7). The lack of distinction in ERA-Interim is a common feature in all three approaches, and ERA-Interim is clearly different from the other reanalyses in this regard. The results are more sensitive to replacing CRE with high cloud cover when constructing the quantiles than they are to replacing CRE with OLR, with $\phi_c$ ranging from 0.4 (MERRA-2) to 0.6 (CFSR) for the HCC classification relative to the CRE classification. Differences are especially stark for Q4, where the MERRA-2 results are altered substantially ($\phi_c \sim 0.3$; consistent with a relatively weak relationship between CWC and cloud fraction) but the CFSR results are hardly altered at all ($\phi_c \sim 0.9$; consistent with cloud fraction being explicitly tied to CWC in the large-scale cloud parameterization). Given these different relationships between CWC and HCC, as well as different lower bounds for defining the 'high cloud' layer (from $p < 500$ hPa to $p < 400$ hPa), we find LWCRE (or OLR) preferable to high cloud fraction for this purpose.**

**We have added a few sentences at the end of section 2.3 to summarize our responses to these two comments: "Results are very similar for ranked quartiles of all-sky OLR, with OLR reversed so that Q4 corresponds to the smallest values of OLR. Using HCC instead of LWCRE produces more substantial differences, particularly for MERRA-2. Given discrepancies in the precise definition of HCC across reanalyses (Table 1) and the difficulty of defining an appropriate observational benchmark for HCC, we judge HCC less suitable for this purpose. We select LWCRE rather than OLR for convenience of presentation."**

YH: P19, Fig. 8. Is it LW or net radiative heating that is used to define LZRH and the relevant results?

**AR: The LZRH is defined based on net radiative heating (SW+LW, all-sky). We have clarified this information in both section 2.3 and section 4.2.**

YH: P20, L20. Isn't it the diabatic heating (instead of radiative only) that better inferences ascent/descent? Why focused on radiative?

**AR: Yes, the ascent / descent is balanced by the total diabatic heating. However, overshooting convection that reaches the tropopause (or even the LZRH) is rare enough that ascent through the TTL is mostly balanced by radiative heating. As a result, some studies have used radiative heating alone to represent vertical transport in this region (e.g. Tzella and Legras, 2011; Tissier and Legras, 2016). These studies use satellite imagery to represent the convective sources of trajectories, thus avoiding uncertainties in whether the reanalysis puts convection in the correct locations with the correct depths and at the correct times. Focusing on radiative heating rates serves a similar purpose here, as it allows us to keep the scope of the paper manageable, avoiding some finer details of the convective parameterizations that are better treated elsewhere. Regardless of whether the all-sky radiative heating or total heating is used, the**

**LZRH is a critical level in that convective influences on the lower stratosphere should be dominated by detrainment occurring at or above the LZRH.**

YH: P21 "Possible origins". The discussions in this section don't distinguish cause and effect of previously presented cloud biases. Maybe worth some clarification and discussion here about this.

**AR: Yes, this is a good idea. We have added: "We cannot fully distinguish here between causes and effects. All of the variables we examine in this section are intimately connected to cloud and convection processes. Differences in these variables may therefore indicate the causes of cloud biases, reflect the effects of those same biases, or both of the above. To address this, we link differences in the examined variables to differences in the model parameterizations or data assimilation procedures whenever possible. Although we cannot unequivocally tie each bias to a single origin of this type, this information may be helpful both for understanding differences between the reanalyses and for highlighting potential targets for improvement in each reanalysis system."**

R2: Page 23, Table 2 caption: 'all data points' (plural)?
**AR: Yes; corrected.**

R2: Page 23, ll2-13: To me, lines 10-13 'Note … independent.' seem to follow more naturally from the statement of similarity that ends on line 4 '(Table 2).'
**AR: Changed as suggested.**

R2: In addition, the lines 4-7 seem to spend a lot of time simply repeating the data that readers can see in Table 2 whereas I think they would add more if they pulled out what seems to be the key message by showing the differences [Q4 –All] that appear scattered either side of the observed 1K. e.g. 'CFSR exhibits the weakest increase in SST (0.7K) between mean cloud and high …' or 'observation …, assigns a mean value to Q4 that is 1K warmer than the tropical mean.'
**AR: Changed as suggested: "The observation-based benchmark, with CERES SYN1Deg used to estimate daily-mean LWCRE and OISST v2 (Reynolds et al., 2007) for daily-mean SST, indicates a mean SST for Q4 that is 1 K warmer than the tropical mean. Q4 in CFSR exhibits the weakest increase relative to tropical mean SST (0.7 K), with values in the other reanalyses ranging from 0.9 K (JRA-55) to 1.2 K (ERA-Interim and MERRA-2)."**

YH: P24, L5. The way the current plot is made makes it difficult to discern the "kink".
**AR: We have added distributions of the three components of MSE at 850 hPa, 500 hPa, and 300 hPa for Q4 from each reanalysis (Figure R4), along with dashed yellow lines to highlight those three levels. The kink is located at the intersection of the thick purple line and the lowest dashed yellow line. We have added the sentence: "This kink arises because the Q4 profile in JRA-55 has a warm bias at 850 hPa (+0.4 kJ kg$^{-1}$ relative to ERA5; Fig 11B, lower row) but a cool and dry bias at 900 hPa (−1.0 kJ kg$^{-1}$; not shown)." The distributions at 900 hPa are omitted from the figure because they only appear in this comparison, whereas the biases at 850, 500, and 300 hPa are discussed in more detail in the text.**

R2: Page 25, l5: The first half of the first sentence would be better if merged into the following sentence 'Distributions of 500hPa grid-scale vertical velocity (w) for the whole tropics (Figure 9C) …' and the second half similarly into line 2 of P26.
**AR: Changed as suggested. We have also added a paragraph break between the discussion of 500-hPa vertical velocity and the discussion of mid-tropospheric RH.**

R2: Page 26, l16-18: there is a lot of repetition that could be reduced by swapping the order in line 16, for instance, 'so that plumes are only permitted to reach the upper troposphere when entrainment rates are small, that is potentially smaller than the … Tokioka parameter.'
**AR: We have changed this to "For MERRA-2 this is consistent with the application of a Tokioka-type entrainment condition (Bacmeister and Stephens, 2011): entrainment rates smaller than a randomly-selected minimum (the Tokioka parameter) are disallowed. For small values of RH, entrainment is efficient in diluting the updraft, so that plumes can only reach the upper troposphere when the entrainment rate is small. The application of the Tokioka condition thus tightens the preference for deeper convection to occur in more humid environments."**

YH: P26, section 5.2 and Fig 11. Besides the confusing way the figure is presented as noted above, I find the discussion here doesn't show enough recognition of the cloud position with respect to the respective levels focused (100 and 150 hPa). A basic signature of clouds is cloud top cooling and bottom warming. The sign and magnitude of the cloud radiative impact is strongly dependent on where the clouds are placed.
**AR: We have expanded on this discussion making reference to the former Fig 12 (now Fig 13), which clarifies the difference in cloud placement between strong positive and strong negative heating rates at 150 hPa. Specifically, we have revised the presentation of the results, and have included the following text at the beginning of the section: "The TTL is located above the typical levels of convective detrainment (200~300 hPa; Fig. 4), with a lower boundary near the LZRH (140~150 hPa; Fig. 9A). Clouds in this layer are most often associated with slow radiatively-balanced ascent, and occasionally with very deep convection that penetrates into the TTL (e.g. Fueglistaler et al., 2009). These two cloud populations are distinguished by their CWCs (smaller for in situ cirrus; larger for convective anvil clouds) and associated radiative heating rates (weak radiative heating for slow ascent; strong cloud-top cooling for most anvil clouds, potentially supplanted by strong warming for cloud layers at very high altitude). This essential radiative signature of cloud-top cooling and cloud-base heating can be seen by comparing the radiative heating profiles shown in Fig. 8A–E and the vertical locations of the anvil cloud layers shown in Fig. 4F. Radiative heating thus helps to distinguish different types of clouds in the lower part of the TTL: (1) in situ cirrus clouds, which are associated with weak positive heating rates balancing large-scale ascent (i.e. close to the 'spine' of the plot); (2) deep convection that detrains just below the TTL, which is associated with large CWCs and negative radiative heating (the left 'wing'); and (3) deep convection that detrains inside the TTL, which is associated with large CWCs and positive heating rates (the right 'wing'). The latter two types are distinguished by both the depth and water content of the anvil cloud (Fig. 13), and the third type grows increasingly rare with increasing altitude. Compositing on RH in addition to radiative heating helps to highlight some differences and unrealistic features among the reanalyses, as discussed below."**

YH: P29, L31. "systemic" => "artificial"?
AR: Changed as suggested.

YH: P29, L32 and Fig. 13. A striking feature is that OLR doesn't seem "jumped" despite of the cloud fraction jump! How could cloud cover change be consistent with OLR and CRE with regard to long-term trend but inconsistent with regard to this jump?
AR: Yes, this is one of several perplexing features of CFSR that we have so far been unable to pin down. One likely contributor is that improvements in humidity near the tropical tropopause (Figure R8A-C) led to increases in HCC above and outside of the core convective regions (cf. Figure R9A-C). Although CFSR/CFSv2 does not provide a vertically resolved estimate of cloud fraction, we do find increases in cloud water content near the tropopause and outside the deep tropics (Figure R8D-F). Cloud fractions in CFSR are determined primarily as a function of CWC (with RH a contributing factor), so that these differences (and/or any undocumented changes in the relationship between CWC/RH and cloud fraction) could result in large changes in HCC but relatively little change in LWCRE. Another possibility (not mutually exclusive) is that tuning of the CFSv2 model following the resolution and physical parameterization changes (e.g. the introduction of McICA) smoothed out the discontinuity in OLR despite the jump in high cloud fraction. In this sense, it is interesting to note that there was a jump of more than 7 W m$^{-2}$ in the net CRE between the last four years of CFSR and the first four years of CFSv2, indicating substantial changes in the SWCRE (Figure R9H). These changes must include the effects of model changes targeting marine low-level clouds (Saha et al., 2014); however, it is clear from the spatial distribution that reductions in planetary albedo are concentrated in places where high clouds are more prevalent (including canonical deep convective regions and locations where increases in HCC are relatively large). It is unclear how increases in HCC and upper-level CWC lead to an unchanged LW effect and a *reduced* planetary albedo – we have double- and triple-checked the data processing history and found no obvious errors. Indeed, the fluxes make sense internally for both CFSR and CFSv2; it is only when we evaluate the changes at the transition that this inconsistency crops up, implicating changes in the model. Despite the lack of a clear explanation, we have added these figures and a tighter version of this text as an appendix to the paper to support the discussion of discontinuities at the CFSR-CFSv2 transition.

Some unique behaviors in CFSR also seem to emerge from the bias correction scheme. These features show up as 'blips' in the time series after every production stream transition, as the model gradually imprints its own bias on any variables that are not sufficiently constrained by data assimilation (the spin-up period seems to be just about long enough to reset water vapor in the stratosphere to zero as shown by Davis et al., 2017; a similar issue might affect clouds, especially at upper levels). This could explain why we see such a large difference in these variables for the 2010 bridge year, since that bridge year was run without spin-up (see Long et al., 2017). Unfortunately, it is not documented whether the model used for 2010 included any changes relative to the original model, and we have so far not managed to confirm any of these speculations with the model development group.

[Figure]

**Figure R8:** Upper row: Zonal-mean distributions of RH based on (A) the last four years of the original CFSR (2006–2009) and (B) the first four years of CFSv2 (2011–2014), along with (C) differences between the two products. Lower row: as in the upper row, but for CWC.

[Figure]

**Figure R9:** At left: distributions of high cloud fraction based on (A) the last four years of the original CFSR (2006–2009) and (B) the first four years of CFSv2 (2011–2014), along with (C) the difference between CFSv2 and CFSR. The change in ISCCP HGM high cloud fraction between the 2006–2009 mean and the 2011–2014 mean is shown for context in panel (D). At right: as in the left column, but for all-sky (shading) and clear-sky (contours) OLR. The change in all-sky and clear-sky upward SW flux between CFSR and CFSv2 is also shown in panel (H). Tropical mean (30°S-30°N) values of HCC (or ΔHCC) are listed at upper right of panels (A) through (D). Tropical mean values of OLR, clear-sky OLR, LWCRE, and net CRE (or corresponding Δ values) are listed at upper right (OLR and clear-sky OLR) or upper left (LWCRE and net CRE) of panels (E) through (G). These are replaced with mean changes in TOA upward SW fluxes in panel (H).

YH: Also, the long-term trends seem similar among the reanalyses – should this be taken seriously as a sign of real trend in nature? It is important to reason and caution whether we can use these reanalyses for studying climate trend in this critical region. It should be noted that the all-sky OLR trend appears to contradict the FAT hypothesis [Hartmann & Larson 2002].

**AR: We are skeptical of the trends, especially without more robust independent evidence from observations and extension to other periods. As noted in the text, it is not surprising that reanalyses produce declining trends in clear-sky OLR over periods ending in 2014, and the further contribution to decreases in all-sky OLR from increases in high cloud cover is not consistently supported by observations. It is a good idea to mention the FAT hypothesis for context here, although the all-sky OLR trend only contradicts FAT to the extent that it cannot be explained by changes in high cloud cover. We have added the text: "Decreasing trends in all-sky OLR seem at first glance to contradict the fixed anvil temperature (FAT) hypothesis of Hartmann and Larson (2002). However, increasing trends in HCC are qualitatively consistent with decreases in all-sky OLR above and beyond those in clear-sky OLR; reductions in all-sky OLR therefore do not necessarily imply reductions in anvil cloud emission temperatures. Indeed, with the exception of CFSR/CFSv2 (affected by discontinuities around the CFSR–CFSv2 transition as discussed in Appendix B) and ERA5 (for which trends are small), relatively large decreasing trends in all-sky OLR among reanalyses reflect relatively large increases in LWCRE, which are linked in turn to increases in HCC."**

YH: P30, L23. The clear-sky OLR change described sounds very interesting and ought to be shown. Relevant to the above point, another important question the reanalyses may or may not answer is whether broadband fluxes, either clear- or all-sky, may be useful for climate change monitoring. As shown by Huang & Ramaswamy [2009, Fig. 5], there may be intrinsic compensation between greenhouse gas forcing and Planck response that results in no trend signal. This point, together with the above one, is worth noting and discussing here.

**AR: We have added a panel showing the evolution of anomalies in clear-sky OLR (Fig. R5). We have expanded the text here to read: "Decreasing trends in clear-sky OLR suggest that increases in atmospheric greenhouse gas absorption outpace increases in the effective emission temperature in the tropics over this period, changes that may be explained by the so-called `hiatus' in surface warming during the early 2000s (Song et al., 2016). Prescribed greenhouse gas concentrations in the reanalyses increased throughout this period (Fujiwara et al., 2017, their Fig. 4), as observed surface temperatures cooled or stayed roughly constant through much of the tropics for more than a decade (Kosaka and Xie, 2013). Although these trends should be interpreted with caution, their consistency with expectations is a promising sign for the use of broadband OLR fluxes in climate monitoring, given the potential for compensating effects to damp signals of climate change in these fields (e.g. Huang and Ramaswamy, 2009)."**

YH: P31, Fig. 13. Some of the time series apparently don't have zero mean. How are the anomalies defined?

**AR: Anomalies are defined relative to the mean annual cycle during 2001-2014. This information is included in the caption and in the text at the beginning of section 6.**

R2: Page 33, Figure 14 caption: are the separation lines 'Solid grey' or 'Solid black'? (I do like the plot, though!).
**AR: We have changed this to 'solid black'.**

YH: P34, Summary. A general suggestion for this section is to reference the respective summary points to the relevant figures.
**AR: We have added figure references to the text in section 7.**

YH: P34, L20. It is striking to find the lack of agreement among the studies in terms of what direction cloud drives the LZRH. Can you discuss why and how would one elucidate this matter?
**AR: We agree! This is a troubling discrepancy, not least as the recent ERA5 agrees with the other reanalyses rather than ERA-Interim, in contrast to what the observation-based calculations suggest. However, to answer this question beyond the short discussion at the end of this paragraph would require a detailed evaluation of the relative uncertainties and deficiencies associated with the observational products (all subject to sampling biases and assumptions about cloud properties) that lies beyond the scope of this paper. We are aware of colleagues who are digging into this question and eagerly await their results.**

R2: Page 36, l18: 'occur in …' rather than 'into'?
**AR: Yes, thank you.**

YH: P36, L29. Cloud top temperature, as related to in some of the above comments, is perhaps another aspect to note.
**AR: We have revised this paragraph to mention cloud top temperature in conjunction with cloud top height throughout.**

R2: Page 36, l33: I found this sentence somewhat hard to digest, especially as a finale. I would suggest putting a stop in at 'in the reanalysis models.' then rewording the rest into a following sentence as appropriate.
**AR: Changed as suggested. The final sentence now reads: "Further investigation along these lines should also consider how these features can imprint upon more widely-used reanalysis products and influence model simulations that use reanalysis fields to drive atmospheric transport."**

YH: P37, Appendix. It may be worth reviewing the difference in assimilated data in this appendix as well. This is apparently relevant to the trend discussions (section 6) and potential affects climatology as well.
**AR: With the exception of ERA5, differences in assimilated data among these reanalyses have been covered by Fujiwara et al. (2017) on conventional observations and satellite radiances, Long et al. (2017) on observations influencing temperature, Davis et al. (2017) on observations of water vapor and ozone, Tegtmeier et al. (2020) on TTL thermal structure (in this case including ERA5), and others. Chapter 2 of the S-RIP report includes 28 pages on this topic. Given the range of observation types that may contribute to biases in clouds, OLR, and heating rates,**

we are unsure how to condense this information into a reasonable length for this paper. We prefer to direct readers to these other resources (and the ERA5 documentation) instead.

YH: P37, L25. What is liquid water temperature?
AR: Liquid water temperature is defined as the air temperature minus $(L_v/c_p) * q_c$, where $L_v$ is the latent heat of vaporization, $c_p$ is the specific heat capacity at constant pressure, and $q_c$ is the cloud water content. We have added this clarification to the appendix.

R2: Page 42, l3: 'discrete' is redundant here as 'discretization' guarantees it!
AR: We have changed 'discrete' to 'predetermined'.

YH: P43, L1. Sufficient info to ensure reproducibility of the results should be included. Regarding the data sources, how were the data, such as ERA5 heating rates, obtained exactly, as they are not normally available from the webpages stated here? If scripts were used, it is useful to post a sample script and explain how relevant parameters, e.g., analysis vs. forecast and, if latter, forecast times and steps, are set. Moreover, are the these parameters set consistently for all the variables: heating rate and state variables such as cloud fraction, temperature, humidity, etc. from the same time steps?
AR: We have added information on the temporal resolution of each product in the data availability statement as well as in the new Table 1 (see Table R1 above). Heating rates and TOA fluxes are in all cases based on time-average forecast fields which are then aggregated into daily means, while other data are typically instantaneous outputs at 1-hourly, 3-hourly, or 6-hourly resolution. We have sub-sampled the ERA5 pressure-level analysis fields to 3-hourly resolution, which matches MERRA-2; all other vertically resolved fields (from ERA-Interim, JRA-55, and CFSR) are only provided 6-hourly. High cloud fraction sampling ranges from hourly (ERA5, MERRA-2) to 3-hourly (JRA-55) to 6-hourly (ERA-Interim, CFSR); this difference in sampling potentially impacts values along the y-axis in the joint distributions shown in Fig 6 (and Fig R4 above), but should not influence any other calculations in the paper.

**Additional changes:**
In addition to the above changes, we have replaced the July 2006–June 2007 CWC-RO profile of IWC in Figure 3 with a 2007–2010 mean profile based on 2C-ICE (Deng et al., 2015) and added one co-author. The profile based on 2C-ICE shows quantitative differences relative to the previous CWC-RO profile, but introduces no changes in interpretation. We elect to make this change for two reasons: (1) the longer period, which is consistent with the KG2009 cloud fraction profile and (2) a clearer data provenance and processing history.

We have also made changes to shorten the text to partially compensate for additions committed to the manuscript during these revisions.